# Statistical Perspective of Top-K Sparse Softmax Gating Mixture of Experts

**Huy Nguyen**

Department of Statistics and Data Sciences

The University of Texas at Austin

Austin, TX 78712

`huynm@utexas.edu`

**Pedram Akbarian**

Department of Electrical and Computer Engineering

The University of Texas at Austin

Austin, TX 78712

`akbarian@utexas.edu`

**Fanqi Yan**

Department of Computer Science

The University of Texas at Austin

Austin, TX 78712

`fanqi.yan@utexas.edu`

**Nhat Ho**

Department of Statistics and Data Sciences

The University of Texas at Austin

Austin, TX 78712

`minhnhat@utexas.edu`

## Abstract

Top-K sparse softmax gating mixture of experts has been widely used for scaling up massive deep-learning architectures without increasing the computational cost. Despite its popularity in real-world applications, the theoretical understanding of that gating function has remained an open problem. The main challenge comes from the structure of the top-K sparse softmax gating function, which partitions the input space into multiple regions with distinct behaviors. By focusing on a Gaussian mixture of experts, we establish theoretical results on the effects of the top-K sparse softmax gating function on both density and parameter estimations. Our results hinge upon defining novel loss functions among parameters to capture different behaviors of the input regions. When the true number of experts $k_*$ is known, we demonstrate that the convergence rates of density and parameter estimations are both parametric on the sample size. However, when $k_*$ becomes unknown and the true model is over-specified by a Gaussian mixture of $k$ experts where $k > k_*$, our findings suggest that the number of experts selected from the top-K sparse softmax gating function must exceed the total cardinality of a certain number of Voronoi cells associated with the true parameters to guarantee the convergence of the density estimation. Moreover, while the density estimation rate remains parametric under this setting, the parameter estimation rates become substantially slow due to an intrinsic interaction between the softmax gating and expert functions.

## 1 Introduction

Introduced by Jacobs et al. (1991) and Jordan & Jacobs (1994), the mixture of experts (MoE) has been known as a statistical machine learning design that leverages softmax gating functions to blend different expert networks together in order to establish a more intricate model. Recently, there has been a huge interest in a variant of this model called top-K sparse softmax gating MoE, which activates only the best $K$ expert networks for each input based on sparse gating functions (Shazeer et al., 2017; Fedus et al., 2022a; Chen et al., 2023). Thus, this surrogate can be seen as a form of conditional computation (Bengio, 2013; Cho & Bengio, 2014) since it significantly scales up the model capacity while avoiding a proportional increase in the computational cost. These benefits have been empirically demonstrated in several deep learning applications, including natural language processing (Lepikhin et al., 2021; Du et al., 2022; Fedus et al., 2022b; Zhou et al., 2023; Pham et al.,

2024), speech recognition (Peng et al., 1996; Gulati et al., 2020; You et al., 2022), computer vision (Dosovitskiy et al., 2021; Riquelme et al., 2021; Liang et al., 2022; Bao et al., 2022), multi-task learning (Hazimeh et al., 2021; Gupta et al., 2022) and other applications (Rives et al., 2021; Chow et al., 2023; Li et al., 2023; Han et al., 2024). Nevertheless, to the best of our knowledge, the full theoretical analysis of the top-K sparse softmax gating function has remained missing in the literature. In this paper, we aim to shed new light on the theoretical understanding of that gating function from a statistical perspective via the convergence analysis of maximum likelihood estimation (MLE) under the top-K sparse softmax gating Gaussian MoE.

There have been previous efforts to study the parameter estimation problem in Gaussian MoE models. Firstly, Ho et al. (2022) utilized the generalized Wasserstein loss functions Villani (2003; 2008) to derive the rates for estimating parameters in the input-free gating Gaussian MoE. They pointed out that due to an interaction among expert parameters, these rates became increasingly slow when the number of extra experts rose. Subsequently, Nguyen et al. (2023) and Nguyen et al. (2024) took into account the Gaussian MoE with softmax gating and Gaussian gating functions, respectively. Since these gating functions depended on the input, another interaction between gating parameters and expert parameters arose. Therefore, they designed Voronoi loss functions to capture these interactions and observe that the convergence rates for parameter estimation under these settings can be characterized by the solvability of systems of polynomial equations (Sturmfels, 2002).

Turning to the top-K sparse softmax gating Gaussian MoE, the convergence analysis of the MLE, however, becomes substantially challenging due to the sophisticated structure of the top-K sparse softmax gating function compared to those of softmax gating and Gaussian gating functions. To comprehend these obstacles better, let us introduce the formal formulation of that model.

**Problem setup.** Suppose that $(X_1, Y_1), \ldots, (X_n, Y_n) \in \mathbb{R}^d \times \mathbb{R}$ are i.i.d. samples of size $n$ from the top-K sparse softmax gating Gaussian MoE of order $k_*$ with the conditional density function

$$g_{G_*}(Y|X) = \sum_{i=1}^{k_*} \mathrm{Softmax}(\mathrm{TopK}((\beta_{1i}^*)^\top X, K; \beta_{0i}^*)) \cdot f(Y|(a_i^*)^\top X + b_i^*, \sigma_i^*), \tag{1}$$

where $G_* := \sum_{i=1}^{k_*} \exp(\beta_{0i}^*) \delta_{(\beta_{1i}^*, a_i^*, b_i^*, \sigma_i^*)}$ is a true but unknown *mixing measure* of order $k_*$ (i.e., a linear combination of $k_*$ Dirac measures $\delta$) associated with true parameters $(\beta_{0i}^*, \beta_{1i}^*, a_i^*, b_i^*, \sigma_i^*)$ for $i \in \{1, 2, \ldots, k_*\}$. Here, $h_1(X, a, b) := a^\top X + b$ is referred to as an expert function, while we denote $f(\cdot|\mu, \sigma)$ as an univariate Gaussian density function with mean $\mu$ and variance $\sigma$ (The results for other settings of $f(\cdot|\mu, \sigma)$ are in Appendix E). Additionally, we define for any vectors $v = (v_1, \ldots, v_{k_*})$ and $u = (u_1, \ldots, u_{k_*})$ in $\mathbb{R}^{k_*}$ that $\mathrm{Softmax}(v_i) := \exp(v_i)/\sum_{j=1}^{k_*} \exp(v_j)$ and

$$\mathrm{TopK}(v_i, K; u_i) := \begin{cases} v_i + u_i, & \text{if } v_i \text{ is in the top } K \text{ elements of } v; \\ -\infty, & \text{otherwise.} \end{cases}$$

More specifically, before applying the softmax function in equation (1), we keep only the top $K$ values of $(\beta_{11}^*)^\top X, \ldots, (\beta_{1k_*}^*)^\top X$ and their corresponding bias vectors among $\beta_{01}^*, \ldots, \beta_{0k_*}^*$, while we set the rest to $-\infty$ to make their gating values vanish. An instance of the top-K sparse softmax gating function is given in equation (3). Furthermore, linear expert functions considered in equation (1) are only for the simplicity of presentation. With similar proof techniques, the results in this work can be extended to general settings of the expert functions, including deep neural networks. In order to obtain an estimate of $G_*$, we use the following maximum likelihood estimation (MLE):

$$\widehat{G}_n \in \arg\max_G \frac{1}{n} \sum_{i=1}^n \log(g_G(Y_i|X_i)). \tag{2}$$

Under the *exact-specified* settings, i.e., when the true number of expert $k_*$ is known, the maximum in equation (2) is subject to the set of all mixing measures of order $k_*$ denoted by $\mathcal{E}_{k_*}(\Omega) := \{G = \sum_{i=1}^{k_*} \exp(\beta_{0i}) \delta_{(\beta_{1i}, a_i, b_i, \sigma_i)} : (\beta_{0i}, \beta_{1i}, a_i, b_i, \sigma_i) \in \Omega\}$. On the other hand, under the *over-specified* settings, i.e., when $k_*$ is unknown and the true model is over-specified by a Gaussian mixture of $k$ experts where $k > k_*$, the maximum is subject to the set of all mixing measures of order at most $k$, i.e., $\mathcal{O}_k(\Omega) := \{G = \sum_{i=1}^{k'} \exp(\beta_{0i}) \delta_{(\beta_{1i}, a_i, b_i, \sigma_i)} : 1 \le k' \le k, \ (\beta_{0i}, \beta_{1i}, a_i, b_i, \sigma_i) \in \Omega\}$.

**Universal assumptions.** For the sake of theory, we impose four main assumptions on the parameters:

**(U.1) Convergence of MLE:** To ensure the convergence of parameter estimation, we assume that the parameter space $\Omega$ is compact subset of $\mathbb{R} \times \mathbb{R}^d \times \mathbb{R}^d \times \mathbb{R} \times \mathbb{R}_+$, while the input space $\mathcal{X}$ is bounded.

**(U.2) Identifiability:** Next, we assume that $\beta^*_{1k_*} = \mathbf{0}_d$ and $\beta^*_{0k_*} = 0$ so that the top-K sparse softmax gating Gaussian mixture of experts is identifiable. Under that assumption, we show in Appendix C that if $G$ and $G'$ are two mixing measures such that $g_G(Y|X) = g_{G'}(Y|X)$ for almost surely $(X, Y)$, then it follows that that $G \equiv G'$. Without that assumption, the result that $G \equiv G'$ does not hold, which leads to unncessarily complicated loss functions (see [Proposition 1,Nguyen et al. (2023)]).

**(U.3) Distinct Experts:** To guarantee that experts in the mixture (1) are different from each other, we assume that parameters $(a^*_1, b^*_1, \sigma^*_1), \ldots, (a^*_{k_*}, b^*_{k_*}, \sigma^*_{k_*})$ are pairwise distinct.

**(U.4) Input-dependent Gating Functions:** To make sure that the gating functions depend on the input $X$, we assume that at least one among parameters $\beta^*_{11}, \ldots, \beta^*_{1k_*}$ is different from zero. Otherwise, the gating functions would be independent of the input $X$, which simplifies the problem significantly. In particular, the model (1) would reduce to an input-free gating Gaussian mixture of experts, which was already studied in Ho et al. (2022).

**Challenge discussion.** In our convergence analysis, there are two main challenges attributed to the structure of the top-K sparse softmax gating function. **(1)** First, since this gating function divides the input space into multiple regions and each of which has different convergence behavior of density estimation, there could be a mismatch between the values of the top-K sparse softmax gating function in the density estimation $g_{\widehat{G}_n}$ and in the true density $g_{G_*}$ (see Figure 1). **(2)** Second, under the over-specified settings, each component of $G_*$ could be fitted by at least two components of $\widehat{G}_n$. Therefore, choosing only the best $K$ experts in the formulation of $g_{\widehat{G}_n}(Y|X)$ is insufficient to estimate the true density $g_{G_*}(Y|X)$. As a result, it is of great importance to figure out the minimum number of experts selected in the top-K sparse softmax gating function necessary for ensuring the convergence of density estimation.

**Contributions.** In this work, we provide rigorous statistical guarantees for density estimation and parameter estimation in the top-K sparse softmax gating Gaussian MoE under both the exact-specified and over-specified settings. Our contributions are two-fold and can be summarized as follows (see also Table 1):

**1. Exact-specified settings:** When the true number of experts $k_*$ is known, we point out that the density estimation $g_{\widehat{G}_n}$ converges to the true density $g_{G_*}$ under the Hellinger distance $h$ at the parametric rate, that is, $\mathbb{E}_X[h(g_{\widehat{G}_n}(\cdot|X), g_{G_*}(\cdot|X))] = \widetilde{\mathcal{O}}(n^{-1/2})$. Then, we propose a novel Voronoi metric $\mathcal{D}_1$ defined in equation (5) to characterize the parameter estimation rates. By establishing the Hellinger lower bound $\mathbb{E}_X[h(g_G(\cdot|X), g_{G_*}(\cdot|X))] \gtrsim \mathcal{D}_1(G, G_*)$ for any mixing measure $G \in \mathcal{E}_{k_*}(\Theta)$, we arrive at another bound $\mathcal{D}_1(\widehat{G}_n, G_*) = \widetilde{\mathcal{O}}(n^{-1/2})$, which indicates that the rates for estimating true parameters $\exp(\beta^*_{0i}), \beta^*_{1i}, a^*_i, b^*_i, \sigma^*_i$ are of the optimal order $\widetilde{\mathcal{O}}(n^{-1/2})$.

**2. Over-specified settings:** When $k_*$ is unknown and the true model is over-specified by a Gaussian of $k$ experts where $k > k_*$, we demonstrate that the density estimation $g_{\widehat{G}_n}$ converges to the true density $g_{G_*}$ only if the number of experts chosen from $g_{\widehat{G}_n}$, denoted by $\overline{K}$, is greater than the total cardinality of a certain number of Voronoi cells generated by the support of $G_*$. Given these results, the density estimation rate is shown to remain parametric on the sample size. Additionally, by designing a novel Voronoi loss function $\mathcal{D}_2$ in equation (8) to capture an interaction between parameters of the softmax gating and expert functions, we prove that the MLE $\widehat{G}_n$ converges to the true mixing measure $G_*$ at a rate of $\widetilde{\mathcal{O}}(n^{-1/2})$. Then, it follows from the formulation of $\mathcal{D}_2$ that the estimation rates for true parameters $\beta^*_{1j}, a^*_j, b^*_j, \sigma^*_j$ depend on the solvability of a system of polynomial equations arising from the previous interaction, and turn out to be significantly slow.

**High-level proof ideas.** Following from the challenge discussion, to ensure the convergence of density estimation under the exact-specified settings (resp. over-specified settings), we show that the input regions divided by the true gating functions match those divided by the estimated gating functions in Lemma 1 (resp. Lemma 2). Then, we leverage fundamental results on density estimation for M-estimators in (van de Geer, 2000) to derive the parametric density estimation rate under the Hellinger distance in Theorem 1 (resp. Theorem 3). Regarding the parameter estimation problem, a key step is to decompose the density discrepancy $g_{\widehat{G}_n}(Y|X) - g_{G_*}(Y|X)$ into a combination of

Table 1: Summary of density estimation and parameter estimation rates under the top-K sparse softmax gating Gaussian MoE model. Here, the function $\bar{r}(\cdot)$ stands for the solvability of the system of polynomial equations (7), and $\mathcal{C}_j$ is a Voronoi cell defined in equation (4). If $|\mathcal{C}_j| = 2$, then we have $\bar{r}(|\mathcal{C}_j|) = 4$. Meanwhile, if $|\mathcal{C}_j| = 3$, then we get $\bar{r}(|\mathcal{C}_j|) = 6$.

| **Setting** | $g_{G_*}(Y|X)$ | $\exp(\beta_{0j}^*)$ | $\beta_{1j}^*, b_j^*$ | $a_j^*, \sigma_j^*$ |
|---|---|---|---|---|
| Exact-specified | $\mathcal{O}(n^{-1/2})$ | $\mathcal{O}(n^{-1/2})$ | $\mathcal{O}(n^{-1/2})$ | $\mathcal{O}(n^{-1/2})$ |
| Over-specified | $\mathcal{O}(n^{-1/2})$ | $\mathcal{O}(n^{-1/2})$ | $\mathcal{O}(n^{-1/2\bar{r}(|\mathcal{C}_j|)})$ | $\mathcal{O}(n^{-1/\bar{r}(|\mathcal{C}_j|)})$ |

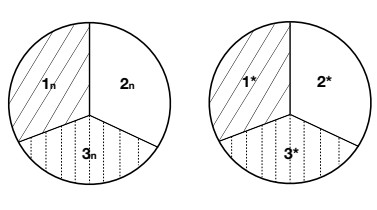

Figure 1: Illustration of two partitions of the input space with respect to the $\mathrm{TopK}$ function in the density estimation $g_{\widehat{G}_n}$ *(left)* and the true density $g_{G_*}$ *(right)* under the exact-specified settings when $k_* = 3$ and $K = 1$. Here, the regions labelled as $\mathbf{1_n}$ and $\mathbf{1^*}$ contain $X \in \mathcal{X}$ such that $(\widehat{\beta}_{11}^n)^\top X$ and $(\beta_{11}^*)^\top X$ are the top-1 elements of $((\widehat{\beta}_{1i}^n)^\top X)_{i=1}^3$ and $((\beta_{1i}^*)^\top X)_{i=1}^3$, respectively. Other regions are defined similarly. Assume that $\widehat{\beta}_{1i}^n \to \beta_{1i}^*$ as $n \to \infty$ for any $i \in \{1, 2, 3\}$, then the regions $\mathbf{1_n}, \mathbf{2_n}, \mathbf{3_n}$ should respectively match their counterparts $\mathbf{1^*}, \mathbf{2^*}, \mathbf{3^*}$ to guarantee the convergence of $g_{\widehat{G}_n}$ to $g_{G_*}$. Lemma 1 reads that this property holds when the sample size $n$ is sufficiently large.

linearly independent terms. Thus, when the density estimation $g_{\widehat{G}_n}(Y|X)$ converges to the true density $g_{G_*}(Y|X)$, the coefficients in that combination also tend to zero, which leads to our desired parameter estimation rates in Theorem 2 (resp. Theorem 4).

**Organization.** The rest of our paper is organized as follows. In Section 2, we establish the convergence rates of density estimation and parameter estimation for the top-K sparse softmax gating Gaussian MoE under the exact-specified settings, while the results for the over-specified settings are presented in Section 3. Subsequently, we provide practical implications of those results in Section 4 before concluding the paper and discussing some future directions in Section 5. Finally, full proofs, numerical experiments and additional results are deferred to the supplementary material.

**Notations.** For any natural numbers $m \geq n$, we denote $[n] := \{1, 2, \ldots, n\}$ and $\binom{m}{n} := \frac{m!}{n!(m-n!)}$. Next, for any vector $u, v \in \mathbb{R}^d$, we let $|u| := \sum_{i=1}^d u_i$, $u! := u_1! \ldots u_d!$, $u^v := u_1^{v_1} \ldots u_d^{v_d}$ and denote $\|u\|$ as its 2-norm value. Meanwhile, we define $|A|$ as the cardinality of some set $A$. Then, for any two probability densities $f_1$ and $f_2$ dominated by the Lebesgue measure $\nu$, we denote $V(f_1, f_2) := \frac{1}{2} \int |f_1 - f_2| \mathrm{d}\nu$ as their Total Variation distance, whereas $h^2(f_1, f_2) := \frac{1}{2} \int (\sqrt{f_1} - \sqrt{f_2})^2 \mathrm{d}\nu$ represents the squared Hellinger distance. Finally, for any two sequences of positive real numbers $(a_n)$ and $(b_n)$, the notations $a_n = \mathcal{O}(b_n)$ and $a_n \lesssim b_n$ both stand for $a_n \leq C b_n$ for all $n \in \mathbb{N}$ for some constant $C > 0$, while the notation $a_n = \widetilde{\mathcal{O}}(b_n)$ indicates that the previous inequality may depend on some logarithmic term.

## 2 EXACT-SPECIFIED SETTINGS

In this section, we characterize respectively the convergence rates of density estimation and parameter estimation in the top-K sparse softmax gating Gaussian MoE under the exact-specified settings, namely when the true number of experts $k_*$ is known.

To begin with, let us introduce some essential notations and key results to deal with the top-K sparse softmax gating function. It can be seen from equation (1) that whether $(a_i^*)^\top X + b_i^*$ belongs to the top $K$ experts in the true density $g_{G_*}(Y|X)$ or not is determined based on the ranking of $(\beta_{1i}^*)^\top X + \beta_{0i}^*$,

which depends on the input $X$. Additionally, it is also worth noting that there are a total of $q := \binom{k_*}{K}$ different potential choices of top $K$ experts. Thus, we first partition the input space $\mathcal{X}$ into $q$ regions in order to specify the top $K$ experts and obtain an according representation of $g_{G_*}(Y|X)$ in each region. In particular, for each $\ell \in [q]$, we denote $\{\ell_1, \ell_2, \ldots, \ell_K\}$ as an $K$-element subset of $[k_*]$ and $\{\ell_{K+1}, \ldots, \ell_{k_*}\} := [k_*] \setminus \{\ell_1, \ell_2, \ldots, \ell_K\}$. Then, the $\ell$-th region of $\mathcal{X}$ is defined as

$$\mathcal{X}_\ell^* := \left\{ x \in \mathcal{X} : (\beta_{1i}^*)^\top x \geq (\beta_{1i'}^*)^\top x, \forall i \in \{\ell_1, \ldots, \ell_K\}, i' \in \{\ell_{K+1}, \ldots, \ell_{k_*}\} \right\},$$

for any $\ell \in [q]$. From this definition, we observe that if $X \in \mathcal{X}_\ell^*$ for some $\ell \in [q]$ such that $\{\ell_1, \ldots, \ell_K\} = [K]$, then it follows that $\mathrm{TopK}((\beta_{1i}^*)^\top X, K; \beta_{0i}^*) = (\beta_{1i}^*)^\top X + \beta_{0i}^*$ for any $i \in [K]$. As a result, $(a_1^*)^\top X + b_1^*, \ldots, (a_K^*)^\top X + b_K^*$ become the top $K$ experts, and the true conditional density $g_{G_*}(Y|X)$ is reduced to:

$$g_{G_*}(Y|X) = \sum_{i=1}^K \frac{\exp((\beta_{1i}^*)^\top X + \beta_{0i}^*)}{\sum_{j=1}^K \exp((\beta_{1j}^*)^\top X + \beta_{0j}^*)} \cdot f(Y|(a_i^*)^\top X + b_i^*, \sigma_i^*). \tag{3}$$

Subsequently, we assume that the MLE $\widehat{G}_n$ takes the form $\widehat{G}_n := \sum_{i=1}^{k_*} \exp(\widehat{\beta}_{0i}^n) \delta_{(\widehat{\beta}_{1i}^n, \widehat{a}_i^n, \widehat{b}_i^n, \widehat{\sigma}_i^n)}$, where the MLE's component $\widehat{\omega}_i^n := (\widehat{\beta}_{0i}^n, \widehat{\beta}_{1i}^n, \widehat{a}_i^n, \widehat{b}_i^n, \widehat{\sigma}_i^n)$ converges to the true component $\omega_i^* := (\beta_{0i}^*, \beta_{1i}^*, a_i^*, b_i^*, \sigma_i^*)$ for any $i \in [k_*]$. We figure out in the following lemma a relation between the values of the TopK function in $g_{G_*}(Y|X)$ and $g_{\widehat{G}_n}(Y|X)$:

**Lemma 1.** *For any $i \in [k_*]$, let $\beta_{1i}, \beta_{1i}^* \in \mathbb{R}^d$ such that $\|\beta_{1i} - \beta_{1i}^*\| \leq \eta_i$ for some sufficiently small $\eta_i > 0$. Then, for any $\ell \in [q]$, unless the set $\mathcal{X}_\ell^*$ has measure zero, we obtain that $\mathcal{X}_\ell^* = \mathcal{X}_\ell$ where*

$$\mathcal{X}_\ell := \{x \in \mathcal{X} : (\beta_{1i})^\top x \geq (\beta_{1i'})^\top x, \forall i \in \{\ell_1, \ldots, \ell_K\}, i' \in \{\ell_{K+1}, \ldots, \ell_{k_*}\}\}.$$

Proof of Lemma 1 is in Appendix A.3. This lemma indicates that for almost surely $X$, $\mathrm{TopK}((\beta_{1i}^*)^\top X, K; \beta_{0i}^*) = (\beta_{1i}^*)^\top X + \beta_{0i}^*$ is equivalent to $\mathrm{TopK}((\widehat{\beta}_{1i}^n)^\top X, K; \widehat{\beta}_{0i}^n) = (\widehat{\beta}_{1i}^n)^\top X + \widehat{\beta}_{0i}^n$, for any $i \in [k_*]$ and sufficiently large $n$. Based on this property, we provide in Theorem 1 the rate for estimating the true conditional density function $g_{G_*}$:

**Theorem 1** (Density estimation rate). *Given the MLE $\widehat{G}_n$ defined in equation (2), the convergence rate of the conditional density estimation $g_{\widehat{G}_n}$ to the true conditional density $g_{G_*}$ under the exact-specified settings is illustrated by the following inequality:*

$$\mathbb{P}\Big(\mathbb{E}_X[h(g_{\widehat{G}_n}(\cdot|X), g_{G_*}(\cdot|X))] > C\sqrt{\log(n)/n}\Big) \lesssim n^{-c},$$

*where $C > 0$ and $c > 0$ are some universal constants that depend on $\Omega$ and $K$.*

Proof of Theorem 1 can be found in Appendix A.1. It follows from the result of Theorem 1 that the conditional density estimation $g_{\widehat{G}_n}$ converges to its true counterpart $g_{G_*}$ under the Hellinger distance at the parametric rate of order $\widetilde{\mathcal{O}}(n^{-1/2})$. This rate plays a critical role in establishing the convergence rates of parameter estimation. In particular, if we are able to derive the following lower bound: $\mathbb{E}_X[h(g_G(\cdot|X), g_{G_*}(\cdot|X))] \gtrsim \mathcal{D}_1(G, G_*)$ for any mixing measure $G \in \mathcal{E}_{k_*}(\Omega)$, where the metric $\mathcal{D}_1$ will be defined in equation (5), we will achieve our desired parameter estimation rates. Before going into further details, let us introduce the formulation of Voronoi metric $\mathcal{D}_1$ that we use for our convergence analysis under the exact-specified settings.

**Voronoi metric.** Given an arbitrary mixing measure $G$ with $k'$ components, we distribute those components to the following Voronoi cells generated by the components of $G_*$ Manole & Ho (2022):

$$\mathcal{C}_j \equiv \mathcal{C}_j(G) := \{i \in [k'] : \|\theta_i - \theta_j^*\| \leq \|\theta_i - \theta_{j'}^*\|, \forall j' \neq j\}, \tag{4}$$

where $\theta_i := (\beta_{1i}, a_i, b_i, \sigma_i)$ and $\theta_j^* := (\beta_{1j}^*, a_j^*, b_j^*, \sigma_j^*)$ for any $j \in [k_*]$. Recall that under the exact-specified settings, the MLE $\widehat{G}_n$ belongs to the set $\mathcal{E}_{k_*}(\Omega)$. Therefore, we consider $k' = k_*$ in this case. Then, the Voronoi metric $\mathcal{D}_1$ is defined as follows:

$$\mathcal{D}_1(G, G_*) := \max_{\{\ell_j\}_{j=1}^K \subset [k_*]} \sum_{j=1}^K \left[ \sum_{i \in \mathcal{C}_{\ell_j}} \exp(\beta_{0i}) \Big( \|\Delta\beta_{1i\ell_j}\| + \|\Delta a_{i\ell_j}\| + \|\Delta b_{i\ell_j}\| + \|\Delta\sigma_{i\ell_j}\| \Big) \right.$$

$$\left. + \Big| \sum_{i \in \mathcal{C}_{\ell_j}} \exp(\beta_{0i}) - \exp(\beta_{0\ell_j}^*) \Big| \right]. \tag{5}$$

Here, we denote $\Delta\beta_{1i\ell_i} := \beta_{1i} - \beta_{1\ell_j}^*$, $\Delta a_{i\ell_j} := a_i - a_{\ell_j}^*$, $\Delta b_{i\ell_j} := b_i - b_{\ell_j}^*$ and $\Delta\sigma_{i\ell_j} := \sigma_i - \sigma_{\ell_j}^*$. Additionally, the above maximum operator helps capture the behaviors of all input regions separated by the top-K sparse softmax gating function. Now, we are ready to characterize the convergence rates of parameter estimation in the top-K sparse softmax gating Gaussian MoE.

**Theorem 2** (Parameter estimation rate). *Under the exact-specified settings, the Hellinger lower bound $\mathbb{E}_X[h(g_G(\cdot|X), g_{G_*}(\cdot|X))] \gtrsim \mathcal{D}_1(G, G_*)$ holds for any mixing measure $G \in \mathcal{E}_{k_*}(\Omega)$. Consequently, we can find a universal constant $C_1 > 0$ depending only on $G_*$, $\Omega$ and $K$ such that*

$$\mathbb{P}\Big(\mathcal{D}_1(\widehat{G}_n, G_*) > C_1\sqrt{\log(n)/n}\Big) \lesssim n^{-c_1},$$

*where $c_1 > 0$ is a universal constant that depends only on $\Omega$.*

Proof of Theorem 2 is in Appendix A.2. This theorem reveals that the Voronoi metric $\mathcal{D}_1$ between the MLE $\widehat{G}_n$ and the true mixing measure $G_*$, i.e. $\mathcal{D}_1(\widehat{G}_n, G_*)$, vanishes at the parametric rate of order $\widetilde{\mathcal{O}}(n^{-1/2})$. As a result, the rates for estimating ground-truth parameters $\exp(\beta_{0i}^*), \beta_{1i}^*, a_i^*, b_i^*, \sigma_i^*$ are optimal at $\widetilde{\mathcal{O}}(n^{-1/2})$ for any $i \in [k_*]$.

## 3 OVER-SPECIFIED SETTINGS

In this section, we continue to carry out the same convergence analysis for the top-K sparse softmax gating Gaussian MoE as in Section 2 but under the over-specified settings, that is, when the true number of experts $k_*$ becomes unknown.

Recall that under the over-specified settings, we look for the MLE $\widehat{G}_n$ within $\mathcal{O}_k(\Omega)$, i.e. the set of all mixing measures with at most $k$ components, where $k > k_*$. Thus, there could be some components $(\beta_{1i}^*, a_i^*, b_i^*, \sigma_i^*)$ of the true mixing measure $G_*$ approximated by at least two fitted components $(\widehat{\beta}_{1i}^n, \widehat{a}_i^n, \widehat{b}_i^n, \widehat{\sigma}_i^n)$ of the MLE $\widehat{G}_n$. Moreover, since the true density $g_{G_*}(Y|X)$ is associated with top $K$ experts and each of which corresponds to one component of $G_*$, we need to select more than $K$ experts in the formulation of density estimation $g_{\widehat{G}_n}(Y|X)$ to guarantee its convergence to $g_{G_*}(Y|X)$. In particular, for any mixing measure $G = \sum_{i=1}^{k'}\exp(\beta_{0i})\delta_{(\beta_{1i}, a_i, b_i, \sigma_i)} \in \mathcal{O}_k(\Omega)$, let us consider a new formulation of conditional density used for estimating the true density under the over-specified settings as follows:

$$\overline{g}_G(Y|X) := \sum_{i=1}^{k'}\mathrm{Softmax}(\mathrm{TopK}((\beta_{1i})^\top X, \overline{K}; \beta_{0i})) \cdot f(Y|(a_i)^\top X + b_i, \sigma_i).$$

Here, $\overline{g}_G(Y|X)$ is equipped with top-$\overline{K}$ sparse softmax gating, where $1 \le \overline{K} \le k'$ is fixed and might be different from $K$. Additionally, the definition of MLE in equation (2) also changes accordingly to this new density function. Then, we demonstrate in Proposition 1 an interesting phenomenon that the density estimation $\overline{g}_{\widehat{G}_n}$ converges to $g_{G_*}$ under the Hellinger distance only if $\overline{K} \ge \max_{\{\ell_1,\ldots,\ell_K\}\subset[k_*]}\sum_{j=1}^K|\mathcal{C}_{\ell_j}|$, where $\mathcal{C}_{\ell_j}$ is a Voronoi cell defined in equation (4).

**Proposition 1.** *If $\overline{K} < \max_{\{\ell_1,\ldots,\ell_K\}\subset[k_*]}\sum_{j=1}^K|\mathcal{C}_{\ell_j}|$, then the following inequality holds true:*

$$\inf_{G\in\mathcal{O}_k(\Omega)}\mathbb{E}_X[h(\overline{g}_G(\cdot|X), g_{G_*}(\cdot|X))] > 0.$$

Proof of Proposition 1 is deferred to Appendix B.3. Following from the result of Proposition 1, we will consider only the regime when $\max_{\{\ell_j\}_{j=1}^K\subset[k_*]}\sum_{j=1}^K|\mathcal{C}_{\ell_j}| \le \overline{K} \le k$ in the rest of this section to ensure the convergence of density estimation. As the number of experts chosen in the density estimation changes from $K$ to $\overline{K}$, it is necessary to partition the input space $\mathcal{X}$ into $\overline{q} := \binom{k}{\overline{K}}$ regions. More specifically, for any $\overline{\ell} \in [\overline{q}]$, we denote $\{\overline{\ell}_1, \overline{\ell}_2, \ldots, \overline{\ell}_{\overline{K}}\}$ as an $\overline{K}$-element subset of $[k]$ and $\{\overline{\ell}_{\overline{K}+1}, \ldots, \overline{\ell}_k\} := [k] \setminus \{\overline{\ell}_1, \overline{\ell}_2, \ldots, \overline{\ell}_{\overline{K}}\}$. Then, we define the $\overline{\ell}$-th region of $\mathcal{X}$ as follows:

$$\overline{\mathcal{X}}_{\overline{\ell}} := \{x \in \mathcal{X} : (\beta_{1i})^\top x \ge (\beta_{1i'})^\top x, \forall i \in \{\overline{\ell}_1, \ldots, \overline{\ell}_{\overline{K}}\}, i' \in \{\overline{\ell}_{\overline{K}+1}, \ldots, \overline{\ell}_k\}\}.$$

Inspired by the result of Lemma 1, we continue to present in Lemma 2 a relation between the values of the TopK functions in the density estimation $\overline{g}_{\widehat{G}_n}$ and the true density $g_{G_*}$.

**Lemma 2.** *For any $j \in [k_*]$ and $i \in \mathcal{C}_j$, let $\beta_{1i}, \beta_{1j}^* \in \mathbb{R}^d$ that satisfy $\|\beta_{1i} - \beta_{1j}^*\| \leq \eta_j$ for some sufficiently small $\eta_j > 0$. Additionally, for $\max_{\{\ell_j\}_{j=1}^K \subset [k_*]} \sum_{j=1}^K |\mathcal{C}_{\ell_j}| \leq \overline{K} \leq k$, we assume that there exist $\ell \in [q]$ and $\overline{\ell} \in [\overline{q}]$ such that $\{\overline{\ell}_1, \ldots, \overline{\ell}_{\overline{K}}\} = \mathcal{C}_{\ell_1} \cup \ldots \cup \mathcal{C}_{\ell_K}$. Then, if the set $\mathcal{X}_\ell^*$ does not have measure zero, we achieve that $\mathcal{X}_\ell^* = \overline{\mathcal{X}}_{\overline{\ell}}$.*

Proof of Lemma 2 is in Appendix B.4. Different from Lemma 1, we need to impose on Lemma 2 an assumption that there exist indices $\ell \in [q]$ and $\overline{\ell} \in [\overline{q}]$ that satisfy $\{\overline{\ell}_1, \ldots, \overline{\ell}_{\overline{K}}\} = \mathcal{C}_{\ell_1} \cup \ldots \cup \mathcal{C}_{\ell_K}$. This assumption means that each component $(\widehat{\beta}_{1i}^n, \widehat{a}_i^n, \widehat{b}_i^n, \widehat{\sigma}_i^n)$ corresponding to the top $\overline{K}$ experts in $g_{\widehat{G}_n}$ must converge to some true component which corresponds to the top $K$ experts in $g_{G_*}$. Consequently, for $X \in \mathcal{X}_\ell^*$, if $\text{TopK}((\beta_{1j}^*)^\top X, K; \beta_{0j}^*) = (\beta_{1j}^*)^\top X + \beta_{0j}^*$ holds true, then we achieve that $\text{TopK}((\widehat{\beta}_{1i}^n)^\top X, \overline{K}; \widehat{\beta}_{0i}^n) = (\widehat{\beta}_{1i}^n)^\top X + \widehat{\beta}_{0i}^n$ and vice versa, for any $j \in [k_*]$ and $i \in \mathcal{C}_j$. Given the result of Lemma 2, we are now able to derive the convergence rate of the density estimation $\overline{g}_{\widehat{G}_n}$ to its true counterpart $g_{G_*}$ under the over-specified settings in Theorem 3.

**Theorem 3** (Density estimation rate). *Under the over-specified settings, the conditional density estimation $\overline{g}_{\widehat{G}_n}$ converges to the true density $g_{G_*}$ under the Hellinger distance at the following rate:*

$$\mathbb{P}\Big(\mathbb{E}_X[h(\overline{g}_{\widehat{G}_n}(\cdot|X), g_{G_*}(\cdot|X))] > C'\sqrt{\log(n)/n}\Big) \lesssim n^{-c'},$$

*where $C' > 0$ and $c' > 0$ are some universal constants that depend on $\Omega$ and $K$.*

Proof of Theorem 3 is in Appendix B.1. Although there is a modification in the number of experts chosen from $\overline{g}_{\widehat{G}_n}$, Theorem 3 verifies that the convergence rate of this density estimation to $g_{G_*}$ under the over-specified settings remains the same as that under the exact-specified settings, which is of order $\widetilde{\mathcal{O}}(n^{-1/2})$. Subsequently, we will leverage this result for our convergence analysis of parameter estimation under the over-specified settings, which requires us to design a new Voronoi metric.

**Voronoi metric.** Regarding the top-K sparse softmax gating function, challenges comes not only from the $\text{TopK}$ function but also from the $\text{Softmax}$ function. In particular, there is an intrinsic interaction between the numerators of softmax weights and the expert functions in the Gaussian density. Moreover, Gaussian density parameters also interacts with each other. These two interactions are respectively illustrated by the following partial differential equations (PDEs):

$$\frac{\partial^2 s(X,Y)}{\partial \beta_1 \partial b} = \frac{\partial s(X,Y)}{\partial a}; \qquad \frac{\partial s(X,Y)}{\partial \sigma} = \frac{1}{2} \cdot \frac{\partial^2 s(X,Y)}{\partial b^2}, \tag{6}$$

where we denote $s(X,Y) := \exp(\beta_1^\top X) \cdot f(Y|a^\top X + b, \sigma)$. These PDEs arise when we decompose the density difference $g_{\widehat{G}_n}(Y|X) - g_{G_*}(Y|X)$ into a linear combination of linearly independent terms using Taylor expansions. However, the above PDEs indicates that derivative terms which admit the forms in equation (6) are linearly dependent. Therefore, we have to group these terms by taking the summation of their coefficients, and then arrive at our desired linear combination of linearly independent elements. Consequently, when $g_{\widehat{G}_n}(Y|X) - g_{G_*}(Y|X)$ converges to zero, all the coefficients in this combination also tend to zero, which leads to the following system of polynomial equations with unknown variables $\{z_{1j}, z_{2j}, z_{3j}, z_{4j}, z_{5j}\}_{i=1}^m$:

$$\sum_{i=1}^m \sum_{(\alpha_1, \alpha_2, \alpha_3, \alpha_4) \in \mathcal{J}_{\eta_1, \eta_2}} \frac{z_{5i}^2 \, z_{1i}^{\alpha_1} \, z_{2i}^{\alpha_2} \, z_{3i}^{\alpha_3} \, z_{4i}^{\alpha_4}}{\alpha_1! \, \alpha_2! \, \alpha_3! \, \alpha_4!} = 0, \tag{7}$$

for all $(\eta_1, \eta_2) \in \mathbb{N}^d \times \mathbb{N}$ such that $0 \leq |\eta_1| \leq r$, $0 \leq \eta_2 \leq r - |\eta_1|$ and $|\eta_1| + \eta_2 \geq 1$, where $\mathcal{J}_{\eta_1, \eta_2} := \{(\alpha_1, \alpha_2, \alpha_3, \alpha_4) \in \mathbb{N}^d \times \mathbb{N}^d \times \mathbb{N} \times \mathbb{N} : \alpha_1 + \alpha_2 = \eta_1, |\alpha_2| + \alpha_3 + \alpha_4 = \eta_2\}$.

Here, a solution to the above system is called non-trivial if all of variables $z_{5i}$ are different from zero, whereas at least one among variables $z_{3i}$ is non-zero. For any $m \geq 2$, let $\overline{r}(m)$ be the smallest natural number $r$ such that the above system does not have any non-trivial solution. In a general case when $d \geq 1$ and $m \geq 2$, it is non-trivial to determine the exact value of $\overline{r}(m)$ Sturmfels (2002). However, when $m$ is small, Nguyen et al. (2023) show that $\overline{r}(2) = 4$ and $\overline{r}(3) = 6$. Additionally, since $\overline{r}(m)$ is a monotonically increasing function of $m$, they also conjecture that $\overline{r}(m) = 2m$ for any $m \geq 2$ and $d \geq 1$.

Given the above results, we design a Voronoi metric $\mathcal{D}_2$ to capture the convergence rates of parameter estimation in the top-K sparse softmax gating Gaussian MoE under the over-specified settings as

$$
\mathcal{D}_2(G, G_*) := \max_{\{\ell_j\}_{j=1}^K \subset [k_*]} \left\{ \sum_{\substack{j \in [K], \; i \in \mathcal{C}_{\ell_j} \\ |\mathcal{C}_{\ell_j}|=1}} \exp(\beta_{0i}) \Big[ \|\Delta\beta_{1i\ell_j}\| + \|\Delta a_{i\ell_j}\| + |\Delta b_{i\ell_j}| + |\Delta\sigma_{i\ell_j}| \Big] \right.
$$

$$
+ \sum_{\substack{j \in [K], \; i \in \mathcal{C}_{\ell_j} \\ |\mathcal{C}_{\ell_j}|>1}} \exp(\beta_{0i}) \Big[ \|\Delta\beta_{1i\ell_j}\|^{\bar{r}(|\mathcal{C}_{\ell_j}|)} + \|\Delta a_{i\ell_j}\|^{\frac{\bar{r}(|\mathcal{C}_{\ell_j}|)}{2}} + |\Delta b_{i\ell_j}|^{\bar{r}(|\mathcal{C}_{\ell_j}|)} + |\Delta\sigma_{i\ell_j}|^{\frac{\bar{r}(|\mathcal{C}_{\ell_j}|)}{2}} \Big]
$$

$$
\left. + \sum_{j=1}^K \Big| \sum_{i \in \mathcal{C}_{\ell_j}} \exp(\beta_{0i}) - \exp(\beta_{0\ell_j}^*) \Big| \right\}, \quad (8)
$$

for any mixing measure $G \in \mathcal{O}_k(\Omega)$. The above maximum operator allows us to capture the behaviors of all input regions partitioned by the top-K sparse softmax gating function in $g_{G_*}$. Then, we show in the following theorem that parameter estimation rates vary with the values of the function $\bar{r}(\cdot)$

**Theorem 4** (Parameter estimation rate). *Under the over-specified settings, the Hellinger lower bound $\mathbb{E}_X[h(\bar{g}_G(\cdot|X), g_{G_*}(\cdot|X))] \gtrsim \mathcal{D}_2(G, G_*)$ holds for any mixing measure $G \in \mathcal{O}_k(\Omega)$. As a consequence, we can find a universal constant $C_2 > 0$ depending only on $G_*$, $\Omega$ and $K$ such that*

$$
\mathbb{P}\Big( \mathcal{D}_2(\widehat{G}_n, G_*) > C_2\sqrt{\log(n)/n} \Big) \lesssim n^{-c_2},
$$

*where $c_2 > 0$ is a universal constant that depends only on $\Omega$.*

Proof of Theorem 4 is in Appendix B.2. A few remarks on this theorem are in order.

**(i)** Under the over-specified settings, the MLE $\widehat{G}_n$ converges to the true mixing measure $G_*$ at the rate of order $\widetilde{\mathcal{O}}(n^{-1/2})$ under the Voronoi metric $\mathcal{D}_2$. Let us denote $\mathcal{C}_j^n = \mathcal{C}_j(\widehat{G}_n)$, then this result indicates that the estimation rates for ground-truth parameters $\exp(\beta_{0j}^*), \beta_{1j}^*, a_j^*, b_j^*, \sigma_j^*$ fitted by only one component, i.e. $|\mathcal{C}_j^n| = 1$, remain the same at $\widetilde{\mathcal{O}}(n^{-1/2})$ as those in the exact-specified settings.

**(ii)** However, for ground-truth parameters which are approximated by at least two components, i.e. $|\mathcal{C}_j^n| > 1$, the rates for estimating them depend on their corresponding Voronoi cells, and become significantly low when the cardinality of those Voronoi cells increase. In particular, both $\beta_{1j}^*$ and $b_j^*$ admit the estimation rate of order $\widetilde{\mathcal{O}}(n^{-1/2\bar{r}(|\mathcal{C}_j^n|)})$. Meanwhile, the convergence rates of estimating $a_j^*$ and $\sigma_j^*$ are of the same order $\mathcal{O}(n^{-1/\bar{r}(|\mathcal{C}_j^n|)})$. For instance, assume that $(\beta_{1j}^*, a_j^*, b_j^*, \sigma_j^*)$ has three fitted components, which means that $|\mathcal{C}_j^n| = 3$ and therefore, $\bar{r}(|\mathcal{C}_j^n|) = 6$. Then, the estimation rates for $\beta_{1j}^*, b_j^*$ and $a_j^*, \sigma_j^*$ are $\widetilde{\mathcal{O}}(n^{-1/12})$ and $\widetilde{\mathcal{O}}(n^{-1/6})$, respectively.

## 4 PRACTICAL IMPLICATIONS

In this section, we present three main practical implications of our theoretical results for the use of top-K sparse softmax gating function in mixture of experts as follows:

**1. No trade-off between model capacity and model performance:** In the top-K sparse softmax gating Gaussian mixture of experts, since the gating function is discontinuous and only a portion of experts are activated for each input to scale up the model capacity, the parameter estimation rates under that model are expected to be slower than those under the dense softmax gating Gaussian mixture of experts (Nguyen et al., 2023). However, from our theories it turns out that the parameter estimation rates under these two models are the same under both the exact-specified and over-specified settings. As a result, we point out that using the top-K sparse softmax gating function allows us to scale up the model capacity without sacrificing the computational cost as well as the convergence rates of parameter and density estimation.

**2. Favourable gating function:** As mentioned in Section 3, due to an intrinsic interaction between gating parameters and expert parameters via the first PDE in equation (6), the rates for estimating those parameters under the over-specified settings are determined by the solvability of the system of

polynomial equations (7), which are significantly slow. However, if we use the top-1 sparse softmax gating function, i.e. activating only a single expert for each input, then the gating value is either one or zero. As a result, the previous interaction no longer occurs, which helps improve the parameter estimation rates. This partially accounts for the efficacy of top-1 sparse softmax gating mixture of experts in scaling up deep learning architectures (see (Fedus et al., 2022b)).

**3. True/ Minimal number of experts:** Another challenge is to choose the true/ minimal number of experts, which can be partially addressed using theories developed from the paper. In particular, suppose that the MLE $\widehat{G}_n$ consists of $\hat{k}_n$ components. When the sample size $n$ goes to infinity, every Voronoi cell among $\mathcal{C}_1^n, \ldots, \mathcal{C}_{k_*}^n$ contains at least one element. Since the total number of elements of those Voronoi cells is $\hat{k}_n$, the maximum cardinality of a Voronoi cell turns out to be $\hat{k}_n - k_* + 1$. This maximum value is attained when there is exactly one ground-truth component $(\beta_{1j}^*, a_j^*, b_j^*, \sigma_j^*)$ fitted by more than one component. An instance for this scenario is when $|\mathcal{C}_1^n| = \hat{k}_n - k_* + 1$ and $|\mathcal{C}_2^n| = \ldots = |\mathcal{C}_{k_*}^n| = 1$. Under this setting, the first true parameters $\beta_{11}^*, b_1^*$ and $a_1^*, \sigma_1^*$ achieve their worst possible estimation rates of order $\widetilde{\mathcal{O}}(n^{-1/2\overline{r}(\hat{k}_n - k_* + 1)})$ and $\widetilde{\mathcal{O}}(n^{-1/\overline{r}(\hat{k}_n - k_* + 1)})$, respectively, which become significantly slow when the difference $\hat{k}_n - k_*$ increases. As a consequence, the estimated number of experts $\hat{k}_n$ should not be very large compared to the true number of experts $k_*$.

## 5 CONCLUSION AND FUTURE DIRECTIONS

In this paper, we provide a convergence analysis for density estimation and parameter estimation in the top-K sparse softmax gating Gaussian MoE. To overcome the complex structure of top-K sparse softmax gating function, we first establish a connection between the outputs of $\mathrm{TopK}$ function associated with the density estimation and the true density in each input region partitioned by the gating function, and then construct novel Voronoi loss functions among parameters to capture different behaviors of these input regions. Under the exact-specified settings, we show that the rates for estimating the true density and true parameters are both parametric on the sample size. On the other hand, although the density estimation rate remains parametric under the over-specified settings, the parameter estimation rates witness a sharp drop because of an interaction between the softmax gating and expert functions.

Based on the results of this paper, there are a few potential future directions. Firstly, as we mentioned in Section 4, a question of how to estimate the true number of experts $k_*$ and the number of experts selected in the top-K sparse softmax gating function $K$ naturally arises from this work. Since the parameter estimation rates under the over-specified settings decrease proportionately to the number of fitted experts $k$, one possible approach to estimating $k_*$ is to reduce $k$ until these rates reach the optimal order $\widetilde{\mathcal{O}}(n^{-1/2})$. That reduction can be done by merging close estimated parameters within their convergence rates to the true parameters (Guha et al., 2021) or by penalizing the log-likelihood function of the top-K sparse softmax gating Gaussian MoE using the differences among the parameters (Manole & Khalili, 2021). As a result, the number of experts chosen in the density estimation $\overline{K}$ also decreases accordingly and approximates the value of $K$. Secondly, the theoretical results established in the paper are under the assumption that the data are assumed to be generated from a top-K sparse softmax gating Gaussian MoE, which can be violated in real-world settings when the data are not necessarily generated from that model. Under those misspecified settings, the MLE converges to a mixing measure $\overline{G} \in \arg\min_{G \in \mathcal{O}_k(\Omega)} \mathrm{KL}(P(Y|X) \| g_G(Y|X))$ where $P(Y|X)$ is the true conditional distribution of $Y$ given $X$ and KL stands for the Kullback-Leibler divergence. As the space $\mathcal{O}_k(\Omega)$ is non-convex, the existence of $\overline{G}$ is not unique. Furthermore, the current analysis of the MLE under the misspecified settings of statistical models is mostly conducted when the function space is convex (van de Geer, 2000), which is inapplicable to the current non-convex misspecified settings. Thus, it is necessary to develop a new analysis and a new metric to establish the convergence rate of the MLE to the set of $\overline{G}$. Finally, since the log-likelihood function of the top-K sparse softmax gating Gaussian MoE is highly non-concave, the EM algorithm is used to approximate the MLE. While the statistical guarantee of the EM has been established for vanilla Gaussian mixture models (Balakrishnan et al., 2017; Dwivedi et al., 2020b;a), to the best of our knowledge such guarantee has not been studied in the setting of top-K sparse softmax gating Gaussian MoE. A potential approach to this problem is to utilize the population-to-sample analysis that has been widely used in previous works to study the EM algorithm Balakrishnan et al. (2017); Ho et al. (2023). We leave that direction for the future work.

ACKNOWLEDGEMENTS

NH acknowledges support from the NSF IFML 2019844 and the NSF AI Institute for Foundations of Machine Learning.

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

# Supplement to "Statistical Perspective of Top-K Sparse Softmax Gating Mixture of Experts"

In this supplementary material, we first provide rigorous proofs for all results under the exact-specified settings in Appendix A, while those for the over-specified settings are then presented in Appendix B. Next, we study the identifiability of the top-K sparse softmax gating Gaussian mixture of experts (MoE) in Appendix C. We then carry out several numerical experiments in Appendix D to empirically justify our theoretical results. Finally, we establish the theories for parameter and density estimation beyond the settings of top-K sparse softmax Gaussian MoE in Appendix E.

## A   PROOF FOR RESULTS UNDER THE EXACT-SPECIFIED SETTINGS

In this appendix, we present the proofs for Theorem 1 in Appendix A.1, while that for Theorem 2 is then given in Appendix A.2. Lastly, the proof of Lemma 1 is provided in Appendix A.3.

### A.1   PROOF OF THEOREM 1

In this appendix, we conduct a convergence analysis for density estimation in the top-K sparse softmax gating Gaussian MoE using proof techniques in (van de Geer, 2000). For that purpose, it is necessary to introduce some essential notations and key results first.

Let $\mathcal{P}_{k_*}(\Theta) := \{g_G(Y|X) : G \in \mathcal{E}_{k_*}(\Omega)\}$ be the set of all conditional density functions of mixing measures in $\mathcal{E}_{k_*}(\Omega)$. Next, we denote by $N(\varepsilon, \mathcal{P}_{k_*}(\Omega), \|\cdot\|_1)$ the covering number of metric space $(\mathcal{P}_{k_*}(\Omega), \|\cdot\|_1)$. Meanwhile, $H_B(\varepsilon, \mathcal{P}_{k_*}(\Omega), h)$ represents for the bracketing entropy of $\mathcal{P}_{k_*}(\Omega)$ under the Hellinger distance. Then, we provide in the following lemma the upper bounds of those terms.

**Lemma 3.** *If $\Omega$ is a bounded set, then the following inequalities hold for any $0 < \eta < 1/2$:*

   *(i)* $\log N(\eta, \mathcal{P}_{k_*}(\Omega), \|\cdot\|_1) \lesssim \log(1/\eta)$;

   *(ii)* $H_B(\eta, \mathcal{P}_{k_*}(\Omega), h) \lesssim \log(1/\eta)$.

Proof of Lemma 3 is in Appendix A.1.2. Subsequently, we denote

$$\widetilde{\mathcal{P}}_{k_*}(\Omega) := \{g_{(G+G_*)/2}(Y|X) : G \in \mathcal{E}_{k_*}(\Omega)\};$$
$$\widetilde{\mathcal{P}}_{k_*}^{1/2}(\Omega) := \{g_{(G+G_*)/2}^{1/2}(Y|X) : G \in \mathcal{E}_{k_*}(\Omega)\}.$$

In addition, for each $\delta > 0$, we define a Hellinger ball centered around the conditional density function $g_{G_*}(Y|X)$ and met with the set $\widetilde{\mathcal{P}}_{k_*}^{1/2}(\Omega)$ as

$$\widetilde{\mathcal{P}}_{k_*}^{1/2}(\Omega, \delta) := \{g^{1/2} \in \widetilde{\mathcal{P}}_{k_*}^{1/2}(\Omega) : h(g, g_{G_*}) \le \delta\}.$$

To capture the size of the above Hellinger ball, van de Geer (2000) suggest using the following quantity:

$$\mathcal{J}_B(\delta, \widetilde{\mathcal{P}}_{k_*}^{1/2}(\Omega, \delta)) := \int_{\delta^2/2^{13}}^{\delta} H_B^{1/2}(t, \widetilde{\mathcal{P}}_{k_*}^{1/2}(\Omega, t), \|\cdot\|) \mathrm{d}t \vee \delta, \tag{9}$$

where $t \vee \delta := \max\{t, \delta\}$. Given those notations, let us recall a standard result for density estimation in van de Geer (2000).

**Lemma 4** (Theorem 7.4, van de Geer (2000)). *Take $\Psi(\delta) \ge \mathcal{J}_B(\delta, \widetilde{\mathcal{P}}_{k_*}^{1/2}(\Omega, \delta))$ such that $\Psi(\delta)/\delta^2$ is a non-increasing function of $\delta$. Then, for some sequence $(\delta_n)$ and universal constant $c$ which satisfy $\sqrt{n}\delta_n^2 \ge c\Psi(\delta)$, we obtain that*

$$\mathbb{P}\left(\mathbb{E}_X\left[h(g_{\widehat{G}_n}(\cdot|X), g_{G_*}(\cdot|X))\right] > \delta\right) \le c\exp(-n\delta^2/c^2),$$

*for any $\delta \ge \delta_n$*

Proof of Lemma 4 can be found in van de Geer (2000). Now, we are ready to provide the proof for convergence rate of density estimation in Theorem 1 in Appendix A.1.1.

### A.1.1 MAIN PROOF

It is worth noting that for any $t > 0$, we have

$$H_B(t, \widetilde{\mathcal{P}}_{k_*}^{1/2}(\Omega, t), \| \cdot \|) \le H_B(t, \mathcal{P}_{k_*}(\Omega, t), h).$$

Then, the integral in equation (9) is upper bounded as follows:

$$\mathcal{J}_B(\delta, \widetilde{\mathcal{P}}_{k_*}^{1/2}(\Omega, \delta)) \le \int_{\delta^2/2^{13}}^{\delta} H_B^{1/2}(t, \mathcal{P}_{k_*}(\Omega, t), h) \mathrm{d}t \vee \delta \lesssim \int_{\delta^2/2^{13}}^{\delta} \log(1/t) \mathrm{d}t \vee \delta, \qquad (10)$$

where the second inequality follows from part (ii) of Lemma 3.

As a result, by choosing $\Psi(\delta) = \delta \cdot \sqrt{\log(1/\delta)}$, we can verify that $\Psi(\delta)/\delta^2$ is a non-increasing function of $\delta$. Furthermore, the inequality in equation (10) indicates that $\Psi(\delta) \ge \mathcal{J}_B(\delta, \widetilde{\mathcal{P}}_{k_*}^{1/2}(\Omega, \delta))$. Next, let us consider a sequence $(\delta_n)$ defined as $\delta_n := \sqrt{\log(n)/n}$. This sequence can be validated to satisfy the condition $\sqrt{n}\delta_n^2 \ge c\Psi(\delta)$ for some universal constant $c$. Therefore, by Lemma 4, we reach the conclusion of Theorem 1:

$$\mathbb{P}\Big(\mathbb{E}_X[h(g_{\widehat{G}_n}(\cdot|X), g_{G_*}(\cdot|X))] > C\sqrt{\log(n)/n}\Big) \lesssim n^{-c},$$

for some universal constant $C$ depending only on $\Omega$.

### A.1.2 PROOF OF LEMMA 3

**Part (i).** In this part, we will derive the following upper bound for the covering number of metric space $(\mathcal{P}_{k_*}(\Omega), \| \cdot \|_1)$ for any $0 < \eta < 1/2$ given the bounded set $\Omega$:

$$\log N(\eta, \mathcal{P}_{k_*}(\Omega), \| \cdot \|_1) \lesssim \log(1/\eta).$$

To begin with, we define $\Theta := \{(a, b, \sigma) \in \mathbb{R}^d \times \mathbb{R} \times \mathbb{R}_+ : (\beta_0, \beta_1, a, b, \sigma) \in \Omega\}$. Note that $\Omega$ is a bounded set, then $\Theta$ also admits this property. Thus, there exists an $\eta$-cover of $\Theta$, denoted by $\overline{\Theta}_\eta$. Additionally, we also define $\Delta := \{(\beta_0, \beta_1) \in \mathbb{R} \times \mathbb{R}^d : (\beta_0, \beta_1, a, b, \sigma) \in \Omega\}$, and $\overline{\Delta}_\eta$ be an $\eta$-cover of $\Delta$. Then, it can be validated that $|\overline{\Theta}_\eta| \le \mathcal{O}(\eta^{-(d+1)k_*})$ and $|\overline{\Delta}_\eta| \le \mathcal{O}(\eta^{-(d+3)k_*})$.

Subsequently, for each $G = \sum_{i=1}^{k_*} \exp(\beta_{0i})\delta_{(\beta_{1i}, a_i, b_i, \sigma_i)} \in \mathcal{E}_{k_*}(\Omega)$, we take into account two other mixing measures. The first measure is $G' = \sum_{i=1}^{k_*} \exp(\beta_{0i})\delta_{(\beta_{1i}, \overline{a}_i, \overline{b}_i, \overline{\sigma}_i)}$, where $(\overline{a}_i, \overline{b}_i, \overline{\sigma}_i) \in \overline{\Theta}_\eta$ is the closest points to $(a_i, b_i, \sigma_i)$ in this set for all $i \in [k_*]$. The second one is $\overline{G} := \sum_{i=1}^{k_*} \exp(\overline{\beta}_{0i})\delta_{(\overline{\beta}_{1i}, \overline{a}_i, \overline{b}_i, \overline{\sigma}_i)}$ in which $(\overline{\beta}_{0i}, \overline{\beta}_{1i}) \in \overline{\Delta}_\eta$ for any $i \in [k_*]$. Next, let us define

$$\mathcal{T} := \{g_{\overline{G}} \in \mathcal{P}_{k_*}(\Omega) : (\overline{\beta}_{0i}, \overline{\beta}_{1i}) \in \overline{\Delta}_\eta, (\overline{a}_i, \overline{b}_i, \overline{\sigma}_i) \in \overline{\Theta}_\eta, \forall i \in [k_*]\},$$

then it is obvious that $g_{\overline{G}} \in \mathcal{T}$. Now, we will show that $\mathcal{T}$ is an $\eta$-cover of metric space $(\mathcal{P}_{k_*}(\Omega), \| \cdot \|_1)$ with a note that it is not necessarily the smallest cover. Indeed, according to the triangle inequality,

$$\|g_G - g_{\overline{G}}\|_1 \le \|g_G - g_{G'}\|_1 + \|g_{G'} - g_{\overline{G}}\|_1. \qquad (11)$$

The first term in the right hand side can be upper bounded as follows:

$$
\begin{aligned}
\|g_G - g_{G'}\|_1 &\le \sum_{i=1}^{k_*} \int_{\mathcal{X} \times \mathcal{Y}} \Big| f(Y|a_i^\top X + b_i, \sigma_i) - f(Y|\overline{a}_i^\top X + \overline{b}_i, \overline{\sigma}_i) \Big| \mathrm{d}(X, Y) \\
&\lesssim \sum_{i=1}^{k_*} \int_{\mathcal{X} \times \mathcal{Y}} \Big( \|a_i - \overline{a}_i\| + \|b_i - \overline{b}_i\| + \|\sigma_i - \overline{\sigma}_i\| \Big) \mathrm{d}(X, Y) \\
&= \sum_{i=1}^{k_*} \Big( \|a_i - \overline{a}_i\| + \|b_i - \overline{b}_i\| + \|\sigma_i - \overline{\sigma}_i\| \Big) \\
&\lesssim \eta.
\end{aligned}
\qquad (12)
$$

Next, we will also demonstrate that $\|g_{G'} - g_{\overline{G}}\|_1 \lesssim \eta$. For that purpose, let us consider $q := \binom{k}{K}$ $K$-element subsets of $\{1, \ldots, k\}$, which are assumed to take the form $\{\ell_1, \ell_2, \ldots, \ell_K\}$ for any $\ell \in [q]$.

Additionally, we also denote $\{\ell_{K+1}, \ldots, \ell_k\} := \{1, \ldots, k\} \setminus \{\ell_1, \ldots, \ell_K\}$ for any $\ell \in [q]$. Then, we define

$$\mathcal{X}_\ell := \{x \in \mathcal{X} : \beta_{1i}^\top x \geq \beta_{1i'}^\top x : i \in \{\ell_1, \ldots, \ell_K\}, i' \in \{\ell_{K+1}, \ldots, \ell_{k_*}\}\},$$

$$\widetilde{\mathcal{X}}_\ell := \{x \in \mathcal{X} : \overline{\beta}_{1i}^\top x \geq \overline{\beta}_{1i'}^\top x : i \in \{\ell_1, \ldots, \ell_K\}, i' \in \{\ell_{K+1}, \ldots, \ell_{k_*}\}\}.$$

By using the same arguments as in the proof of Lemma 1 in Appendix A.3, we achieve that either $\mathcal{X}_\ell = \widetilde{\mathcal{X}}_\ell$ or $\mathcal{X}_\ell$ has measure zero for any $\ell \in [q]$. As the Softmax function is differentiable, it is a Lipschitz function with some Lipschitz constant $L \geq 0$. Since $\mathcal{X}$ is a bounded set, we may assume that $\|X\| \leq B$ for any $X \in \mathcal{X}$. Next, we denote

$$\pi_\ell(X) := \left(\beta_{1\ell_i}^\top x + \beta_{0\ell_i}^\top\right)_{i=1}^K; \qquad \overline{\pi}_\ell(X) := \left(\overline{\beta}_{1\ell_i}^\top x + \overline{\beta}_{0\ell_i}^\top\right)_{i=1}^K,$$

for any $K$-element subset $\{\ell_1, \ldots \ell_K\}$ of $\{1, \ldots, k_*\}$. Then, we get

$$\|\mathrm{Softmax}(\pi_\ell(X)) - \mathrm{Softmax}(\overline{\pi}_\ell(X))\| \leq L \cdot \|\pi_\ell(X) - \overline{\pi}_\ell(X)\|$$

$$\leq L \cdot \sum_{i=1}^K \left(\|\beta_{1\ell_i} - \overline{\beta}_{1\ell_i}\| \cdot \|X\| + |\beta_{0\ell_i} - \overline{\beta}_{0\ell_i}|\right)$$

$$\leq L \cdot \sum_{i=1}^K \left(\eta B + \eta\right)$$

$$\lesssim \eta.$$

Back to the proof for $\|g_{G'} - g_{\overline{G}}\|_1 \lesssim \eta$, it follows from the above results that

$$\|g_{G'} - g_{\overline{G}}\|_1 = \int_{\mathcal{X} \times \mathcal{Y}} |g_{G'}(Y|X) - g_{\overline{G}}(Y|X)| \, \mathrm{d}(X, Y)$$

$$\leq \sum_{\ell=1}^q \int_{\mathcal{X}_\ell \times \mathcal{Y}} |g_{G'}(Y|X) - g_{\overline{G}}(Y|X)| \, \mathrm{d}(X, Y)$$

$$\leq \sum_{\ell=1}^q \int_{\mathcal{X}_\ell \times \mathcal{Y}} \sum_{i=1}^K \left|\mathrm{Softmax}(\pi_\ell(X)_i) - \mathrm{Softmax}(\overline{\pi}_\ell(X)_i)\right| \cdot \left|f(Y|\overline{a}_{\ell_i}^\top X + \overline{b}_{\ell_i}, \overline{\sigma}_{\ell_i})\right| \, \mathrm{d}(X, Y)$$

$$\lesssim \eta, \tag{13}$$

From the results in equations (11), (12) and (13), we deduce that $\|g_G - g_{\overline{G}}\|_1 \lesssim \eta$. This implies that $\mathcal{T}$ is an $\eta$-cover of the metric space $(\mathcal{P}_{k_*}(\Omega), \|\cdot\|_1)$. Consequently, we achieve that

$$N(\eta, \mathcal{P}_{k_*}(\Omega), \|\cdot\|_1) \lesssim |\overline{\Delta}_\eta| \times |\overline{\Theta}_\eta| \leq \mathcal{O}(1/\eta^{(d+3)k}),$$

which induces the conclusion of this part

$$\log N(\eta, \mathcal{P}_{k_*}(\Omega), \|\cdot\|_1) \lesssim \log(1/\eta).$$

**Part (ii).** Moving to this part, we will provide an upper bound for the bracketing entropy of $\mathcal{P}_{k_*}(\Omega)$ under the Hellinger distance:

$$H_B(\eta, \mathcal{P}_{k_*}(\Omega), h) \lesssim \log(1/\eta).$$

Recall that $\Theta$ and $\mathcal{X}$ are bounded sets, we can find positive constants $-\gamma \leq a^\top X + b \leq \gamma$ and $u_1 \leq \sigma \leq u_2$. Let us define

$$Q(Y|X) := \begin{cases} \frac{1}{\sqrt{2\pi u_1}} \exp\left(-\frac{Y^2}{8u_2}\right), & \text{for } |Y| \geq 2\gamma \\ \frac{1}{\sqrt{2\pi u_1}}, & \text{for } |Y| < 2\gamma \end{cases}$$

Then, it can be validated that $f(Y|a^\top X + b, \sigma) \leq Q(X, Y)$ for any $(X, Y) \in \mathcal{X} \times \mathcal{Y}$.

Next, let $\tau \leq \eta$ which will be chosen later and $\{g_1, \ldots, g_N\}$ be an $\tau$-cover of metric space $(\mathcal{P}_{k_*}(\Omega), \|\cdot\|_1)$ with the covering number $N := N(\tau, \mathcal{P}_{k_*}(\Omega), \|\cdot\|_1)$. Additionally, we also consider brackets of the form $[\Psi_i^L(Y|X), \Psi_i^U(Y|X)]$ where

$$\Psi_i^L(Y|X) := \max\{g_i(Y|X) - \tau, 0\}$$

$$\Psi_i^U(Y|X) := \max\{g_i(Y|X) + \tau, Q(Y|X)\}.$$

Then, we can check that $\mathcal{P}_{k_*}(\Omega) \subseteq \bigcup_{i=1}^{N}[\Psi_i^L(Y|X), \Psi_i^U(Y|X)]$ and $\Psi_i^U(Y|X) - \Psi_i^L(Y|X) \leq \min\{2\eta, Q(Y|X)\}$. Let $S := \max\{2\gamma, \sqrt{8u_2}\}\log(1/\tau)$, we have for any $i \in [N]$ that

$$\|\Psi_i^U - \Psi_i^L\|_1 = \int_{|Y|<2\gamma}[\Psi_i^U(Y|X) - \Psi_i^L(Y|X)]\,\mathrm{d}X\mathrm{d}Y + \int_{|Y|\geq 2\gamma}[\Psi_i^U(Y|X) - \Psi_i^L(Y|X)]\,\mathrm{d}X\mathrm{d}Y$$

$$\leq S\tau + \exp\left(-\frac{S^2}{2u_2}\right) \leq S'\tau,$$

where $S'$ is some positive constant. This inequality indicates that

$$H_B(S'\tau, \mathcal{P}_{k_*}(\Omega), \|\cdot\|_1) \leq \log N(\tau, \mathcal{P}_{k_*}(\Omega), \|\cdot\|_1) \leq \log(1/\tau).$$

By setting $\tau = \eta/S'$, we obtain that $H_B(\eta, \mathcal{P}_{k_*}(\Omega), \|\cdot\|_1) \lesssim \log(1/\eta)$. Finally, since the norm $\|\cdot\|_1$ is upper bounded by the Hellinger distance, we reach the conclusion of this part:

$$H_B(\eta, \mathcal{P}_{k_*}(\Omega), h) \lesssim \log(1/\eta).$$

Hence, the proof is completed.

## A.2    Proof of Theorem 2

Since the Hellinger distance is lower bounded by the Total Variation distance, that is $h \geq V$, we will prove the following Total Variation lower bound:

$$\mathbb{E}_X[V(g_G(\cdot|X), g_{G_*}(\cdot|X))] \gtrsim \mathcal{D}_1(G, G_*),$$

which is then respectively broken into local part and global part as follows:

$$\inf_{G \in \mathcal{E}_{k_*}(\Omega):\mathcal{D}_1(G,G_*)\leq\varepsilon'} \frac{\mathbb{E}_X[V(g_G(\cdot|X), g_{G_*}(\cdot|X))]}{\mathcal{D}_1(G, G_*)} > 0, \tag{14}$$

$$\inf_{G \in \mathcal{E}_{k_*}(\Omega):\mathcal{D}_1(G,G_*)>\varepsilon'} \frac{\mathbb{E}_X[V(g_G(\cdot|X), g_{G_*}(\cdot|X))]}{\mathcal{D}_1(G, G_*)} > 0, \tag{15}$$

for some constant $\varepsilon' > 0$.

**Proof of claim** (14): It is sufficient to show that

$$\lim_{\varepsilon\to 0} \inf_{G \in \mathcal{E}_{k_*}(\Omega):\mathcal{D}_1(G,G_*)\leq\varepsilon} \frac{\mathbb{E}_X[V(g_G(\cdot|X), g_{G_*}(\cdot|X))]}{\mathcal{D}_1(G, G_*)} > 0.$$

Assume that this inequality does not hold, then since the number of experts $k_*$ is known in this case, there exists a sequence of mixing measure $G_n := \sum_{i=1}^{k_*} \exp(\beta_{0i}^n)\delta_{(\beta_{1i}^n, a_i^n, b_i^n, \sigma_i^n)} \in \mathcal{E}_{k_*}(\Omega)$ such that both $\mathcal{D}_1(G_n, G_*)$ and $\mathbb{E}_X[V(g_{G_n}(\cdot|X), g_{G_*}(\cdot|X))]/\mathcal{D}_1(G_n, G_*)$ approach zero as $n$ tends to infinity. Now, we define

$$\mathcal{C}_j^n = \mathcal{C}_j(G_n) := \{i \in [k_*] : \|\omega_i^n - \omega_j^*\| \leq \|\omega_i^n - \omega_s^*\|, \forall s \neq j\},$$

for any $j \in [k_*]$ as $k_*$ Voronoi cells with respect to the mixing measure $G_n$, where we denote $\omega_i^n := (\beta_{1i}^n, a_i^n, b_i^n, \sigma_i^n)$ and $\omega_j^* := (\beta_{1j}^*, a_j^*, b_j^*, \sigma_j^*)$. As we use asymptotic arguments in this proof, we can assume without loss of generality (WLOG) that these Voronoi cells does not depend on $n$, that is, $\mathcal{C}_j = \mathcal{C}_j^n$. Next, it follows from the hypothesis $\mathcal{D}_1(G_n, G_*) \to 0$ as $n \to \infty$ that each Voronoi cell contains only one element. Thus, we continue to assume WLOG that $\mathcal{C}_j = \{j\}$ for any $j \in [k_*]$, which implies that $(\beta_{1j}^n, a_j^n, b_j^n, \sigma_j^n) \to (\beta_{1j}^*, a_j^*, b_j^*, \sigma_j^*)$ and $\exp(\beta_{0j}^n) \to \exp(\beta_{0j}^*)$ as $n \to \infty$.

Subsequently, to specify the top $K$ selection in the formulations of $g_{G_n}(Y|X)$ and $g_{G_*}(Y|X)$, we divide the covariate space $\mathcal{X}$ into some subsets in two ways. In particular, we first consider $q := \binom{k_*}{K}$ different $K$-element subsets of $[k_*]$, which are assumed to take the form $\{\ell_1, \ldots, \ell_K\}$, for $\ell \in [q]$. Additionally, we denote $\{\ell_{K+1}, \ldots, \ell_{k_*}\} := [k_*] \setminus \{\ell_1, \ldots, \ell_K\}$. Then, we define for each $\ell \in [q]$ two following subsets of $\mathcal{X}$:

$$\mathcal{X}_\ell^n := \left\{x \in \mathcal{X} : (\beta_{1j}^n)^\top x \geq (\beta_{1j'}^n)^\top x : \forall j \in \{\ell_1, \ldots, \ell_K\}, j' \in \{\ell_{K+1}, \ldots, \ell_{k_*}\}\right\},$$

$$\mathcal{X}_\ell^* := \left\{x \in \mathcal{X} : (\beta_{1j}^*)^\top x \geq (\beta_{1j'}^*)^\top x : \forall j \in \{\ell_1, \ldots, \ell_K\}, j' \in \{\ell_{K+1}, \ldots, \ell_{k_*}\}\right\}.$$

Since $(\beta_{0j}^n, \beta_{1j}^n) \to (\beta_{0j}^*, \beta_{1j}^*)$ as $n \to \infty$ for any $j \in [k_*]$, we have for any arbitrarily small $\eta_j > 0$ that $\|\beta_{1j}^n - \beta_{1j}^*\| \leq \eta_j$ and $|\beta_{0j}^n - \beta_{0j}^*| \leq \eta_j$ for sufficiently large $n$. By applying Lemma 1, we obtain that $\mathcal{X}_\ell^n = \mathcal{X}_\ell^*$ for any $\ell \in [q]$ for sufficiently large $n$. WLOG, we assume that

$$\mathcal{D}_1(G_n, G_*) = \sum_{i=1}^{K} \left[ \exp(\beta_{0i}^n)\left( \|\Delta\beta_{1i}^n\| + \|\Delta a_i^n\| + \|\Delta b_i^n\| + \|\Delta\sigma_i^n\| \right) + \left| \exp(\beta_{0i}^n) - \exp(\beta_{0i}^*) \right| \right],$$

where we denote $\Delta\beta_{1i}^n := \beta_{1i}^n - \beta_{1i}^*$, $\Delta a_i^n := a_i^n - a_i^*$, $\Delta b_i^n := b_i^n - b_i^*$ and $\Delta\sigma_i^n := \sigma_i^n - \sigma_i^*$.

Let $\ell \in [q]$ such that $\{\ell_1, \ldots, \ell_K\} = \{1, \ldots, K\}$. Then, for almost surely $(X, Y) \in \mathcal{X}_\ell^* \times \mathcal{Y}$, we can rewrite the conditional densities $g_{G_n}(Y|X)$ and $g_{G_*}(Y|X)$ as

$$g_{G_n}(Y|X) = \sum_{i=1}^{K} \frac{\exp((\beta_{1i}^n)^\top X + \beta_{0i}^n)}{\sum_{j=1}^{K} \exp((\beta_{1j}^n)^\top X + \beta_{0j}^n)} \cdot f(Y|(a_i^n)^\top X + b_i^n, \sigma_i^n),$$

$$g_{G_*}(Y|X) = \sum_{i=1}^{K} \frac{\exp((\beta_{1i}^*)^\top X + \beta_{0i}^*)}{\sum_{j=1}^{K} \exp((\beta_{1j}^*)^\top X + \beta_{0j}^*)} \cdot f(Y|(a_i^*)^\top X + b_i^*, \sigma_i^*).$$

Now, we break the rest of our arguments into three steps:

**Step 1 - Taylor expansion**:

In this step, we take into account $H_n := \left[ \sum_{i=1}^{K} \exp((\beta_{1i}^*)^\top X + \beta_{0i}^*) \right] \cdot [g_{G_n}(Y|X) - g_{G_*}(Y|X)]$. Then, $H_n$ can be represented as follows:

$$H_n = \sum_{i=1}^{K} \exp(\beta_{0i}^n)\left[ \exp((\beta_{1i}^n)^\top X)f(Y|(a_i^n)^\top X + b_i^n, \sigma_i^n) - \exp((\beta_{1i}^n)^\top X)f(Y|(a_i^*)^\top X + b_i^*, \sigma_i^*) \right]$$

$$- \sum_{i=1}^{K} \exp(\beta_{0i}^n)\left[ \exp((\beta_{1i}^n)^\top X)g_{G_n}(Y|X) - \exp((\beta_{1i}^*)^\top X)g_{G_n}(Y|X) \right]$$

$$+ \sum_{i=1}^{K} \left[ \exp(\beta_{0i}^n) - \exp(\beta_{0i}^*) \right]\left[ \exp((\beta_{1i}^*)^\top X)f(Y|(a_i^*)^\top X + b_i^*, \sigma_i^*) - \exp((\beta_{1i}^*)^\top X)g_{G_n}(Y|X) \right].$$

By applying the first-order Taylor expansion to the first term in the above representation, which is denoted by $A_n$, we get that

$$A_n = \sum_{i=1}^{K} \sum_{|\alpha|=1} \frac{\exp(\beta_{0i}^n)}{\alpha!} \cdot (\Delta\beta_{1i}^n)^{\alpha_1}(\Delta a_i^n)^{\alpha_2}(\Delta b_i^n)^{\alpha_3}(\Delta\sigma_i^n)^{\alpha_4}$$

$$\times X^{\alpha_1+\alpha_2} \exp((\beta_{1i}^*)^\top X) \cdot \frac{\partial^{|\alpha_2|+\alpha_3+\alpha_4} f}{\partial h_1^{|\alpha_2|+\alpha_3}\partial\sigma^{\alpha_4}}(Y|(a_i^*)^\top X + b_i^*, \sigma_i^*) + R_1(X, Y),$$

where $R_1(X, Y)$ is a Taylor remainder that satisfies $R_1(X, Y)/\mathcal{D}_1'(X, Y) \to 0$ as $n \to \infty$. Recall that $f$ is the univariate Gaussian density, then we have

$$\frac{\partial^{\alpha_4} f}{\partial\sigma^{\alpha_4}}(Y|(a_i^*)^\top X + b_i^*, \sigma_i^*) = \frac{1}{2^{\alpha_4}} \cdot \frac{\partial^{2\alpha_4} f}{\partial h_1^{2\alpha_4}}(Y|(a_i^*)^\top X + b_i^*, \sigma_i^*),$$

which leads to

$$A_n = \sum_{i=1}^{K} \sum_{|\alpha|=1} \frac{\exp(\beta_{0i}^n)}{2^{\alpha_4}\alpha!} \cdot (\Delta\beta_{1i}^n)^{\alpha_1}(\Delta a_i^n)^{\alpha_2}(\Delta b_i^n)^{\alpha_3}(\Delta\sigma_i^n)^{\alpha_4}$$

$$\times X^{\alpha_1+\alpha_2} \exp((\beta_{1i}^*)^\top X) \cdot \frac{\partial^{|\alpha_2|+\alpha_3+2\alpha_4} f}{\partial h_1^{|\alpha_2|+\alpha_3+2\alpha_4}}(Y|(a_i^*)^\top X + b_i^*, \sigma_i^*) + R_1(X, Y)$$

$$= \sum_{i=1}^{K} \sum_{|\eta_1|+\eta_2=1} \sum_{\alpha\in\mathcal{J}_{\eta_1,\eta_2}} \frac{\exp(\beta_{0i}^n)}{2^{\alpha_4}\alpha!} \cdot (\Delta\beta_{1i}^n)^{\alpha_1}(\Delta a_i^n)^{\alpha_2}(\Delta b_i^n)^{\alpha_3}(\Delta\sigma_i^n)^{\alpha_4}$$

$$\times X^{\eta_1} \exp((\beta_{1i}^*)^\top X) \cdot \frac{\partial^{\eta_2} f}{\partial h_1^{\eta_2}}(Y|(a_i^*)^\top X + b_i^*, \sigma_i^*) + R_1(X, Y), \tag{16}$$

where we denote $\eta_1 = \alpha_1 + \alpha_2 \in \mathbb{N}^d$, $\eta_2 = |\alpha_2| + \alpha_3 + 2\alpha_4 \in \mathbb{N}$ and an index set

$$\mathcal{J}_{\eta_1,\eta_2} := \{(\alpha_i)_{i=1}^4 \in \mathbb{N}^d \times \mathbb{N}^d \times \mathbb{N} \times \mathbb{N} : \alpha_1 + \alpha_2 = \eta_1, \ \alpha_3 + 2\alpha_4 = \eta_2 - |\alpha_2|\}. \qquad (17)$$

By arguing in a similar fashion for the second term in the representation of $H_n$, we also get that

$$B_n := -\sum_{i=1}^K \sum_{|\gamma|=1} \frac{\exp(\beta_{0i}^n)}{\gamma!}(\Delta\beta_{1i}^n) \cdot X^\gamma \exp((\beta_{1i}^n)^\top X)g_{G_n}(Y|X) + R_2(X,Y),$$

where $R_2(X,Y)$ is a Taylor remainder such that $R_2(X,Y)/\mathcal{D}_1(G_n,G_*) \to 0$ as $n \to \infty$. Putting the above results together, we rewrite the quantity $H_n$ as follows:

$$H_n = \sum_{i=1}^K \sum_{0 \le |\eta_1|+\eta_2 \le 2} U_{i,\eta_1,\eta_2}^n \cdot X^{\eta_1} \exp((\beta_{1i}^*)^\top X)\frac{\partial^{\eta_2} f}{\partial h_1^{\eta_2}}(Y|(a_i^*)^\top X + b_i^*, \sigma_i^*)$$

$$+ \sum_{i=1}^K \sum_{0 \le |\gamma| \le 1} W_{i,\gamma}^n \cdot X^\gamma \exp((\beta_{1i}^*)^\top X)g_{G_n}(Y|X) + R_1(X,Y) + R_2(X,Y), \qquad (18)$$

in which we respectively define for each $i \in [K]$ that

$$U_{i,\eta_1,\eta_2}^n := \sum_{\alpha \in \mathcal{J}_{\eta_1,\eta_2}} \frac{\exp(\beta_{0i}^n)}{2^{\alpha_4}\alpha!} \cdot (\Delta\beta_{1i}^n)^{\alpha_1}(\Delta a_i^n)^{\alpha_2}(\Delta b_i^n)^{\alpha_3}(\Delta\sigma_i^n)^{\alpha_4},$$

$$W_{i,\gamma}^n := -\frac{\exp(\beta_{0i}^n)}{\gamma!}(\Delta\beta_{1i}^n)^\gamma,$$

for any $(\eta_1,\eta_2) \ne (\mathbf{0}_d, 0)$ and $|\gamma| \ne \mathbf{0}_d$. Otherwise, $U_{i,\mathbf{0}_d,0}^n = -W_{i,\mathbf{0}_d}^n := \exp(\beta_{0i}^n) - \exp(\beta_{0i}^*)$.

**Step 2 - Non-vanishing coefficients**:

Moving to the second step, we will show that not all the ratios $U_{i,\eta_1,\eta_2}^n/\mathcal{D}_1(G_n,G_*)$ tend to zero as $n$ goes to infinity. Assume by contrary that all of them approach zero when $n \to \infty$, then for $(\eta_1,\eta_2) = (\mathbf{0}_d, 0)$, it follows that

$$\frac{1}{\mathcal{D}_1(G_n,G_*)} \cdot \sum_{i=1}^K \left| \exp(\beta_{0i}^n) - \exp(\beta_{0i}^*) \right| = \sum_{i=1}^K \frac{|U_{j,\eta_1,\eta_2}^n|}{\mathcal{D}_1(G_n,G_*)} \to 0. \qquad (19)$$

Additionally, for tuples $(\eta_1,\eta_2)$ where $\eta_1 \in \{e_1, e_2, \dots, e_d\}$ with $e_j := (0, \dots, 0, \underbrace{1}_{j-th}, 0, \dots, 0)$ and $\eta_2 = 0$, we get

$$\frac{1}{\mathcal{D}_1(G_n,G_*)} \cdot \sum_{i=1}^K \exp(\beta_{0i}^n)\|\Delta\beta_{1i}^n\|_1 = \sum_{i=1}^K \frac{|U_{j,\eta_1,\eta_2}^n|}{\mathcal{D}_1(G_n,G_*)} \to 0.$$

By using similar arguments, we end up having

$$\frac{1}{\mathcal{D}_1(G_n,G_*)} \cdot \sum_{i=1}^K \exp(\beta_{0i}^n)\Big[\|\Delta\beta_{1i}^n\|_1 + \|\Delta a_i^n\|_1 + |\Delta b_i^n| + |\Delta\sigma_i^n|\Big] \to 0.$$

Due to the topological equivalence between norm-1 and norm-2, the above limit implies that

$$\frac{1}{\mathcal{D}_1(G_n,G_*)} \cdot \sum_{i=1}^K \exp(\beta_{0i}^n)\Big[\|\Delta\beta_{1i}^n\| + \|\Delta a_i^n\| + |\Delta b_i^n| + |\Delta\sigma_i^n|\Big] \to 0. \qquad (20)$$

Combine equation (19) with equation (20), we deduce that $\mathcal{D}_1(G_n,G_*)/\mathcal{D}_1(G_n,G_*) \to 0$, which is a contradiction. Consequently, at least one among the ratios $U_{i,\eta_1,\eta_2}^n/\mathcal{D}_1(G_n,G_*)$ does not vanish as $n$ tends to infinity.

**Step 3 - Fatou's contradiction**:

Let us denote by $m_n$ the maximum of the absolute values of $U_{i,\eta_1,\eta_2}^n/\mathcal{D}_1(G_n,G_*)$ and $W_{i,\gamma}^n/\mathcal{D}_1(G_n,G_*)$. It follows from the result achieved in Step 2 that $1/m_n \not\to \infty$.

Recall from the hypothesis that $\mathbb{E}_X[V(g_{G_n}(\cdot|X), g_{G_*}(\cdot|X))]/\mathcal{D}_1(G_n, G_*) \to 0$ as $n \to \infty$. Thus, by the Fatou's lemma, we have

$$0 = \lim_{n \to \infty} \frac{\mathbb{E}_X[V(g_{G_n}(\cdot|X), g_{G_*}(\cdot|X))]}{\mathcal{D}_1(G_n, G_*)} \geq \frac{1}{2} \cdot \int \liminf_{n \to \infty} \frac{|g_{G_n}(Y|X) - g_{G_*}(Y|X)|}{\mathcal{D}_1(G_n, G_*)} \mathrm{d}X \mathrm{d}Y.$$

This result indicates that $|g_{G_n}(Y|X) - g_{G_*}(Y|X)|/\mathcal{D}_1(G_n, G_*)$ tends to zero as $n$ goes to infinity for almost surely $(X, Y)$. As a result, it follows that

$$\lim_{n \to \infty} \frac{H_n}{m_n \mathcal{D}_1(G_n, G_*)} = \lim_{n \to \infty} \frac{|g_{G_n}(Y|X) - g_{G_*}(Y|X)|}{m_n \mathcal{D}_1(G_n, G_*)} = 0.$$

Next, let us denote $U^n_{i,\eta_1,\eta_2}/[m_n \mathcal{D}_1(G_n, G_*)] \to \tau_{i,\eta_1,\eta_2}$ and $W^n_{i,\gamma}/[m_n \mathcal{D}_1(G_n, G_*)] \to \kappa_{i,\gamma}$ with a note that at least one among them is non-zero. From the formulation of $H_n$ in equation (18), we deduce that

$$\sum_{i=1}^{K} \sum_{0 \leq |\eta_1| + \eta_2 \leq 2} \tau_{i,\eta_1,\eta_2} \cdot X^{\eta_1} \exp((\beta^*_{1i})^\top X) \frac{\partial^{\eta_2} f}{\partial h_1^{\eta_2}}(Y|(a^*_i)^\top X + b^*_i, \sigma^*_i)$$

$$+ \sum_{i=1}^{K} \sum_{0 \leq |\gamma| \leq 1} \kappa_{i,\gamma} \cdot X^\gamma \exp((\beta^*_{1i})^\top X) g_{G_*}(Y|X) = 0, \qquad (21)$$

for almost surely $(X, Y)$. This equation is equivalent to

$$\sum_{i=1}^{K} \sum_{0 \leq |\eta_1| \leq 1} \left[ \sum_{0 \leq \eta_2 \leq 2 - |\gamma|} \tau_{i,\eta_1,\eta_2} \frac{\partial^{\eta_2} f}{\partial h_1^{\eta_2}}(Y|(a^*_i)^\top X + b^*_i, \sigma^*_i) + \kappa_{i,\eta_1} g_{G_*}(Y|X) \right]$$

$$\times X^{\eta_1} \exp((\beta^*_{1i})^\top X) = 0,$$

for almost surely $(X, Y)$. Note that $\beta^*_{11}, \ldots, \beta^*_{1K}$ admits pair-wise different values, then $\{\exp((\beta^*_{1i})^\top X) : i \in [K]\}$ is a linearly independent set, which leads to

$$\sum_{0 \leq |\eta_1| \leq 1} \left[ \sum_{0 \leq \eta_2 \leq 2 - |\gamma|} \tau_{i,\eta_1,\eta_2} \frac{\partial^{\eta_2} f}{\partial h_1^{\eta_2}}(Y|(a^*_i)^\top X + b^*_i, \sigma^*_i) + \kappa_{i,\eta_1} g_{G_*}(Y|X) \right] X^{\eta_1} = 0,$$

for any $i \in [K]$ for almost surely $(X, Y)$. It is clear that the left hand side of the above equation is a polynomial of $X$ belonging to the compact set $\mathcal{X}$. As a result, we get that

$$\sum_{0 \leq \eta_2 \leq 2 - |\gamma|} \tau_{i,\eta_1,\eta_2} \frac{\partial^{\eta_2} f}{\partial h_1^{\eta_2}}(Y|(a^*_i)^\top X + b^*_i, \sigma^*_i) + \kappa_{i,\eta_1} g_{G_*}(Y|X) = 0,$$

for any $i \in [K]$, $0 \leq |\eta_1| \leq 1$ and almost surely $(X, Y)$. Since $(a^*_1, b^*_1, \sigma^*_1), \ldots, (a^*_K, b^*_K, \sigma^*_K)$ have pair-wise distinct values, those of $((a^*_1)^\top X + b^*_1, \sigma^*_1), \ldots, ((a^*_K)^\top X + b^*_K, \sigma^*_K)$ are also pair-wise different. Thus, the set $\left\{ \frac{\partial^{\eta_2} f}{\partial h_1^{\eta_2}}(Y|(a^*_i)^\top X + b^*_i, \sigma^*_i), g_{G_*}(Y|X) : i \in [K] \right\}$ is linearly independent. Consequently, we obtain that $\tau_{i,\eta_1,\eta_2} = \kappa_{i,\gamma} = 0$ for any $i \in [K]$, $0 \leq |\eta_1| + \eta_2 \leq 2$ and $0 \leq |\gamma| \leq 1$, which contradicts the fact that at least one among these terms is different from zero.

Hence, we can find some constant $\varepsilon' > 0$ such that

$$\inf_{G \in \mathcal{E}_{k_*}(\Omega) : \mathcal{D}_1(G, G_*) \leq \varepsilon'} \frac{\mathbb{E}_X[V(g_G(\cdot|X), g_{G_*}(\cdot|X))]}{\mathcal{D}_1(G, G_*)} > 0.$$

**Proof of claim** (15): Assume by contrary that this claim is not true, then we can seek a sequence $G'_n \in \mathcal{E}_{k_*}(\Omega)$ such that $\mathcal{D}_1(G'_n, G_*) > \varepsilon'$ and

$$\lim_{n \to \infty} \frac{\mathbb{E}_X[V(g_{G'_n}(\cdot|X), g_{G_*}(\cdot|X))]}{\mathcal{D}_1(G'_n, G_*)} = 0,$$

which directly implies that $\mathbb{E}_X[V(g_{G'_n}(\cdot|X), g_{G_*}(\cdot|X))] \to 0$ as $n \to \infty$. Recall that $\Omega$ is a compact set, therefore, we can replace the sequence $G'_n$ by one of its subsequences that converges to a mixing measure $G' \in \mathcal{E}_{k_*}(\Omega)$. Since $\mathcal{D}_1(G'_n, G_*) > \varepsilon'$, this result induces that $\mathcal{D}_1(G', G_*) > \varepsilon'$.

Subsequently, by means of the Fatou's lemma, we achieve that

$$0 = \lim_{n \to \infty} \mathbb{E}_X[2V(g_{G'_n}(\cdot|X), g_{G_*}(\cdot|X))] \geq \int \liminf_{n \to \infty} \left| g_{G'_n}(Y|X) - g_{G_*}(Y|X) \right| \, \mathrm{d}\mu(Y)\nu(X).$$

It follows that $g_{G'}(Y|X) = g_{G_*}(Y|X)$ for almost surely $(X, Y)$. From Proposition 2, we know that the top-K sparse softmax gating Gaussian mixture of experts is identifiable, thus, we obtain that $G' \equiv G_*$. As a consequence, $\mathcal{D}_1(G', G_*) = 0$, contradicting the fact that $\mathcal{D}_1(G', G_*) > \varepsilon' > 0$.

Hence, the proof is completed.

### A.3 Proof of Lemma 1

Let $\eta_i = M_i\varepsilon$, where $\varepsilon$ is some fixed positive constant and $M_i$ will be chosen later. For an arbitrary $\ell \in [q]$, since $\mathcal{X}$ and $\Omega$ are bounded sets, there exists some constant $c_\ell^* \geq 0$ such that

$$\min_{x,i,i'} \left[ (\beta_{1i}^*)^\top x - (\beta_{1i'}^*)^\top x \right] = c_\ell^* \varepsilon, \tag{22}$$

where the minimum is subject to $x \in \mathcal{X}_\ell^*, i \in \{\ell_1, \dots, \ell_K\}$ and $i' \in \{\ell_{K+1}, \dots, \ell_{k_*}\}$. We will point out that $c_\ell^* > 0$. Assume by contrary that $c_\ell^* = 0$. For $x \in \mathcal{X}_\ell^*$, we may assume for any $1 \leq i < j \leq k_*$ that

$$(\beta_{1\ell_i}^*)^\top x \geq (\beta_{1\ell_j}^*)^\top x.$$

Since $c_\ell^* = 0$, it follows from equation (22) that $(\beta_{1\ell_K}^*)^\top x - (\beta_{1\ell_{K+1}}^*)^\top x = 0$, or equivalently

$$(\beta_{1\ell_K}^* - \beta_{1\ell_{K+1}}^*)^\top x = 0.$$

In other words, $\mathcal{X}_\ell^*$ is a subset of

$$\mathcal{Z} := \{ x \in \mathcal{X} : (\beta_{1\ell_K}^* - \beta_{1\ell_{K+1}}^*)^\top x = 0 \}.$$

Since $\beta_{1\ell_K} - \beta_{1\ell_{K+1}} \neq \mathbf{0}_d$ and the distribution of $X$ is continuous, it follows that the set $\mathcal{Z}$ has measure zero. Since $\mathcal{X}_\ell^* \subseteq \mathcal{Z}$, we can conclude that $\mathcal{X}_\ell^*$ also has measure zero, which contradicts the hypothesis of Lemma 1. Therefore, we must have $c_\ell^* > 0$.

As $\mathcal{X}$ is a bounded set, we assume that $\|x\| \leq B$ for any $x \in \mathcal{X}$. Let $x \in \mathcal{X}_\ell^*$, then we have for any $i \in \{\ell_1, \dots, \ell_K\}$ and $i' \in \{\ell_{K+1}, \dots, \ell_{k_*}\}$ that

$$\begin{aligned} \beta_{1i}^\top x &= (\beta_{1i} - \beta_{1i}^*)^\top x + (\beta_{1i}^*)^\top x \\ &\geq -M_i\varepsilon B + (\beta_{1i'}^*)^\top x + c_\ell^* \varepsilon \\ &= -M_i\varepsilon B + c_\ell^* \varepsilon + (\beta_{1i'}^* - \beta_{1i'})^\top x + \beta_{1i'}^\top x \\ &\geq -2M_i\varepsilon B + + c_\ell^* \varepsilon + \beta_{1i'}^\top x. \end{aligned}$$

By setting $M_i \leq \dfrac{c_\ell^*}{2B}$, we get that $x \in \mathcal{X}_\ell$, which means that $\mathcal{X}_\ell^* \subseteq \mathcal{X}_\ell$. Similarly, assume that there exists some constant $c_\ell \geq 0$ that satisfies

$$\min_{x,i,i'} \left[ (\beta_{1i}^*)^\top x - (\beta_{1i'}^*)^\top x \right] = c_\ell^* \varepsilon.$$

Here, the above minimum is subject to $x \in \mathcal{X}_\ell$, $i \in \{\ell_1, \dots, \ell_K\}$ and $i' \in \{\ell_{K+1}, \dots, \ell_{k_*}\}$. If $M_i \leq \dfrac{c_\ell}{2B}$, then we also receive that $\mathcal{X}_\ell \subseteq \mathcal{X}_\ell^*$.

Hence, if we set $M_i = \dfrac{1}{2B} \min\{c_\ell^*, c_\ell\}$, we reach the conclusion that $\mathcal{X}_\ell^* = \mathcal{X}_\ell$.

## B Proof for Results under Over-specified Settings

In this appendix, we first provide the proofs of Theorem B.1 and Theorem 4 in Appendix B.1 and Appendix B.2, respectively. Subsequently, we present the proof for Proposition 1 in Appendix B.3, while that for Lemma 2 is put in Appendix B.4.

### B.1 PROOF OF THEOREM 3

In this appendix, we follow proof techniques presented in Appendix A.1 to demonstrate the result of Theorem 3. Recall that under the over-specified settings, the MLE $\widehat{G}_n$ belongs to the set of all mixing measures with at most $k > k_*$ components, i.e. $\mathcal{O}_k(\Omega)$. Interestingly, if we can adapt the result of part (i) of Lemma 3 to the over-specified settings, then other results presented in Appendix A.1 will also hold true. Therefore, our main goal is to derive following bound for any $0 < \eta < 1/2$ under the over-specified settings:

$$\log N(\eta, \mathcal{P}_k(\Omega), \|\cdot\|_1) \lesssim \log(1/\eta),$$

where $\mathcal{P}_k(\Omega) := \{\overline{g}_G(Y|X) : G \in \mathcal{O}_k(\Omega)\}$. For ease of presentation, we will reuse the notations defined in Appendix A.1 with $\mathcal{E}_{k_*}(\Omega)$ being replaced by $\mathcal{O}_k(\Omega)$. Now, let us recall necessary notations for this proof.

Firstly, we define $\Theta = \{(a, b, \sigma) \in \mathbb{R}^d \times \mathbb{R} \times \mathbb{R}_+ : (\beta_0, \beta_1, a, b, \sigma) \in \Omega\}$, and $\overline{\Theta}_\eta$ is an $\eta$-cover of $\Theta$. Additionally, we also denote $\Delta := \{(\beta_0, \beta_1) \in \mathbb{R}^d \times \mathbb{R} : (\beta_0, \beta_1, a, b, \sigma) \in \Omega\}$, and $\overline{\Delta}_\eta$ be an $\eta$-cover of $\Delta$. Next, for each mixing measure $G = \sum_{i=1}^k \exp(\beta_{0i})\delta_{(\beta_{1i}, a_i, b_i, \sigma_i)} \in \mathcal{O}_k(\Omega)$, we denote $G' = \sum_{i=1}^k \exp(\beta_{0i})\delta_{(\beta_{1i}, \overline{a}_i, \overline{b}_i, \overline{\sigma}_i)}$ in which $(\overline{a}_i, \overline{b}_i, \overline{\sigma}_i) \in \overline{\Theta}_\eta$ is the closest point to $(a_i, b_i, \sigma_i)$ in this set for any $i \in [k]$. We also consider another mixing measure $\overline{G} := \sum_{i=1}^k \exp(\overline{\beta}_{0i})\delta_{(\overline{\beta}_{1i}, \overline{a}_i, \overline{b}_i, \overline{\sigma}_i)} \in \mathcal{O}_k(\Omega)$ where $(\overline{\beta}_{0i}, \overline{\beta}_{1i}) \in \overline{\Delta}_\eta$ is the closest point to $(\beta_{0i}, \beta_{1i})$ in this set for any $i \in [k]$.

Subsequently, we define

$$\mathcal{L} := \{\overline{g}_{\overline{G}} \in \mathcal{P}_k(\Omega) : (\overline{\beta}_{0i}, \overline{\beta}_{1i}) \in \overline{\Delta}_\eta, (\overline{a}_i, \overline{b}_i, \overline{\sigma}_i) \in \overline{\Theta}_\eta\}.$$

We demonstrate that $\mathcal{L}$ is an $\eta$-cover of the metric space $(\mathcal{P}_k(\Omega), \|\cdot\|_1)$, that is, for any $\overline{g}_G \in \mathcal{P}_k(\Omega)$, there exists a density $\overline{g}_{\overline{G}} \in \mathcal{L}$ such that $\|\overline{g}_G - \overline{g}_{\overline{G}}\|_1 \le \eta$. By the triangle inequality, we have

$$\|\overline{g}_G - \overline{g}_{\overline{G}}\|_1 \le \|\overline{g}_G - \overline{g}_{G'}\|_1 + \|\overline{g}_{G'} - \overline{g}_{\overline{G}}\|_1. \tag{23}$$

From the formulation of $G'$, we get that

$$\|\overline{g}_G - \overline{g}_{G'}\|_1 \le \sum_{i=1}^k \int_{\mathcal{X} \times \mathcal{Y}} \left| f(Y|a_i^\top X + b_i, \sigma_i) - f(Y|\overline{a}_i^\top X + \overline{b}_i, \overline{\sigma}_i) \right| \mathrm{d}(X, Y)$$

$$\lesssim \sum_{i=1}^k \int_{\mathcal{X} \times \mathcal{Y}} \left( \|a_i - \overline{a}_i\| + |b_i - \overline{b}_i| + |\sigma_i - \overline{\sigma}_i| \right) \mathrm{d}(X, Y)$$

$$\lesssim \eta \tag{24}$$

Based on inequalities in equations (23) and (24), it is sufficient to show that $\|\overline{g}_{G'} - \overline{g}_{\overline{G}}\|_1 \lesssim \eta$. For any $\overline{\ell} \in [\overline{q}]$, let us define

$$\overline{\mathcal{X}}_{\overline{\ell}} := \{x \in \mathcal{X} : (\beta_{1i})^\top x \ge (\beta_{1i'})^\top x, \forall i \in \{\overline{\ell}_1, \ldots, \overline{\ell}_{\overline{K}}\}, i' \in \{\overline{\ell}_{\overline{K}+1}, \ldots, \overline{\ell}_k\}\},$$

$$\mathcal{X}'_{\overline{\ell}} := \{x \in \mathcal{X} : (\overline{\beta}_{1i})^\top x \ge (\overline{\beta}_{1i'})^\top x, \forall i \in \{\overline{\ell}_1, \ldots, \overline{\ell}_{\overline{K}}\}, i' \in \{\overline{\ell}_{\overline{K}+1}, \ldots, \overline{\ell}_k\}\}.$$

Since the Softmax function is differentiable, it is a Lipschitz function with some Lipschitz constant $L \ge 0$. Assume that $\|X\| \le B$ for any $X \in \mathcal{X}$ and denote

$$\pi_{\overline{\ell}}(X) := \left( \beta_{1\overline{\ell}_i}^\top x + \beta_{0\overline{\ell}_i}^\top \right)_{i=1}^{\overline{K}}; \qquad \overline{\pi}_{\overline{\ell}}(X) := \left( \overline{\beta}_{1\overline{\ell}_i}^\top x + \overline{\beta}_{0\overline{\ell}_i}^\top \right)_{i=1}^{\overline{K}},$$

for any $\overline{K}$-element subset $\{\overline{\ell}_1, \ldots \overline{\ell}_K\}$ of $\{1, \ldots, k\}$. Then, we have

$$\|\mathrm{Softmax}(\pi_{\overline{\ell}}(X)) - \mathrm{Softmax}(\overline{\pi}_{\overline{\ell}}(X))\| \le L \cdot \|\pi_{\overline{\ell}}(X) - \overline{\pi}_{\overline{\ell}}(X)\|$$

$$\le L \cdot \sum_{i=1}^{\overline{K}} \left( \|\beta_{1\overline{\ell}_i} - \overline{\beta}_{1\overline{\ell}_i}\| \cdot \|X\| + |\beta_{0\overline{\ell}_i} - \overline{\beta}_{0\overline{\ell}_i}| \right)$$

$$\le L \cdot \sum_{i=1}^{\overline{K}} \left( \eta B + \eta \right)$$

$$\lesssim \eta.$$

By arguing similarly to the proof of Lemma 2 in Appendix B.4, we receive that either $\overline{\mathcal{X}}_{\overline{\ell}} = \mathcal{X}'_{\overline{\ell}}$ or $\overline{\mathcal{X}}_{\overline{\ell}}$ has measure zero for any $\overline{\ell} \in [\overline{q}]$. As a result, we deduce that

$$
\|\overline{g}_{G'} - \overline{g}_{G_*}\|_1 \leq \sum_{\overline{\ell}=1}^{\overline{q}} \int_{\overline{\mathcal{X}}_{\overline{\ell}} \times \mathcal{Y}} |\overline{g}_{G'}(Y|X) - \overline{g}_{\overline{G}}(Y|X)| \mathrm{d}(X,Y)
$$

$$
\leq \sum_{\overline{\ell}=1}^{\overline{q}} \int_{\overline{\mathcal{X}}_{\overline{\ell}} \times \mathcal{Y}} \sum_{i=1}^{\overline{K}} \left| \mathrm{Softmax}(\pi_{\overline{\ell}}(X)_i) - \mathrm{Softmax}(\overline{\pi}_{\overline{\ell}}(X)_i) \right| \cdot \left| f(Y|\overline{a}_{\overline{\ell}_i}^\top X + \overline{b}_{\overline{\ell}_i}, \overline{\sigma}_{\overline{\ell}_i}) \right| \mathrm{d}(X,Y)
$$

$$
\lesssim \eta.
$$

Thus, $\mathcal{L}$ is an $\eta$-cover of the metric space $(\mathcal{P}_k(\Omega), \|\cdot\|_1)$, which implies that

$$
N(\eta, \mathcal{P}_k(\Omega), \|\cdot\|_1) \lesssim |\overline{\Delta}_\eta| \times |\overline{\Theta}_\eta| \leq \mathcal{O}(\eta^{-(d+1)k}) \times \mathcal{O}(\eta^{-(d+3)k}) = \mathcal{O}(\eta^{-(2d+4)k}). \tag{25}
$$

Hence, $\log N(\eta, \mathcal{P}_k(\Omega), \|\cdot\|_1) \lesssim \log(1/\eta)$.

## B.2  PROOF OF THEOREM 4

Similar to the proof of Theorem 2 in Appendix A, our objective here is also to derive the Total Variation lower bound adapted to the over-fitted settings:

$$
\mathbb{E}_X[V(\overline{g}_G(\cdot|X), g_{G_*}(\cdot|X))] \gtrsim \mathcal{D}_2(G, G_*).
$$

Since the global part of the above inequality can be argued in the same fashion as in Appendix A, we will focus only on demonstrating the following local part via the proof by contradiction method:

$$
\lim_{\varepsilon \to 0} \inf_{G \in \mathcal{O}_k(\Theta): \mathcal{D}_2(G, G_*) \leq \varepsilon} \frac{\mathbb{E}_X[V(\overline{g}_G(\cdot|X), g_{G_*}(\cdot|X))]}{\mathcal{D}_2(G, G_*)} > 0. \tag{26}
$$

Assume that the above claim does not hold true, then we can find a sequence of mixing measures $G_n := \sum_{i=1}^{k_n} \exp(\beta_{0i}^n) \delta_{(\beta_{1i}^n, a_i^n, b_i^n, \sigma_i^n)} \in \mathcal{O}_k(\Omega)$ such that both $\mathcal{D}_2(G_n, G_*)$ and $\mathbb{E}_X[V(\overline{g}_{G_n}(\cdot|X), g_{G_*}(\cdot|X))]/\mathcal{D}_2(G_n, G_*)$ vanish when $n$ goes to infinity. Additionally, by abuse of notation, we reuse the set of Voronoi cells $\mathcal{C}_j$, for $j \in [k_*]$, defined in Appendix A. Due to the limit $\mathcal{D}_2(G_n, G_*) \to 0$ as $n \to \infty$, it follows that for any $j \in [k_*]$, we have $\sum_{i \in \mathcal{C}_j} \exp(\beta_{0i}^n) \to \exp(\beta_{0j}^*)$ and $(\beta_{1i}^n, a_i^n, b_i^n, \sigma_i^n) \to (\beta_{1j}^*, a_j^*, b_j^*, \sigma_j^*)$ for all $i \in \mathcal{C}_j$. WLOG, we may assume that

$$
\mathcal{D}_2(G_n, G_*) = \sum_{\substack{j \in [K], \, i \in \mathcal{C}_j \\ |\mathcal{C}_j| > 1}} \exp(\beta_{0i}^n) \left[ \|\Delta\beta_{1ij}^n\|^{\overline{r}(|\mathcal{C}_j|)} + \|\Delta a_{ij}^n\|^{\frac{\overline{r}(|\mathcal{C}_j|)}{2}} + |\Delta b_{ij}^n|^{\overline{r}(|\mathcal{C}_j|)} + |\Delta\sigma_{ij}^n|^{\frac{\overline{r}(|\mathcal{C}_j|)}{2}} \right]
$$

$$
+ \sum_{\substack{j \in [K], \, i \in \mathcal{C}_j \\ |\mathcal{C}_j| = 1}} \exp(\beta_{0i}^n) \left[ \|\Delta\beta_{1ij}^n\| + \|\Delta a_{ij}^n\| + |\Delta b_{ij}^n| + |\Delta\sigma_{ij}^n| \right] + \sum_{j=1}^{K} \left| \sum_{i \in \mathcal{C}_j} \exp(\beta_{0i}^n) - \exp(\beta_{0j}^*) \right|.
$$

Regarding the top-$K$ selection in the conditional density $g_{G_*}$, we partition the covariate space $\mathcal{X}$ in a similar fashion to Appendix A. More specifically, we consider $q = \binom{k_*}{K}$ subsets $\{\ell_1, \dots, \ell_K\}$ of $\{1, \dots, k_*\}$ for any $\ell \in [q]$, and denote $\{\ell_{K+1}, \dots, \ell_{k_*}\} := [k_*] \setminus \{\ell_1, \dots, \ell_K\}$. Then, we define

$$
\mathcal{X}_\ell^* := \left\{ x \in \mathcal{X} : (\beta_{1j}^*)^\top x \geq (\beta_{1j'}^*)^\top x, \forall j \in \{\ell_1, \dots, \ell_K\}, j' \in \{\ell_{K+1}, \dots, \ell_{k_*}\} \right\},
$$

for any $\ell \in [q]$. On the other hand, we need to introduce a new partition method of the covariate space for the weight selection in the conditional density $g_{G_n}$. In particular, let $\overline{K} \in \mathbb{N}$ such that $\max_{\{\ell_j\}_{j=1}^K \subset [k_*]} \sum_{j=1}^K |\mathcal{C}_{\ell_j}| \leq \overline{K} \leq k$ and $\overline{q} := \binom{k}{\overline{K}}$. Then, for any $\overline{\ell} \in [\overline{q}]$, we denote $(\overline{\ell}_1, \dots, \overline{\ell}_k)$ as a subset of $[k]$ and $\{\overline{\ell}_{\overline{K}+1}, \dots, \overline{\ell}_k\} := [k] \setminus \{\overline{\ell}_1, \dots, \overline{\ell}_{\overline{K}}\}$. Additionally, we define

$$
\mathcal{X}_{\overline{\ell}}^n := \left\{ x \in \mathcal{X} : (\beta_{1i}^n)^\top x \geq (\beta_{1i'}^n)^\top x, \forall i \in \{\overline{\ell}_1, \dots, \overline{\ell}_{\overline{K}}\}, i' \in \{\overline{\ell}_{\overline{K}+1}, \dots, \overline{\ell}_k\} \right\}.
$$

Let $X \in \mathcal{X}_\ell^*$ for some $\ell \in [q]$ such that $\{\ell_1, \ldots, \ell_K\} = \{1, \ldots, K\}$. If $\{\overline{\ell}_1, \ldots \overline{\ell}_{\overline{K}}\} \neq \mathcal{C}_1 \cup \ldots \cup \mathcal{C}_K$ for any $\overline{\ell} \in [\overline{q}]$, then $\mathbb{E}_X[V(\overline{g}_{G_n}(\cdot|X), g_{G_*}(\cdot|X))]/\mathcal{D}_2(G_n, G_*) \nrightarrow 0$ as $n$ tends to infinity. This contradicts the fact that this term must approach zero. Therefore, we only need to consider the scenario when there exists $\overline{\ell} \in [\overline{q}]$ such that $\{\overline{\ell}_1, \ldots \overline{\ell}_{\overline{K}}\} = \mathcal{C}_1 \cup \ldots \cup \mathcal{C}_K$. Recall that we have $(\beta_{0i}^n, \beta_{1i}^n) \to (\beta_{0j}^*, \beta_{1j}^*)$ as $n \to \infty$ for any $j \in [k_*]$ and $i \in \mathcal{C}_j$. Thus, for any arbitrarily small $\eta_j > 0$, we have that $\|\beta_{1i}^n - \beta_{1j}^*\| \leq \eta_j$ and $|\beta_{0i}^n - \beta_{0j}^*| \leq \eta_j$ for sufficiently large $n$. Then, it follows from Lemma 2 that $\mathcal{X}_\ell^* = \mathcal{X}_{\overline{\ell}}^n$ for sufficiently large $n$. This result indicates that $X \in \mathcal{X}_{\overline{\ell}}^n$.

Then, we can represent the conditional densities $g_{G_*}(Y|X)$ and $g_{G_n}(Y|X)$ for any sufficiently large $n$ as follows:

$$g_{G_*}(Y|X) = \sum_{j=1}^K \frac{\exp((\beta_{1j}^*)^\top X + \beta_{0j}^*)}{\sum_{j'=1}^K \exp((\beta_{1j'}^*)^\top X + \beta_{0j'}^*)} \cdot f(Y|(a_j^*)^\top X + b_j^*, \sigma_j^*),$$

$$\overline{g}_{G_n}(Y|X) = \sum_{j=1}^K \sum_{i \in \mathcal{C}_j} \frac{\exp((\beta_{1i}^n)^\top X + \beta_{0i}^n)}{\sum_{j'=1}^K \sum_{i' \in \mathcal{C}_{j'}} \exp((\beta_{1i'}^n)^\top X + \beta_{0i'}^n)} \cdot f(Y|(a_i^n)^\top X + b_i^n, \sigma_i^n).$$

Now, we reuse the three-step framework in Appendix A.

**Step 1 - Taylor expansion**:

Firstly, by abuse of notations, let us consider the quantity

$$H_n := \Big[\sum_{j=1}^K \exp((\beta_{1j}^*)^\top X + \beta_{0j}^*)\Big] \cdot [\overline{g}_{G_n}(Y|X) - g_{G_*}(Y|X)].$$

Similar to Step 1 in Appendix A, we can express this term as

$$H_n = \sum_{j=1}^K \sum_{i \in \mathcal{C}_j} \exp(\beta_{0i}^n)\Big[\exp((\beta_{1i}^n)^\top X)f(Y|(a_i^n)^\top X + b_i^n, \sigma_i^n) - \exp((\beta_{1j}^*)^\top X)f(Y|(a_j^*)^\top X + b_j^*, \sigma_j^*)\Big]$$

$$- \sum_{j=1}^K \sum_{i \in \mathcal{C}_j} \exp(\beta_{0i}^n)\Big[\exp((\beta_{1i}^n)^\top X)g_{G_n}(Y|X) - \exp((\beta_{1j}^*)^\top X)g_{G_n}(Y|X)\Big]$$

$$+ \sum_{j=1}^K \Big[\sum_{i \in \mathcal{C}_j} \exp(\beta_{0i}^n) - \exp(\beta_{0j}^*)\Big]\Big[\exp((\beta_{1j}^*)^\top X)f(Y|(a_i^*)^\top X + b_i^*, \sigma_i^*) - \exp((\beta_{1j}^*)^\top X)g_{G_n}(Y|X)\Big]$$

$$:= A_n + B_n + E_n.$$

Next, we proceed to decompose $A_n$ based on the cardinality of the Voronoi cells as follows:

$$A_n = \sum_{j:|\mathcal{C}_j|=1} \sum_{i \in \mathcal{C}_j} \exp(\beta_{0i}^n)\Big[\exp((\beta_{1i}^n)^\top X)f(Y|(a_i^n)^\top X + b_i^n, \sigma_i^n) - \exp((\beta_{1j}^*)^\top X)f(Y|(a_j^*)^\top X + b_j^*, \sigma_j^*)\Big]$$

$$+ \sum_{j:|\mathcal{C}_j|>1} \sum_{i \in \mathcal{C}_j} \exp(\beta_{0i}^n)\Big[\exp((\beta_{1i}^n)^\top X)f(Y|(a_i^n)^\top X + b_i^n, \sigma_i^n) - \exp((\beta_{1j}^*)^\top X)f(Y|(a_j^*)^\top X + b_j^*, \sigma_j^*)\Big].$$

By applying the Taylor expansions of order 1 and $\overline{r}(|\mathcal{C}_j|)$ to the first and second terms of $A_n$, respectively, and following the derivation in equation (16), we arrive at

$$A_n = \sum_{j:|\mathcal{C}_j|=1} \sum_{i \in \mathcal{C}_j} \sum_{1 \leq |\eta_1|+\eta_2 \leq 2} \sum_{\alpha \in \mathcal{J}_{\eta_1,\eta_2}} \frac{\exp(\beta_{0i}^n)}{2^{\alpha_4}\alpha!} \cdot (\Delta\beta_{1i}^n)^{\alpha_1}(\Delta a_i^n)^{\alpha_2}(\Delta b_i^n)^{\alpha_3}(\Delta\sigma_i^n)^{\alpha_4}$$

$$\times X^{\eta_1} \exp((\beta_{1i}^*)^\top X) \cdot \frac{\partial^{\eta_2} f}{\partial h_1^{\eta_2}}(Y|(a_i^*)^\top X + b_i^*, \sigma_i^*) + R_3(X, Y)$$

$$+ \sum_{j:|\mathcal{C}_j|>1} \sum_{i \in \mathcal{C}_j} \sum_{1 \leq |\eta_1|+\eta_2 \leq 2\overline{r}(|\mathcal{C}_j|)} \sum_{\alpha \in \mathcal{J}_{\eta_1,\eta_2}} \frac{\exp(\beta_{0i}^n)}{2^{\alpha_4}\alpha!} \cdot (\Delta\beta_{1i}^n)^{\alpha_1}(\Delta a_i^n)^{\alpha_2}(\Delta b_i^n)^{\alpha_3}(\Delta\sigma_i^n)^{\alpha_4}$$

$$\times X^{\eta_1} \exp((\beta_{1i}^*)^\top X) \cdot \frac{\partial^{\eta_2} f}{\partial h_1^{\eta_2}}(Y|(a_i^*)^\top X + b_i^*, \sigma_i^*) + R_4(X, Y),$$

where the set $\mathcal{J}_{\eta_1,\eta_2}$ is defined in equation (17) while $R_i(X,Y)$ is a Taylor remainder such that $R_i(X,Y)/\mathcal{D}_2(G_n,G_*) \to 0$ as $n \to \infty$ for $i \in \{3,4\}$. Similarly, we also decompose $B_n$ according to the Voronoi cells as $A_n$ but then invoke the Taylor expansions of order 1 and 2 to the first term and the second term, respectively. In particular, we get

$$
B_n = -\sum_{j:|\mathcal{C}_j|=1}\sum_{i\in\mathcal{C}_j}\sum_{|\gamma|=1}\frac{\exp(\beta_{0i}^n)}{\gamma!}(\Delta\beta_{1i}^n)\cdot X^\gamma \exp((\beta_{1i}^n)^\top X)g_{G_n}(Y|X) + R_5(X,Y)
$$
$$
-\sum_{j:|\mathcal{C}_j|>1}\sum_{i\in\mathcal{C}_j}\sum_{1\leq|\gamma|\leq2}\frac{\exp(\beta_{0i}^n)}{\gamma!}(\Delta\beta_{1i}^n)\cdot X^\gamma \exp((\beta_{1i}^n)^\top X)g_{G_n}(Y|X) + R_6(X,Y),
$$

where $R_5(X,Y)$ and $R_6(X,Y)$ are Taylor remainders such that their ratios over $\mathcal{D}_2(G_n,G_*)$ approach zero as $n \to \infty$. Subsequently, let us define

$$
S_{j,\eta_1,\eta_2}^n := \sum_{i\in\mathcal{C}_j}\sum_{\alpha\in\mathcal{J}_{\eta_1,\eta_2}}\frac{\exp(\beta_{0i}^n)}{2^{\alpha_4}\alpha!}\cdot(\Delta\beta_{1i}^n)^{\alpha_1}(\Delta a_i^n)^{\alpha_2}(\Delta b_i^n)^{\alpha_3}(\Delta\sigma_i^n)^{\alpha_4},
$$

$$
T_{j,\gamma}^n := \sum_{i\in\mathcal{C}_j}\frac{\exp(\beta_{0i}^n)}{\gamma!}(\Delta\beta_{1i}^n)^\gamma,
$$

for any $(\eta_1,\eta_2)\neq(\mathbf{0}_d,0)$ and $|\gamma|\neq\mathbf{0}_d$, while for $(\eta_1,\eta_2)=(\mathbf{0}_d,0)$ we set

$$
S_{i,\mathbf{0}_d,0}^n = -T_{i,\mathbf{0}_d}^n := \sum_{i\in\mathcal{C}_j}\exp(\beta_{0i}^n) - \exp(\beta_{0i}^*).
$$

As a consequence, it follows that

$$
H_n = \sum_{j=1}^K\sum_{|\eta_1|+\eta_2=0}^{2\bar{r}(|\mathcal{C}_j|)}S_{j,\eta_1,\eta_2}^n\cdot X^{\eta_1}\exp((\beta_{1i}^*)^\top X)\cdot\frac{\partial^{\eta_2}f}{\partial h_1^{\eta_2}}(Y|(a_i^*)^\top X+b_i^*,\sigma_i^*)
$$
$$
+\sum_{j=1}^K\sum_{|\gamma|=0}^{1+\mathbf{1}_{\{|\mathcal{C}_j|>1\}}}T_{j,\gamma}^n\cdot X^\gamma\exp((\beta_{1i}^n)^\top X)g_{G_n}(Y|X)+R_5(X,Y)+R_6(X,Y). \qquad (27)
$$

**Step 2 - Non-vanishing coefficients**:

In this step, we will prove by contradiction that at least one among the ratios $S_{j,\eta_1,\eta_2}^n/\mathcal{D}_2(G_n,G_*)$ does not converge to zero as $n \to \infty$. Assume that all these terms go to zero, then by employing arguments for deriving equations (19) and (20), we get that

$$
\frac{1}{\mathcal{D}_2(G_n,G_*)}\cdot\Big[\sum_{j=1}^K\Big|\sum_{i\in\mathcal{C}_j}\exp(\beta_{0i}^n)-\exp(\beta_{0j}^*)\Big|
$$
$$
+\sum_{j:|\mathcal{C}_j|=1}\sum_{i\in\mathcal{C}_j}\exp(\beta_{0i}^n)\Big(\|\Delta\beta_{1ij}^n\|+\|\Delta a_{ij}^n\|+|\Delta b_{ij}^n|+|\Delta\sigma_{ij}^n|\Big)\Big]\to0.
$$

Combine this limit with the representation of $\mathcal{D}_2(G_n,G_*)$, we have that

$$
\frac{1}{\mathcal{D}_2(G_n,G_*)}\cdot\sum_{j:|\mathcal{C}_j|>1}\sum_{i\in\mathcal{C}_j}\exp(\beta_{0i}^n)\Big(\|\Delta\beta_{1ij}^n\|^{\bar{r}(|\mathcal{C}_j|)}+\|\Delta a_{ij}^n\|^{\frac{\bar{r}(|\mathcal{C}_j|)}{2}}+|\Delta b_{ij}^n|^{\bar{r}(|\mathcal{C}_j|)}+|\Delta\sigma_{ij}^n|^{\frac{\bar{r}(|\mathcal{C}_j|)}{2}}\Big)\not\to0.
$$

This result implies that we can find some index $j'\in[K]:|\mathcal{C}_{j'}|>1$ that satisfies

$$
\frac{1}{\mathcal{D}_2(G_n,G_*)}\cdot\sum_{i\in\mathcal{C}_{j'}}\exp(\beta_{0i}^n)\Big(\|\Delta\beta_{1ij'}^n\|^{\bar{r}(|\mathcal{C}_{j'}|)}+\|\Delta a_{ij'}^n\|^{\frac{\bar{r}(|\mathcal{C}_{j'}|)}{2}}+|\Delta b_{ij'}^n|^{\bar{r}(|\mathcal{C}_{j'}|)}+|\Delta\sigma_{ij'}^n|^{\frac{\bar{r}(|\mathcal{C}_{j'}|)}{2}}\Big)\not\to0.
$$

For simplicity, we may assume that $j'=1$. Since $S_{1,\eta_1,\eta_2}^n/\mathcal{D}_2(G_n,G_*)$ vanishes as $n\to\infty$ for any $(\eta_1,\eta_2)\in\mathbb{N}^d\times\mathbb{N}$ such that $1\leq|\eta_1|+\eta_2\leq\bar{r}(|\mathcal{C}_j|)$, we divide this term by the left hand side of the above equation and achieve that

$$
\frac{\sum_{i\in\mathcal{C}_1}\sum_{\alpha\in\mathcal{J}_{\eta_1,\eta_2}}\frac{\exp(\beta_{0i}^n)}{2^{\alpha_4}\alpha!}(\Delta\beta_{1i1}^n)^{\alpha_1}(\Delta a_{i1}^n)^{\alpha_2}(\Delta b_{i1}^n)^{\alpha_3}(\Delta\sigma_{i1}^n)^{\alpha_4}}{\sum_{i\in\mathcal{C}_1}\exp(\beta_{0i}^n)\Big(\|\Delta b_{i1}^n\|^{\bar{r}(|\mathcal{C}_1|)}+\|\Delta a_{i1}^n\|^{\frac{\bar{r}(|\mathcal{C}_1|)}{2}}+|\Delta b_{i1}^n|^{\bar{r}(|\mathcal{C}_1|)}+|\Delta\sigma_{i1}^n|^{\frac{\bar{r}(|\mathcal{C}_1|)}{2}}\Big)}\to0, \qquad (28)
$$

for any $(\eta_1, \eta_2) \in \mathbb{N}^d \times \mathbb{N}$ such that $1 \leq |\eta_1| + \eta_2 \leq \bar{r}(|\mathcal{C}_1|)$.

Subsequently, we define $M_n := \max\{\|\Delta\beta_{1i1}^n\|, \|\Delta a_{i1}^n\|^{1/2}, |\Delta b_{i1}^n|, |\Delta\sigma_{i1}^n|^{1/2} : i \in \mathcal{C}_1\}$ and $p_n := \max\{\exp(\beta_{0i}^n) : i \in \mathcal{C}_1\}$. As a result, the sequence $\exp(\beta_{0i}^n)/p_n$ is bounded, which indicates that we can substitute it with its subsequence that admits a positive limit $z_{5i}^2 := \lim_{n\to\infty} \exp(\beta_{0i}^n)/p_n$. Therefore, at least one among the limits $z_{5i}^2$ equals to one. Furthermore, we also denote

$$(\Delta\beta_{1i1}^n)/M_n \to z_{1i}, \ (\Delta a_{i1}^n)/M_n \to z_{2i}, \ (\Delta b_{i1}^n)/M_n \to z_{3i}, \ (\Delta\sigma_{i1}^n)/(2M_n) \to z_{4i}.$$

From the above definition, it follows that at least one among the limits $z_{1i}, z_{2i}, z_{3i}$ and $z_{4i}$ equals to either 1 or $-1$. By dividing both the numerator and the denominator of the term in equation (28) by $p_n M_n^{|\eta_1|+\eta_2}$, we arrive at the following system of polynomial equations:

$$\sum_{i\in\mathcal{C}_1} \sum_{\alpha\in\mathcal{J}_{\eta_1,\eta_2}} \frac{z_{5i}^2 \, z_{1i}^{\alpha_1} \, z_{2i}^{\alpha_2} \, z_{3i}^{\alpha_3} \, z_{4i}^{\alpha_4}}{\alpha_1! \, \alpha_2! \, \alpha_3! \, \alpha_4!} = 0,$$

for all $(\eta_1, \eta_2) \in \mathbb{N}^d \times \mathbb{N} : 1 \leq |\eta_1| + \eta_2 \leq \bar{r}(|\mathcal{C}_1|)$. Nevertheless, from the definition of $\bar{r}(|\mathcal{C}_1|)$, we know that the above system does not admit any non-trivial solutions, which is a contradiction. Consequently, not all the ratios $S_{j,\eta_1,\eta_2}^n/\mathcal{D}_2(G_n, G_*)$ tend to zero as $n$ goes to infinity.

**Step 3 - Fatou's contradiction**:

It follows from the hypothesis that $\mathbb{E}_X[V(\bar{g}_{G_n}(\cdot|X), g_{G_*}(\cdot|X))]/\mathcal{D}_2(G_n, G_*) \to 0$ as $n \to \infty$. Then, by applying the Fatou's lemma, we get

$$0 = \lim_{n\to\infty} \frac{\mathbb{E}_X[V(\bar{g}_{G_n}(\cdot|X), g_{G_*}(\cdot|X))]}{\mathcal{D}_2(G_n, G_*)} = \frac{1}{2} \cdot \int \liminf_{n\to\infty} \frac{|\bar{g}_{G_n}(Y|X) - g_{G_*}(Y|X)|}{\mathcal{D}_2(G_n, G_*)} \mathrm{d}X\mathrm{d}Y,$$

which implies that $|\bar{g}_{G_n}(Y|X) - g_{G_*}(Y|X)|/\mathcal{D}_2(G_n, G_*) \to 0$ as $n \to \infty$ for almost surely $(X, Y)$.

Next, we define $m_n$ as the maximum of the absolute values of $S_{j,\eta_1,\eta_2}^n/\mathcal{D}_2(G_n, G_*)$. It follows from Step 2 that $1/m_n \not\to \infty$. Moreover, by arguing in the same way as in Step 3 in Appendix A, we receive that

$$H_n/[m_n\mathcal{D}_2(G_n, G_*)] \to 0 \tag{29}$$

as $n \to \infty$. By abuse of notations, let us denote

$$S_{j,\eta_1,\eta_2}^n/[m_n\mathcal{D}_2(G_n, G_*)] \to \tau_{j,\eta_1,\eta_2},$$
$$T_{j,\gamma}^n/[m_n\mathcal{D}_2(G_n, G_*)] \to \kappa_{j,\gamma}.$$

Here, at least one among $\tau_{j,\eta_1,\eta_2}, \kappa_{j,\gamma}$ is non-zero. Then, by putting the results in equations (27) and (29) together, we get

$$\sum_{i=1}^K \sum_{|\eta_1|+\eta_2=0}^{2\bar{r}(|\mathcal{C}_j|)} \tau_{i,\eta_1,\eta_2} \cdot X^{\eta_1} \exp((\beta_{1i}^*)^\top X) \frac{\partial^{\eta_2} f}{\partial h_1^{\eta_2}}(Y|(a_i^*)^\top X + b_i^*, \sigma_i^*)$$

$$+ \sum_{i=1}^K \sum_{|\gamma|=0}^{1+\mathbf{1}_{\{|\mathcal{C}_j|>1\}}} \kappa_{i,\gamma} \cdot X^\gamma \exp((\beta_{1i}^*)^\top X) g_{G_*}(Y|X) = 0.$$

Arguing in a similar fashion as in Step 3 of Appendix A, we obtain that $\tau_{j,\eta_1,\eta_2} = \kappa_{j,\gamma} = 0$ for any $j \in [K], 0 \leq |\eta_1| + \eta_2 \leq 2\bar{r}(|\mathcal{C}_j|)$ and $0 \leq |\gamma| \leq 1 + \mathbf{1}_{\{|\mathcal{C}_j|>1\}}$. This contradicts the fact that at least one among them is non-zero. Hence, the proof is completed.

### B.3 Proof of Proposition 1

Since the Hellinger distance is lower bounded by the Total Variation distance, i.e. $h \geq V$, it is sufficient to show that

$$\inf_{G\in\mathcal{O}_k(\Omega)} \mathbb{E}_X[V(\bar{g}_G(\cdot|X), g_{G_*}(\cdot|X))] > 0.$$

For that purpose, we first demonstrate that

$$\lim_{\varepsilon \to 0} \inf_{G \in \mathcal{O}_k(\Omega): \mathcal{D}_2(G, G_*) \leq \varepsilon} \mathbb{E}_X[V(\overline{g}(\cdot|X), g_{G_*}(\cdot|X))] > 0. \tag{30}$$

Assume by contrary that the above claim is not true, then we can find a sequence $G_n = \sum_{i=1}^{k_n} \exp(\beta_{0i}^n) \delta_{(\beta_{1i}^n, a_i^n, b_i^n, \sigma_i^n)} \in \mathcal{O}_k(\Omega)$ that satisfies $\mathcal{D}_2(G_n, G_*) \to 0$ and

$$\mathbb{E}_X[V(\overline{g}_{G_n}(\cdot|X), g_{G_*}(\cdot|X))] \to 0$$

when $n$ tends to infinity. By applying the Fatou's lemma, we have

$$0 = \lim_{n \to \infty} \mathbb{E}_X[V(\overline{g}_{G_n}(\cdot|X), g_{G_*}(\cdot|X))]$$

$$\geq \frac{1}{2} \int_{\mathcal{X} \times \mathcal{Y}} \liminf_{n \to \infty} |\overline{g}_{G_n}(Y|X) - g_{G_*}(Y|X)| \mathrm{d}(X, Y). \tag{31}$$

The above results indicates that $\overline{g}_{G_n}(Y|X) - g_{G_*}(Y|X) \to 0$ as $n \to \infty$ for almost surely $(X, Y)$. WLOG, we may assume that

$$\max_{\{\ell_1, \ldots, \ell_K\}} \sum_{j=1}^K |\mathcal{C}_{\ell_j}| = |\mathcal{C}_1| + |\mathcal{C}_2| + \ldots + |\mathcal{C}_K|.$$

Let us consider $X \in \mathcal{X}_\ell^*$, where $\ell \in [q]$ such that $\{\ell_1, \ldots, \ell_K\} = \{1, \ldots, K\}$. Since $\mathcal{D}_2(G_n, G_*)$ converges to zero, it follows that $(\beta_{1i}^n, a_i^n, b_i^n, \sigma_i^n) \to (\beta_{1j}^*, a_j^*, b_j^*, \sigma_j^*)$ and $\sum_{i \in \mathcal{C}_j} \exp(\beta_{0i}^n) \to \exp(\beta_{0j}^*)$ for any $i \in \mathcal{C}_j$ and $j \in [k_*]$. Thus, we must have that $X \in \overline{\mathcal{X}}_{\overline{\ell}}$ for some $\overline{\ell} \in [\overline{q}]$ such that $\{\overline{\ell}_1, \ldots, \overline{\ell}_{\overline{K}}\} = \mathcal{C}_1 \cup \ldots \cup \mathcal{C}_K$. Otherwise, $\overline{g}_{G_n}(Y|X) - g_{G_*}(Y|X) \not\to 0$, which is a contradiction. However, as $\overline{K} < \sum_{j=1}^K |\mathcal{C}_j|$, the fact that $\{\overline{\ell}_1, \ldots, \overline{\ell}_{\overline{K}}\} = \mathcal{C}_1 \cup \ldots \cup \mathcal{C}_K$ cannot occur. Therefore, we reach the claim in equation (30). Consequently, there exists some positive constant $\varepsilon'$ such that

$$\inf_{G \in \mathcal{O}_k(\Omega): \mathcal{D}_2(G, G_*) \leq \varepsilon'} \mathbb{E}_X[V(\overline{g}_G(\cdot|X), g_{G_*}(\cdot|X))] > 0.$$

Given the above result, it suffices to point out that

$$\inf_{G \in \mathcal{O}_k(\Omega): \mathcal{D}_2(G, G_*) > \varepsilon'} \mathbb{E}_X[V(\overline{g}_G(\cdot|X), g_{G_*}(\cdot|X))] > 0. \tag{32}$$

We continue to use the proof by contradiction method here. In particular, assume that the inequality (32) does not hold, then there exists a sequence of mixing measures $G_n' \in \mathcal{O}_k(\Omega)$ such that $\mathcal{D}_2(G_n', G_*) > \varepsilon'$ and

$$\mathbb{E}_X[V(\overline{g}_{G_n'}(\cdot|X), g_{G_*}(\cdot|X))] \to 0.$$

By invoking the Fatou's lemma as in equation (31), we get that $\overline{g}_{G_n'}(Y|X) - g_{G_*}(Y|X) \to 0$ as $n \to \infty$ for almost surely $(X, Y)$. Since $\Omega$ is a compact set, we can substitute $(G_n)$ with its subsequence which converges to some mixing measure $G' \in \mathcal{O}_k(\Omega)$. Then, the previous limit implies that $\overline{g}_{G'}(Y|X) = g_{G_*}(Y|X)$ for almost surely $(X, Y)$. From the result of Proposition 2 in Appendix C, we know that the top-K sparse softmax gating Gaussian MoE is identifiable. Therefore, we obtain that $G' \equiv G_*$, or equivalently, $\mathcal{D}_2(G', G_*) = 0$

On the other hand, due to the hypothesis $\mathcal{D}_2(G_n', G_*) > \varepsilon'$ for any $n \in \mathbb{N}$, we also get that $\mathcal{D}_2(G', G_*) > \varepsilon' > 0$, which contradicts the previous result. Hence we reach the claim in equation (32) and totally completes the proof.

## B.4 Proof of Lemma 2

Let $\eta_j = M_j \varepsilon$, where $\varepsilon$ is some fixed positive constant and $M_i$ will be chosen later. As $\mathcal{X}$ and $\Omega$ are bounded sets, we can find some constant $c_\ell^* \geq 0$ such that

$$\min_{x, j, j'} \left[(\beta_{1j}^*)^\top x - (\beta_{1j'}^*)^\top x\right] = c_\ell^* \varepsilon,$$

where the above minimum is subject to $x \in \mathcal{X}_\ell^*, j \in \{\ell_1, \ldots, \ell_K\}$ and $j' \in \{\ell_{K+1}, \ldots, \ell_{k_*}\}$. By arguing similarly to the proof of Lemma 1 in Appendix A.3, we deduce that $c_\ell^* > 0$.

Since $\mathcal{X}$ is a bounded set, we may assume that $\|x\| \leq B$ for any $x \in \mathcal{X}$. Let $x \in \mathcal{X}_{\ell}^*$ and $\overline{\ell} \in [\overline{q}]$ such that $\{\overline{\ell}_1, \ldots, \overline{\ell}_{\overline{K}}\} = \mathcal{C}_{\ell_1} \cup \ldots \cup \mathcal{C}_{\ell_K}$. Then, for any $i \in \{\overline{\ell}_1, \ldots, \overline{\ell}_{\overline{K}}\}$ and $i' \in \{\overline{\ell}_{\overline{K}+1}, \ldots, \overline{\ell}_k\}$, we have that

$$
\begin{aligned}
\beta_{1i}^\top x &= (\beta_{1i} - \beta_{1j}^*)^\top x + (\beta_{1j}^*)^\top x \\
&\geq -M_i \varepsilon B + (\beta_{1j'}^*)^\top x + c_\ell^* \varepsilon \\
&= -M_i \varepsilon B + c_\ell^* \varepsilon + (\beta_{1j'}^* - \beta_{1i'})^\top x + \beta_{1i'}^\top x \\
&\geq -2M_i \varepsilon B + c_\ell^* \varepsilon + \beta_{1i'}^\top x,
\end{aligned}
$$

where $j \in \{\ell_1, \ldots, \ell_K\}$ and $j' \in \{\ell_{K+1}, \ldots, \ell_{k_*}\}$ such that $i \in \mathcal{C}_j$ and $i' \in \mathcal{C}_{j'}$. If $M_j \leq \dfrac{c_\ell^*}{2B}$, then we get that $x \in \mathcal{X}_{\overline{\ell}}$, which leads to $\mathcal{X}_\ell^* \subseteq \overline{\mathcal{X}}_{\overline{\ell}}$.

Analogously, assume that there exists some constant $c_\ell \geq 0$ such that

$$
\min_{x,j,j'} \left[ (\beta_{1j}^*)^\top x - (\beta_{1j'}^*)^\top x \right] = c_\ell^* \varepsilon,
$$

where the minimum is subject to $x \in \overline{\mathcal{X}}_{\overline{\ell}}$, $i \in \{\overline{\ell}_1, \ldots, \overline{\ell}_{\overline{K}}\}$ and $i' \in \{\overline{\ell}_{\overline{K}+1}, \ldots, \overline{\ell}_k\}$. Then, if $M_j \leq \dfrac{c_\ell}{2B}$, then we receive that $\overline{\mathcal{X}}_{\overline{\ell}} \subseteq \mathcal{X}_\ell^*$.

As a consequence, by setting $M_j = \dfrac{1}{2B} \min\{c_\ell^*, c_\ell\}$, we achieve the conclusion that $\overline{\mathcal{X}}_{\overline{\ell}} = \mathcal{X}_\ell^*$.

## C  IDENTIFIABILITY OF THE TOP-K SPARSE SOFTMAX GATING GAUSSIAN MIXTURE OF EXPERTS

In this appendix, we study the identifiability of the top-K sparse softmax gating Gaussian MoE, which plays an essential role in ensuring the convergence of the MLE $\widehat{G}_n$ to the true mixing measure $G_*$ under Voronoi loss functions.

**Proposition 2** (Identifiability). *Let $G$ and $G'$ be two arbitrary mixing measures in $\mathcal{O}_k(\Theta)$. Suppose that the equation $g_G(Y|X) = g_{G'}(Y|X)$ holds for almost surely $(X, Y) \in \mathcal{X} \times \mathcal{Y}$, then it follows that $G \equiv G'$.*

*Proof of Proposition 2.* First, we assume that two mixing measures $G$ and $G'$ take the following forms: $G = \sum_{i=1}^k \exp(\beta_{0i}) \delta_{(\beta_{1i}, a_i, b_i, \sigma_i)}$ and $G' = \sum_{i=1}^{k'} \exp(\beta_{0i}') \delta_{(\beta_{1i}', a_i', b_i', \sigma_i')}$. Recall that $g_G(Y|X) = g_{G'}(Y|X)$ for almost surely $(X, Y)$, then we have

$$
\sum_{i=1}^k \mathrm{Softmax}(\mathrm{TopK}((\beta_{1i})^\top X, K; \beta_{0i})) \cdot f(Y|a_i^\top X + b_i, \sigma_i)
$$

$$
= \sum_{i=1}^{k'} \mathrm{Softmax}(\mathrm{TopK}((\beta_{1i}')^\top X, K; \beta_{0i}')) \cdot f(Y|(a_i')^\top + b_i', \sigma_i'). \tag{33}
$$

Due to the identifiability of the location-scale Gaussian mixtures Teicher (1960; 1961; 1963), we get that $k = k'$ and

$$
\left\{ \mathrm{Softmax}(\mathrm{TopK}((\beta_{1i})^\top X, K; \beta_{0i})) : i \in [k] \right\} \equiv \left\{ \mathrm{Softmax}(\mathrm{TopK}((\beta_{1i}')^\top X, K; \beta_{0i}')) : i \in [k] \right\},
$$

for almost surely $X$. WLOG, we may assume that

$$
\mathrm{Softmax}(\mathrm{TopK}((\beta_{1i})^\top X, K; \beta_{0i})) = \mathrm{Softmax}(\mathrm{TopK}((\beta_{1i}')^\top X, K; \beta_{0i}')), \tag{34}
$$

for almost surely $X$ for any $i \in [k]$. Since the Softmax function is invariant to translations, it follows from equation (34) that $\beta_{1i} = \beta_{1i}' + v_1$ and $\beta_{0i} = \beta_{0i}' + v_0$ for some $v_1 \in \mathbb{R}^d$ and $v_0 \in \mathbb{R}$. Notably, from the assumption of the model, we have $\beta_{1k} = \beta_{1k}' = \mathbf{0}_d$ and $\beta_{0k} = \beta_{0k}' = 0$, which implies that $v_1 = \mathbf{0}_d$ and $v_0 = 0$. As a result, we obtain that $\beta_{1i} = \beta_{1i}'$ and $\beta_{0i} = \beta_{0i}'$ for any $i \in [k]$.

Let us consider $X \in \mathcal{X}_\ell$ where $\ell \in [q]$ such that $\{\ell_1, \dots, \ell_K\} = \{1, \dots, K\}$. Then, equation (33) can be rewritten as

$$\sum_{i=1}^{K} \exp(\beta_{0i}) \exp(\beta_{1i}^\top X) f(Y | a_i^\top X + b_i, \sigma_i) = \sum_{i=1}^{K} \exp(\beta_{0i}) \exp(\beta_{1i}^\top X) f(Y | (a_i')^\top X + b_i', \sigma_i'),$$

(35)

for almost surely $(X, Y)$. Next, we denote $J_1, J_2, \dots, J_m$ as a partition of the index set $[k]$, where $m \leq k$, such that $\exp(\beta_{0i}) = \exp(\beta_{0i'})$ for any $i, i' \in J_j$ and $j \in [m]$. On the other hand, when $i$ and $i'$ do not belong to the same set $J_j$, we let $\exp(\beta_{0i}) \neq \exp(\beta_{0i'})$. Thus, we can reformulate equation (35) as

$$\sum_{j=1}^{m} \sum_{i \in J_j} \exp(\beta_{0i}) \exp(\beta_{1i}^\top X) f(Y | a_i^\top X + b_i, \sigma_i)$$

$$= \sum_{j=1}^{m} \sum_{i \in J_j} \exp(\beta_{0i}) \exp(\beta_{1i}^\top X) f(Y | (a_i')^\top X + b_i', \sigma_i'),$$

for almost surely $(X, Y)$. This results leads to $\{((a_i)^\top X + b_i, \sigma_i) : i \in J_j\} \equiv \{((a_i')^\top X + b_i', \sigma_i') : i \in J_j\}$, for almost surely $X$ for any $j \in [m]$. Therefore, we have

$$\{(a_i, b_i, \sigma_i) : i \in J_j\} \equiv \{(a_i', b_i', \sigma_i') : i \in J_j\},$$

for any $j \in [m]$. As a consequence,

$$G = \sum_{j=1}^{m} \sum_{i \in J_j} \exp(\beta_{0i}) \delta_{(\beta_{1i}, a_i, b_i, \sigma_i)} = \sum_{j=1}^{m} \sum_{i \in J_j} \exp(\beta_{0i}') \delta_{(\beta_{1i}', a_i', b_i', \sigma_i')} = G'.$$

Hence, we reach the conclusion of this proposition. $\square$

## D NUMERICAL EXPERIMENTS

In this appendix, we conduct a few numerical experiments to illustrate the theoretical convergence rates of the MLE $\widehat{G}_n$ to the true mixing measure $G_*$ under both the exact-specified and the over-specified settings.

### D.1 EXPERIMENTAL SETUP

**Synthetic Data.** First, we assume that the true mixing measure $G_* = \sum_{i=1}^{k_*} \exp(\beta_{0i}^*) \delta_{(\beta_{1i}^*, a_i^*, b_i^*, \sigma_i^*)}$ is of order $k_* = 2$ and associated with the following ground-truth parameters:

$$\begin{array}{lllll} \beta_{01}^* = -8, & \beta_{11}^* = 25, & a_1^* = -20, & b_1^* = 15, & \sigma_1^* = 0.3, \\ \beta_{02}^* = 0, & \beta_{12}^* = 0, & a_2^* = 20, & b_2^* = -5, & \sigma_2^* = 0.4. \end{array}$$

Then, we generate i.i.d samples $\{(X_i, Y_i)\}_{i=1}^n$ by first sampling $X_i$'s from the uniform distribution Uniform$[0, 1]$ and then sampling $Y_i$'s from the true conditional density $g_{G_*}(Y|X)$ of top-K sparse softmax gating Gaussian mixture of experts (MoE) given in equation (1). In Figure 2, we visualize the relationship between $X$ and $Y$ when the numbers of experts chosen from $g_{G_*}(Y|X)$ are $K = 1$ (Figure 2a) and $K = 2$ (Figure 2b), respectively. However, throughout the following experiments, we will consider only the scenario when $K = 1$, that is, we choose the best expert from the true conditional density $g_{G_*}(Y|X)$.

**Maximum Likelihood Estimation (MLE).** A popular approach to determining the MLE $\widehat{G}_n$ for each set of samples is to use the EM algorithm Dempster et al. (1977). However, since there are not any closed-form expressions for updating the gating parameters $\beta_{0i}, \beta_{1i}$ in the maximization steps, we have to leverage an EM-based numerical scheme, which was previously used in Chamroukhi et al. (2009). In particular, we utilize a simple coordinate gradient descent algorithm in the maximization steps. Additionally, we select the convergence criterion of $\epsilon = 10^{-6}$ and run a maximum of 2000 EM iterations.

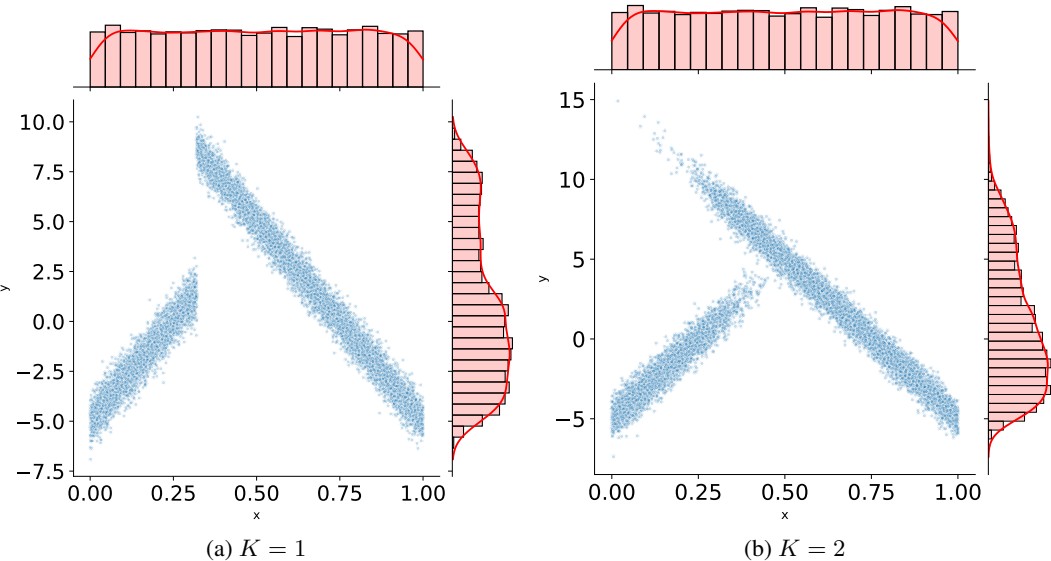

(a) $K = 1$        (b) $K = 2$

Figure 2: A visual representation showcasing the relationship between $X$ and $Y$, along with their respective marginal distributions when $K = 1$ and $K = 2$.

**Initialization.** For each $k \in \{k_*, k_* + 1\}$, we randomly distribute elements of the set $\{1, 2, ..., k\}$ into $k_*$ different Voronoi cells $\mathcal{C}_1, \mathcal{C}_2, \ldots, \mathcal{C}_{k_*}$, each contains at least one element. Moreover, we repeat this process for each replication. Subsequently, for each $j \in [k_*]$, we initialize parameters $\beta_{1i}$ by sampling from a Gaussian distribution centered around its true counterpart $\beta_{1j}^*$ with a small variance, where $i \in \mathcal{C}_j$. Other parameters $\beta_{0i}, a_i, b_i, \sigma_i$ are also initialized in a similar fashion.

### D.2 EXACT-SPECIFIED SETTINGS

Under the exact-specified settings, we conduct 40 sample generations for each configuration, across a spectrum of 200 different sample sizes $n$ ranging from $10^2$ to $10^5$. It can be seen from Figure 3a that the MLE $\widehat{G}_n$ empirically converges to the true mixing measure $G_*$ under the Voronoi metric $\mathcal{D}_1$ at the rate of order $\widetilde{\mathcal{O}}(n^{-1/2})$, which perfectly matches the theoretical parametric convergence rate established in Theorem 2.

### D.3 OVER-SPECIFIED SETTINGS

Under the over-specified settings, we continue to generate 40 samples of size $n$ for each setting, given 100 different choices of sample size $n \in [10^2, 10^5]$. As discussed in Section 3, to guarantee the convergence of density estimation to the true density, we need to select $\overline{K} = 2$ experts from the density estimation. As far as we know, existing works, namely Kwon et al. (2019); Kwon & Caramanis (2020); Kwon et al. (2021), only focus on the global convergence of the EM algorithm for parameter estimation under the input-free gating MoE, while that under the top-K sparse softmax gating MoE has remained poorly understood. Additionally, it is worth noting that the sample size must be sufficiently large so that the empirical convergence rate of the MLE returned by the EM algorithm aligns with the theoretical rate of order $\widetilde{\mathcal{O}}(n^{-1/2})$ derived in Theorem 4.

### E ADDITIONAL RESULTS

In this appendix, we study the convergence rates of parameter estimation under the model (1) when $f$ is a probability density function of an arbitrary location-scale distribution. For that purpose, we first characterize the family of probability density functions of location-scale distributions

$$\mathcal{F} := \{f(Y|h_1(X, a, b), \sigma) : (a, b, \sigma) \in \Theta\}, \tag{36}$$

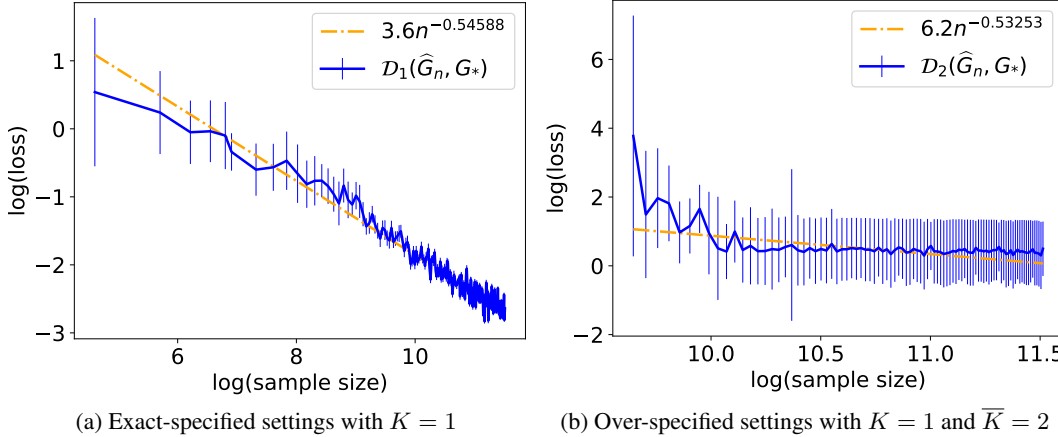

(a) Exact-specified settings with $K = 1$  (b) Over-specified settings with $K = 1$ and $\overline{K} = 2$

Figure 3: Log-log scaled plots illustrating simulation results under the exact-specified and the over-specified settings. We analyze the MLE $\widehat{G}_n$ across 40 independent samples, spanning sample sizes from $10^2$ to $10^5$. The blue curves depict the mean discrepancy between the MLE $\widehat{G}_n$ and the true mixing measure $G_*$, accompanied by error bars signifying two empirical standard deviations under the exact-specified settings. Additionally, an orange dash-dotted line represents the least-squares fitted linear regression line for these data points.

where $h_1(X, a, b) := a^\top X + b$ stands for the location, $\sigma$ denotes the scale and $\Theta$ is a compact subset of $\mathbb{R}^d \times \mathbb{R} \times \mathbb{R}_+$, based on the following notion of strong identifiability, which was previously studied in Manole & Ho (2022) and Ho & Nguyen (2016):

**Definition 1** (Strong Identifiability). *We say that the family $\mathcal{F}$ is strongly identifiable if the probability density function $f(Y|h_1(X, a, b), \sigma)$ is twice differentiable w.r.t its parameters and the following assumption holds true:*

*For any $k \geq 1$ and $k$ pairwise different tuples $(a_1, b_1, \sigma_1), \ldots, (a_k, b_k, \sigma_k) \in \Theta$, if there exist real coefficients $\alpha_{\ell_1, \ell_2}^{(i)}$, for $i \in [k_*]$ and $0 \leq \ell_1 + \ell_2 \leq 2$, such that*

$$\sum_{i=1}^{k} \sum_{\ell_1 + \ell_2 = 0}^{2} \alpha_{\ell_1, \ell_2}^{(i)} \cdot \frac{\partial^{\ell_1 + \ell_2} f}{\partial h_1^{\ell_1} \partial \sigma^{\ell_2}} (Y|h_1(X, a_i, b_i), \sigma(X, \sigma_i)) = 0,$$

*for almost surely $(X, Y)$, then we obtain that $\alpha_{\ell_1, \ell_2}^{(i)} = 0$ for any $i \in [k_*]$ and $0 \leq \ell_1 + \ell_2 \leq 2$.*

**Example 1.** *The families of Student's t-distributions and Laplace distributions are strongly identifiable, while the family of location-scale Gaussian distributions is not.*

In high level, we need to establish the Total Variation lower bound $\mathbb{E}_X[V(g_G(\cdot|X), g_{G_*}(\cdot|X))] \gtrsim \mathcal{D}(G, G_*)$ for any $G \in \mathcal{O}_k(\Omega)$. Then, this bound together with the density estimation rate in Theorem 1 (resp. Theorem 3) leads to the parameter estimation rates in Theorem 2 (resp. Theorem 4). Here, the key step is to decompose the difference $g_{\widehat{G}_n}(Y|X) - g_{G_*}(Y|X)$ into a combination of linearly independent terms using Taylor expansions. Therefore, we have to involve the above notion of strong identifiability, and separate our convergence analysis based on that notion.

Subsequently, since the convergence rates of maximum likelihood estimation when $\mathcal{F}$ is a family of location-scale Gaussian distributions, which is not strongly identifiable, have already been studied in Section 2 and Section 3, we will focus on the scenario when the family $\mathcal{F}$ is strongly identifiable in the sequel. Under that assumption, the density $f(Y|h_1, \sigma)$ is twice differentiable in $(h_1, \sigma)$, therefore, it is also Lipschitz continuous. As a consequence, the density estimation rates under both the exact-specified and over-specified in Theorem 1 and Theorem 3 still hold true. Therefore, we aim to establish the parameter estimation rates under those settings in Appendix E.1 and Appendix E.2, respectively.

### E.1 EXACT-SPECIFIED SETTINGS

In this appendix, by using the Voronoi loss function $\mathcal{D}_1(G, G_*)$ defined in equation (5), we demonstrate in the following theorem that the rates for estimating true parameters $\exp(\beta_{0i}^*), \beta_{1i}^*, a_i^*, b_i^*, \sigma_i^*$ are of parametric order $\widetilde{\mathcal{O}}(n^{-1/2})$, which totally match those in Theorem 2.

**Theorem 5.** *Under the exact-specified settings, if the family $\mathcal{F}$ is strongly identifiable, then the Hellinger lower bound $\mathbb{E}_X[h(g_G(\cdot|X), g_{G_*}(\cdot|X))] \gtrsim \mathcal{D}_1(G, G_*)$ holds for any mixing measure $G \in \mathcal{E}_{k_*}(\Omega)$. Consequently, we can find a universal constant $C_3 > 0$ depending only on $G_*$, $\Omega$ and $K$ such that*

$$\mathbb{P}\Big(\mathcal{D}_1(\widehat{G}_n, G_*) > C_3\sqrt{\log(n)/n}\Big) \lesssim n^{-c_3},$$

*where $c_3 > 0$ is a universal constant that depends only on $\Omega$.*

Proof of Theorem 5 is in Appendix E.3.1.

### E.2 OVER-SPECIFIED SETTINGS

In this appendix, we capture the convergence rates of parameter estimation under the over-specified settings when the family $\mathcal{F}$ is strongly identifiable.

**Voronoi metric.** It is worth noting that when $\mathcal{F}$ is strongly identifiable, the interaction among expert parameters in the second PDE (6) no longer holds true. As a result, it is not necessary to involve the solvability of the system (7) in the formulation of the Voronoi loss as in equation (8). Instead, let us consider the Voronoi loss function $\mathcal{D}_3(G, G_*)$ defined as

$$\mathcal{D}_3(G, G_*) := \max_{\{\ell_j\}_{j=1}^K \subset [k_*]} \Bigg\{ \sum_{\substack{j \in [K], \; i \in \mathcal{C}_{\ell_j} \\ |\mathcal{C}_{\ell_j}|=1}} \exp(\beta_{0i})\Big[\|\Delta\beta_{1i\ell_j}\| + \|\Delta a_{i\ell_j}\| + |\Delta b_{i\ell_j}| + |\Delta\sigma_{i\ell_j}|\Big]$$
$$+ \sum_{\substack{j \in [K], \; i \in \mathcal{C}_{\ell_j} \\ |\mathcal{C}_{\ell_j}|>1}} \exp(\beta_{0i})\Big[\|\Delta\beta_{1i\ell_j}\|^2 + \|\Delta a_{i\ell_j}\|^2 + |\Delta b_{i\ell_j}|^2 + |\Delta\sigma_{i\ell_j}|^2\Big]$$
$$+ \sum_{j=1}^K \Big|\sum_{i \in \mathcal{C}_{\ell_j}} \exp(\beta_{0i}) - \exp(\beta_{0\ell_j}^*)\Big| \Bigg\}, \quad (37)$$

for any mixing measure $G \in \mathcal{O}_k(\Omega)$. Equipped with loss function, we are ready to illustrate the convergence behavior of maximum likelihood estimation in the following theorem:

**Theorem 6.** *Under the over-specified settings, if the family $\mathcal{F}$ is strongly identifiable, then the Hellinger lower bound $\mathbb{E}_X[h(g_G(\cdot|X), g_{G_*}(\cdot|X))] \gtrsim \mathcal{D}_2(G, G_*)$ holds for any mixing measure $G \in \mathcal{O}_k(\Omega)$. As a consequence, we can find a universal constant $C_4 > 0$ depending only on $G_*$, $\Omega$ and $K$ such that*

$$\mathbb{P}\Big(\mathcal{D}_3(\widehat{G}_n, G_*) > C_4\sqrt{\log(n)/n}\Big) \lesssim n^{-c_4},$$

*where $c_4 > 0$ is a universal constant that depends only on $\Omega$.*

Proof of Theorem 6 is in Appendix E.3.2. Theorem 6 indicates that true parameters $\beta_{1i}^*, a_i^*, b_i^*, \sigma_i^*$, which are fitted by a single component, share the same estimation rates of order $\widetilde{\mathcal{O}}(n^{-1/2})$ as those in Theorem 4. By contrast, the estimation rates for true parameters $\beta_{1i}^*, a_i^*, b_i^*, \sigma_i^*$, which are fitted by more than one component, are of order $\widetilde{\mathcal{O}}(n^{-1/4})$. Notably, these rates are no longer determined by the solvability of the system (7). Thus, they are significantly faster than those in Theorem 4. The main reason for this improvement is when $\mathcal{F}$ is strongly identifiable, the interaction among expert parameters via the second PDE in equation (6) does not occur.

### E.3   PROOFS OF ADDITIONAL RESULTS

#### E.3.1   PROOF OF THEOREM 5

Following from the result of Theorem 1 and since the Hellinger distance is lower bounded by the Total Variation distance, i.e. $h \geq V$, it is sufficient to demonstrate for any mixing measure $G \in \mathcal{O}_k(\Theta)$ that

$$\mathbb{E}_X[V(g_G(\cdot|X), g_{G_*}(\cdot|X))] \gtrsim \mathcal{D}_1(G, G_*),$$

which is then respectively broken into local part and global part as follows:

$$\inf_{G \in \mathcal{E}_{k_*}(\Omega) : \mathcal{D}_1(G, G_*) \leq \varepsilon'} \frac{\mathbb{E}_X[V(g_G(\cdot|X), g_{G_*}(\cdot|X))]}{\mathcal{D}_1(G, G_*)} > 0, \tag{38}$$

$$\inf_{G \in \mathcal{E}_{k_*}(\Omega) : \mathcal{D}_1(G, G_*) > \varepsilon'} \frac{\mathbb{E}_X[V(g_G(\cdot|X), g_{G_*}(\cdot|X))]}{\mathcal{D}_1(G, G_*)} > 0, \tag{39}$$

for some constant $\varepsilon' > 0$. Subsequently, we will prove only the local part (38), while the proof of the global part (39) can be done similarly to that in Appendix A.2.

**Proof of claim** (38): It is sufficient to show that

$$\lim_{\varepsilon \to 0} \inf_{G \in \mathcal{E}_{k_*}(\Omega) : \mathcal{D}_1(G, G_*) \leq \varepsilon} \frac{\mathbb{E}_X[V(g_G(\cdot|X), g_{G_*}(\cdot|X))]}{\mathcal{D}_1(G, G_*)} > 0.$$

Assume that this inequality does not hold, then since the number of experts $k_*$ is known in this case, there exists a sequence of mixing measure $G_n := \sum_{i=1}^{k_*} \exp(\beta_{0i}^n) \delta_{(\beta_{1i}^n, a_i^n, b_i^n, \sigma_i^n)} \in \mathcal{E}_{k_*}(\Omega)$ such that both $\mathcal{D}_1(G_n, G_*)$ and $\mathbb{E}_X[V(g_{G_n}(\cdot|X), g_{G_*}(\cdot|X))]/\mathcal{D}_1(G_n, G_*)$ approach zero as $n$ tends to infinity. Since $\mathcal{D}_1(G_n, G_*) \to 0$ as $n \to \infty$, each Voronoi cell contains only one element. Thus, we may assume WLOG that $\mathcal{C}_j = \{j\}$ for any $j \in [k_*]$, which implies that $(\beta_{1j}^n, a_j^n, b_j^n, \sigma_j^n) \to (\beta_{1j}^*, a_j^*, b_j^*, \sigma_j^*)$ and $\exp(\beta_{0j}^n) \to \exp(\beta_{0j}^*)$ as $n \to \infty$. WLOG, we assume that

$$\mathcal{D}_1(G_n, G_*) = \sum_{i=1}^K \left[ \exp(\beta_{0i}^n)\left( \|\Delta\beta_{1i}^n\| + \|\Delta a_i^n\| + \|\Delta b_i^n\| + \|\Delta\sigma_i^n\| \right) + \left| \exp(\beta_{0i}^n) - \exp(\beta_{0i}^*) \right| \right],$$

where we denote $\Delta\beta_{1i}^n := \beta_{1i}^n - \beta_{1i}^*$, $\Delta a_i^n := a_i^n - a_i^*$, $\Delta b_i^n := b_i^n - b_i^*$ and $\Delta\sigma_i^n := \sigma_i^n - \sigma_i^*$.

Subsequently, by arguing in the same fashion as in Appendix A.2, we obtain that $\mathcal{X}_\ell^n = \mathcal{X}_\ell^*$, where

$$\mathcal{X}_\ell^n := \left\{ x \in \mathcal{X} : (\beta_{1j}^n)^\top x \geq (\beta_{1j'}^n)^\top x : \forall j \in \{\ell_1, \dots, \ell_K\}, j' \in \{\ell_{K+1}, \dots, \ell_{k_*}\} \right\},$$

$$\mathcal{X}_\ell^* := \left\{ x \in \mathcal{X} : (\beta_{1j}^*)^\top x \geq (\beta_{1j'}^*)^\top x : \forall j \in \{\ell_1, \dots, \ell_K\}, j' \in \{\ell_{K+1}, \dots, \ell_{k_*}\} \right\},$$

for any $\ell \in [q]$ for sufficiently large $n$.

Let $\ell \in [q]$ such that $\{\ell_1, \dots, \ell_K\} = \{1, \dots, K\}$. Then, for almost surely $(X, Y) \in \mathcal{X}_\ell^* \times \mathcal{Y}$, we can rewrite the conditional densities $g_{G_n}(Y|X)$ and $g_{G_*}(Y|X)$ as

$$g_{G_n}(Y|X) = \sum_{i=1}^K \frac{\exp((\beta_{1i}^n)^\top X + \beta_{0i}^n)}{\sum_{j=1}^K \exp((\beta_{1j}^n)^\top X + \beta_{0j}^n)} \cdot f(Y|(a_i^n)^\top X + b_i^n, \sigma_i^n),$$

$$g_{G_*}(Y|X) = \sum_{i=1}^K \frac{\exp((\beta_{1i}^*)^\top X + \beta_{0i}^*)}{\sum_{j=1}^K \exp((\beta_{1j}^*)^\top X + \beta_{0j}^*)} \cdot f(Y|(a_i^*)^\top X + b_i^*, \sigma_i^*).$$

Now, we break the rest of our arguments into three steps:

**Step 1 - Taylor expansion**:

In this step, we take into account $H_n := \left[ \sum_{i=1}^{K} \exp((\beta_{1i}^*)^\top X + \beta_{0i}^*) \right] \cdot [g_{G_n}(Y|X) - g_{G_*}(Y|X)]$. Note that the quantity $H_n$ can be represented as follows:

$$H_n = \sum_{i=1}^{K} \exp(\beta_{0i}^n) \left[ \exp((\beta_{1i}^n)^\top X) f(Y|(a_i^n)^\top X + b_i^n, \sigma_i^n) - \exp((\beta_{1i}^n)^\top X) f(Y|(a_i^*)^\top X + b_i^*, \sigma_i^*) \right]$$

$$- \sum_{i=1}^{K} \exp(\beta_{0i}^n) \left[ \exp((\beta_{1i}^n)^\top X) g_{G_n}(Y|X) - \exp((\beta_{1i}^*)^\top X) g_{G_n}(Y|X) \right]$$

$$+ \sum_{i=1}^{K} \left[ \exp(\beta_{0i}^n) - \exp(\beta_{0i}^*) \right] \left[ \exp((\beta_{1i}^*)^\top X + \beta_{0i}^*) f(Y|(a_i^*)^\top X + b_i^*, \sigma_i^*) - \exp((\beta_{1i}^*)^\top X) g_{G_n}(Y|X) \right].$$

By applying the first-order Taylor expansion to the first term in the above representation, which is denoted by $A_n$, we get that

$$A_n = \sum_{i=1}^{K} \sum_{|\alpha|=1} \frac{\exp(\beta_{0i}^n)}{\alpha!} \cdot (\Delta\beta_{1i}^n)^{\alpha_1} (\Delta a_i^n)^{\alpha_2} (\Delta b_i^n)^{\alpha_3} (\Delta\sigma_i^n)^{\alpha_4}$$

$$\times X^{\alpha_1+\alpha_2} \exp((\beta_{1i}^*)^\top X) \cdot \frac{\partial^{|\alpha_2|+\alpha_3+\alpha_4} f}{\partial h_1^{|\alpha_2|+\alpha_3} \partial\sigma^{\alpha_4}} (Y|(a_i^*)^\top X + b_i^*, \sigma_i^*) + R_1(X,Y),$$

where $R_1(X,Y)$ is a Taylor remainder that satisfies $R_1(X,Y)/\mathcal{D}_1(X,Y) \to 0$ as $n \to \infty$. Let $\eta_1 = \alpha_1 + \alpha_2 \in \mathbb{N}^d$, $\eta_2 = |\alpha_2| + \alpha_3 \in \mathbb{N}$ and $\eta_3 = \alpha_4 \in \mathbb{N}$, then we can rewrite $A_n$ as follows:

$$A_n = \sum_{i=1}^{K} \sum_{\eta_3=0}^{1} \sum_{|\eta_1|+\eta_2=1-\eta_3}^{2-\eta_3} \sum_{\alpha\in\mathcal{I}_{\eta_1,\eta_2,\eta_3}} \frac{\exp(\beta_{0i}^n)}{\alpha!} \cdot (\Delta\beta_{1i}^n)^{\alpha_1} (\Delta a_i^n)^{\alpha_2} (\Delta b_i^n)^{\alpha_3} (\Delta\sigma_i^n)^{\alpha_4}$$

$$\times X^{\eta_1} \exp((\beta_{1i}^*)^\top X) \cdot \frac{\partial^{\eta_2+\eta_3} f}{\partial h_1^{\eta_2} \partial\sigma^{\eta_3}} (Y|(a_i^*)^\top X + b_i^*, \sigma_i^*) + R_1(X,Y), \quad (40)$$

where we define

$$\mathcal{I}_{\eta_1,\eta_2,\eta_3} := \{ (\alpha_i)_{i=1}^{4} \in \mathbb{N}^d \times \mathbb{N}^d \times \mathbb{N} \times \mathbb{N} : \alpha_1 + \alpha_2 = \eta_1, |\alpha_2| + \alpha_3 = \eta_2, \alpha_4 = \eta_3 \}. \quad (41)$$

By arguing in a similar fashion for the second term in the representation of $H_n$, we also get that

$$B_n := - \sum_{i=1}^{K} \sum_{|\gamma|=1} \frac{\exp(\beta_{0i}^n)}{\gamma!} (\Delta\beta_{1i}^n) \cdot X^\gamma \exp((\beta_{1i}^n)^\top X) g_{G_n}(Y|X) + R_2(X,Y),$$

where $R_2(X,Y)$ is a Taylor remainder such that $R_2(X,Y)/\mathcal{D}_1(G_n, G_*) \to 0$ as $n \to \infty$. Putting the above results together, we rewrite the quantity $H_n$ as follows:

$$H_n = \sum_{i=1}^{K} \sum_{\eta_3=0}^{1} \sum_{|\eta_1|+\eta_2=0}^{2-\eta_3} U_{i,\eta_1,\eta_2,\eta_3}^n \cdot X^{\eta_1} \exp((\beta_{1i}^*)^\top X) \frac{\partial^{\eta_2+\eta_3} f}{\partial h_1^{\eta_2} \partial\sigma^{\eta_3}} (Y|(a_i^*)^\top X + b_i^*, \sigma_i^*)$$

$$+ \sum_{i=1}^{K} \sum_{0\leq|\gamma|\leq1} W_{i,\gamma}^n \cdot X^\gamma \exp((\beta_{1i}^*)^\top X) g_{G_n}(Y|X) + R_1(X,Y) + R_2(X,Y), \quad (42)$$

in which we respectively define for each $i \in [K]$ that

$$U_{i,\eta_1,\eta_2,\eta_3}^n := \sum_{\alpha\in\mathcal{I}_{\eta_1,\eta_2,\eta_3}} \frac{\exp(\beta_{0i}^n)}{\alpha!} \cdot (\Delta\beta_{1i}^n)^{\alpha_1} (\Delta a_i^n)^{\alpha_2} (\Delta b_i^n)^{\alpha_3} (\Delta\sigma_i^n)^{\alpha_4},$$

$$W_{i,\gamma}^n := -\frac{\exp(\beta_{0i}^n)}{\gamma!} (\Delta\beta_{1i}^n)^\gamma,$$

for any $(\eta_1, \eta_2, \eta_3) \neq (\mathbf{0}_d, 0, 0)$ and $|\gamma| \neq \mathbf{0}_d$. Additionally, $U_{i,\mathbf{0}_d,0,0}^n = -W_{i,\mathbf{0}_d}^n := \exp(\beta_{0i}^n) - \exp(\beta_{0i}^*)$.

**Step 2 - Non-vanishing coefficients**:

Moving to the second step, we will show that not all the ratios $U^n_{i,\eta_1,\eta_2,\eta_3}/\mathcal{D}_1(G_n, G_*)$ tend to zero as $n$ goes to infinity. Assume by contrary that all of them approach zero when $n \to \infty$, then for $(\eta_1, \eta_2, \eta_3) = (\mathbf{0}_d, 0, 0)$, it follows that

$$\frac{1}{\mathcal{D}_1(G_n, G_*)} \cdot \sum_{i=1}^{K} \left| \exp(\beta^n_{0i}) - \exp(\beta^*_{0i}) \right| = \sum_{i=1}^{K} \frac{|U^n_{j,\eta_1,\eta_2,\eta_3}|}{\mathcal{D}_1(G_n, G_*)} \to 0. \tag{43}$$

Additionally, for tuples $(\eta_1, \eta_2, \eta_3)$ where $\eta_1 \in \{e_1, e_2, \ldots, e_d\}$ with $e_j := (0, \ldots, 0, \underbrace{1}_{j-th}, 0, \ldots, 0)$

and $\eta_2 = \eta_3 = 0$, we get

$$\frac{1}{\mathcal{D}_1(G_n, G_*)} \cdot \sum_{i=1}^{K} \exp(\beta^n_{0i}) \|\Delta\beta^n_{1i}\|_1 = \sum_{i=1}^{K} \sum_{\eta_1 \in \{e_1, \ldots, e_d\}} \frac{|U^n_{j,\eta_1,0,0}|}{\mathcal{D}_1(G_n, G_*)} \to 0.$$

By using similar arguments, we end up with

$$\frac{1}{\mathcal{D}_1(G_n, G_*)} \cdot \sum_{i=1}^{K} \exp(\beta^n_{0i}) \Big[ \|\Delta\beta^n_{1i}\|_1 + \|\Delta a^n_i\|_1 + |\Delta b^n_i| + |\Delta \sigma^n_i| \Big] \to 0.$$

Due to the topological equivalence between norm-1 and norm-2, the above limit implies that

$$\frac{1}{\mathcal{D}_1(G_n, G_*)} \cdot \sum_{i=1}^{K} \exp(\beta^n_{0i}) \Big[ \|\Delta\beta^n_{1i}\| + \|\Delta a^n_i\| + |\Delta b^n_i| + |\Delta \sigma^n_i| \Big] \to 0. \tag{44}$$

Combine equation (43) with equation (44), we deduce that $\mathcal{D}_1(G_n, G_*)/\mathcal{D}_1(G_n, G_*) \to 0$, which is a contradiction. Consequently, at least one among the ratios $U^n_{i,\eta_1,\eta_2,\eta_3}/\mathcal{D}_1(G_n, G_*)$ does not vanish as $n$ tends to infinity.

**Step 3 - Fatou's contradiction**:

Let us denote by $m_n$ the maximum of the absolute values of $U^n_{i,\eta_1,\eta_2,\eta_3}/\mathcal{D}_1(G_n, G_*)$ and $W^n_{i,\gamma}/\mathcal{D}_1(G_n, G_*)$. It follows from the result achieved in Step 2 that $1/m_n \not\to \infty$.

Recall from the hypothesis that $\mathbb{E}_X[V(g_{G_n}(\cdot|X), g_{G_*}(\cdot|X))]/\mathcal{D}_1(G_n, G_*) \to 0$ as $n \to \infty$. Thus, by the Fatou's lemma, we have

$$0 = \lim_{n \to \infty} \frac{\mathbb{E}_X[V(g_{G_n}(\cdot|X), g_{G_*}(\cdot|X))]}{\mathcal{D}_1(G_n, G_*)} = \frac{1}{2} \cdot \int \liminf_{n \to \infty} \frac{|g_{G_n}(Y|X) - g_{G_*}(Y|X)|}{\mathcal{D}_1(G_n, G_*)} \mathrm{d}X \mathrm{d}Y.$$

This result indicates that $|g_{G_n}(Y|X) - g_{G_*}(Y|X)|/\mathcal{D}_1(G_n, G_*)$ tends to zero as $n$ goes to infinity for almost surely $(X, Y)$. As a result, it follows that

$$\lim_{n \to \infty} \frac{H_n}{m_n \mathcal{D}(G_n, G_*)} = \Big[ \sum_{i=1}^{K} \exp((\beta^*_{1i})^\top X + \beta^*_{0i}) \Big] \cdot \lim_{n \to \infty} \frac{|g_{G_n}(Y|X) - g_{G_*}(Y|X)|}{m_n \mathcal{D}_1(G_n, G_*)} = 0.$$

Next, let us denote $U^n_{i,\eta_1,\eta_2,\eta_3}/[m_n \mathcal{D}_1(G_n, G_*)] \to \tau_{i,\eta_1,\eta_2,\eta_3}$ and $W^n_{i,\gamma}/[m_n \mathcal{D}_1(G_n, G_*)] \to \kappa_{i,\gamma}$ with a note that at least one among them is non-zero. From the formulation of $H_n$ in equation (42), we deduce that

$$\sum_{i=1}^{K} \sum_{\eta_3=0}^{1} \sum_{|\eta_1|+\eta_2=0}^{2-\eta_3} \tau_{i,\eta_1,\eta_2,\eta_3} \cdot X^{\eta_1} \exp((\beta^*_{1i})^\top X) \frac{\partial^{\eta_2+\eta_3} f}{\partial h_1^{\eta_2} \partial \sigma^{\eta_3}}(Y|(a^*_i)^\top X + b^*_i, \sigma^*_i)$$

$$+ \sum_{i=1}^{K} \sum_{0 \le |\gamma| \le 1} \kappa_{i,\gamma} \cdot X^\gamma \exp((\beta^*_{1i})^\top X) g_{G_*}(Y|X) = 0,$$

for almost surely $(X, Y)$. This equation is equivalent to

$$\sum_{|\eta_1|=0}^{1} \Big[ \sum_{i=1}^{K} \sum_{\eta_2+\eta_3=0}^{2-|\eta_1|} \tau_{i,\eta_1,\eta_2,\eta_3} \exp((\beta^*_{1i})^\top X) \frac{\partial^{\eta_2+\eta_3} f}{\partial h_1^{\eta_2} \partial \sigma^{\eta_3}}(Y|(a^*_i)^\top X + b^*_i, \sigma^*_i)$$

$$+ \kappa_{i,\eta_1} \exp((\beta^*_{1i})^\top X) g_{G_*}(Y|X) \Big] \times X^{\eta_1} = 0,$$

for almost surely $(X, Y)$. It is clear that the left hand side of the above equation is a polynomial of $X$ belonging to the compact set $\mathcal{X}$. As a result, we get that

$$
\sum_{i=1}^{K} \sum_{\eta_2+\eta_3=0}^{2-|\eta_1|} \tau_{i,\eta_1,\eta_2,\eta_3} \exp((\beta_{1i}^*)^\top X) \frac{\partial^{\eta_2+\eta_3} f}{\partial h_1^{\eta_2} \partial \sigma^{\eta_3}}(Y|(a_i^*)^\top X + b_i^*, \sigma_i^*)
$$
$$
+ \kappa_{i,\eta_1} g_{G_*}(Y|X) \exp((\beta_{1i}^*)^\top X) = 0,
$$

for any $i \in [K]$, $0 \leq |\eta_1| \leq 1$ and almost surely $(X, Y)$. Since $(a_1^*, b_1^*, \sigma_1^*), \ldots, (a_K^*, b_K^*, \sigma_K^*)$ have pair-wise distinct values and the family $\mathcal{F}$ is strongly identifiable, the set

$$
\left\{ \frac{\partial^{\eta_2+\eta_3} f}{\partial h_1^{\eta_2} \partial \sigma^{\eta_3}}(Y|(a_i^*)^\top X + b_i^*, \sigma_i^*) : i \in [K],\ 0 \leq \eta_2 + \eta_3 \leq 2 - |\eta_1| \right\}
$$

is linearly independent w.r.t $(X, Y)$. Consequently, we obtain that $\tau_{i,\eta_1,\eta_2,\eta_3} = \kappa_{i,\eta_1} = 0$ for any $i \in [K]$, $0 \leq |\eta_1| \leq 1$ and $0 \leq \eta_2 + \eta_3 \leq 2 - |\eta_1|$, which contradicts the fact that at least one among these terms is different from zero.

Hence, we reach the desired conclusion.

### E.3.2 PROOF OF THEOREM 6

Similar to the proof of Theorem 5 in Appendix E.3.1, our objective here is also to derive the local part of the following Total Variation lower bound:

$$
\mathbb{E}_X[V(\overline{g}_G(\cdot|X), g_{G_*}(\cdot|X))] \gtrsim \mathcal{D}_3(G, G_*),
$$

for any $G \in \mathcal{O}_k(\Theta)$. In particular, we aim to show that

$$
\lim_{\varepsilon \to 0} \inf_{G \in \mathcal{O}_k(\Theta):\mathcal{D}_3(G,G_*) \leq \varepsilon} \frac{\mathbb{E}_X[V(\overline{g}_G(\cdot|X), g_{G_*}(\cdot|X))]}{\mathcal{D}_2(G, G_*)} > 0. \tag{45}
$$

Assume that the above claim does not hold true, then we can find a sequence of mixing measures $G_n := \sum_{i=1}^{k_n} \exp(\beta_{0i}^n) \delta_{(\beta_{1i}^n, a_i^n, b_i^n, \sigma_i^n)} \in \mathcal{O}_k(\Omega)$ such that both $\mathcal{D}_3(G_n, G_*)$ and $\mathbb{E}_X[V(\overline{g}_{G_n}(\cdot|X), g_{G_*}(\cdot|X))]/\mathcal{D}_3(G_n, G_*)$ vanish when $n$ goes to infinity. Then, it follows that for any $j \in [k_*]$, we have $\sum_{i \in \mathcal{C}_j} \exp(\beta_{0i}^n) \to \exp(\beta_{0j}^*)$ and $(\beta_{1i}^n, a_i^n, b_i^n, \sigma_i^n) \to (\beta_{1j}^*, a_j^*, b_j^*, \sigma_j^*)$ for all $i \in \mathcal{C}_j$. WLOG, we may assume that

$$
\mathcal{D}_3(G_n, G_*) = \sum_{\substack{j \in [K],\ i \in \mathcal{C}_j \\ |\mathcal{C}_j| > 1}} \exp(\beta_{0i}^n) \left[ \|\Delta\beta_{1ij}^n\|^2 + \|\Delta a_{ij}^n\|^2 + |\Delta b_{ij}^n|^2 + |\Delta\sigma_{ij}^n|^2 \right]
$$

$$
+ \sum_{\substack{j \in [K],\ i \in \mathcal{C}_j \\ |\mathcal{C}_j| = 1}} \exp(\beta_{0i}^n) \left[ \|\Delta\beta_{1ij}^n\| + \|\Delta a_{ij}^n\| + |\Delta b_{ij}^n| + |\Delta\sigma_{ij}^n| \right] + \sum_{j=1}^{K} \left| \sum_{i \in \mathcal{C}_j} \exp(\beta_{0i}^n) - \exp(\beta_{0j}^*) \right|.
$$

Subsequently, let $X \in \mathcal{X}_\ell^*$ for some $\ell \in [q]$ such that $\{\ell_1, \ldots, \ell_K\} = \{1, \ldots, K\}$, where

$$
\mathcal{X}_\ell^* := \left\{ x \in \mathcal{X} : (\beta_{1j}^*)^\top x \geq (\beta_{1j'}^*)^\top x, \forall j \in \{\ell_1, \ldots, \ell_K\}, j' \in \{\ell_{K+1}, \ldots, \ell_{k_*}\} \right\}.
$$

Then, for any $\overline{\ell} \in [\overline{q}]$, we denote $(\overline{\ell}_1, \ldots, \overline{\ell}_k)$ as a permutation of $(1, \ldots, k)$ and

$$
\mathcal{X}_{\overline{\ell}}^n := \left\{ x \in \mathcal{X} : (\beta_{1i}^n)^\top x \geq (\beta_{1i'}^n)^\top x, \forall i \in \{\overline{\ell}_1, \ldots, \overline{\ell}_{\overline{K}}\}, i' \in \{\overline{\ell}_{\overline{K}+1}, \ldots, \overline{\ell}_k\} \right\}.
$$

If $\{\overline{\ell}_1, \ldots \overline{\ell}_{\overline{K}}\} \neq \mathcal{C}_1 \cup \ldots \cup \mathcal{C}_K$ for any $\overline{\ell} \in [\overline{q}]$, then $V(\overline{g}_{G_n}(\cdot|X), g_{G_*}(\cdot|X))/\mathcal{D}_3(G_n, G_*) \not\to 0$ as $n$ tends to infinity. This contradicts the fact that this term must approach zero. Therefore, we only need to consider the scenario when there exists $\overline{\ell} \in [\overline{q}]$ such that $\{\overline{\ell}_1, \ldots \overline{\ell}_{\overline{K}}\} = \mathcal{C}_1 \cup \ldots \cup \mathcal{C}_K$. By using the same arguments as in Appendix B.2, we obtain that $X \in \mathcal{X}_{\overline{\ell}}^n$.

Then, we can represent the conditional densities $g_{G_*}(Y|X)$ and $g_{G_n}(Y|X)$ for any sufficiently large $n$ as follows:

$$g_{G_*}(Y|X) = \sum_{j=1}^{K} \frac{\exp((\beta_{1j}^*)^\top X + \beta_{0j}^*)}{\sum_{j'=1}^{K} \exp((\beta_{1j'}^*)^\top X + \beta_{0j'}^*)} \cdot f(Y|(a_j^*)^\top X + b_j^*, \sigma_j^*),$$

$$\overline{g}_{G_n}(Y|X) = \sum_{j=1}^{K} \sum_{i \in \mathcal{C}_j} \frac{\exp((\beta_{1i}^n)^\top X + \beta_{0i}^n)}{\sum_{j'=1}^{K} \sum_{i' \in \mathcal{C}_{j'}} \exp((\beta_{1i'}^n)^\top X + \beta_{0i'}^n)} \cdot f(Y|(a_i^n)^\top X + b_i^n, \sigma_i^n).$$

Now, we reuse the three-step framework in Appendix E.3.1.

**Step 1 - Taylor expansion**:

Firstly, by abuse of notations, let us consider the quantity

$$H_n := \Big[ \sum_{j=1}^{K} \exp((\beta_{1j}^*)^\top X + \beta_{0j}^*) \Big] \cdot [\overline{g}_{G_n}(Y|X) - g_{G_*}(Y|X)].$$

Similar to Step 1 in Appendix A, we can express this term as

$$H_n = \sum_{j=1}^{K} \sum_{i \in \mathcal{C}_j} \exp(\beta_{0i}^n) \Big[ \exp((\beta_{1i}^n)^\top X) f(Y|(a_i^n)^\top X + b_i^n, \sigma_i^n) - \exp((\beta_{1j}^*)^\top X) f(Y|(a_j^*)^\top X + b_j^*, \sigma_j^*) \Big]$$

$$- \sum_{j=1}^{K} \sum_{i \in \mathcal{C}_j} \exp(\beta_{0i}^n) \Big[ \exp((\beta_{1i}^n)^\top X) g_{G_n}(Y|X) - \exp((\beta_{1j}^*)^\top X) g_{G_n}(Y|X) \Big]$$

$$+ \sum_{j=1}^{K} \Big[ \sum_{i \in \mathcal{C}_j} \exp(\beta_{0i}^n) - \exp(\beta_{0j}^*) \Big] \Big[ \exp((\beta_{1j}^*)^\top X) f(Y|(a_i^*)^\top X + b_i^*, \sigma_i^*) - \exp((\beta_{1j}^*)^\top X) g_{G_n}(Y|X) \Big]$$

$$:= A_n + B_n + E_n.$$

Next, we proceed to decompose $A_n$ based on the cardinality of the Voronoi cells as follows:

$$A_n = \sum_{j:|\mathcal{C}_j|=1} \sum_{i \in \mathcal{C}_j} \exp(\beta_{0i}^n) \Big[ \exp((\beta_{1i}^n)^\top X) f(Y|(a_i^n)^\top X + b_i^n, \sigma_i^n) - \exp((\beta_{1i}^*)^\top X) f(Y|(a_i^*)^\top X + b_i^*, \sigma_i^*) \Big]$$

$$+ \sum_{j:|\mathcal{C}_j|>1} \sum_{i \in \mathcal{C}_j} \exp(\beta_{0i}^n) \Big[ \exp((\beta_{1i}^n)^\top X) f(Y|(a_i^n)^\top X + b_i^n, \sigma_i^n) - \exp((\beta_{1i}^*)^\top X) f(Y|(a_i^*)^\top X + b_i^*, \sigma_i^*) \Big].$$

By applying the Taylor expansions of first and second orders to the first and second terms of $A_n$, respectively, and following the derivation in equation (40), we arrive at

$$A_n = \sum_{j:|\mathcal{C}_j|=1} \sum_{i \in \mathcal{C}_j} \sum_{\eta_3=0}^{1} \sum_{|\eta_1|+\eta_2=1-\eta_3}^{2-\eta_3} \sum_{\alpha \in \mathcal{I}_{\eta_1,\eta_2,\eta_3}} \frac{\exp(\beta_{0i}^n)}{\alpha!} \cdot (\Delta\beta_{1ij}^n)^{\alpha_1} (\Delta a_{ij}^n)^{\alpha_2} (\Delta b_{ij}^n)^{\alpha_3} (\Delta\sigma_i^n)^{\alpha_4}$$

$$\times X^{\eta_1} \exp((\beta_{1j}^*)^\top X) \cdot \frac{\partial^{\eta_2+\eta_3} f}{\partial h_1^{\eta_2} \partial\sigma^{\eta_3}} (Y|(a_j^*)^\top X + b_j^*, \sigma_j^*) + R_3(X,Y)$$

$$+ \sum_{j:|\mathcal{C}_j|>1} \sum_{i \in \mathcal{C}_j} \sum_{\eta_3=0}^{2} \sum_{|\eta_1|+\eta_2=1-\mathbf{1}_{\{\eta_3>0\}}}^{4-\eta_3} \sum_{\alpha \in \mathcal{I}_{\eta_1,\eta_2,\eta_3}} \frac{\exp(\beta_{0i}^n)}{\alpha!} \cdot (\Delta\beta_{1ij}^n)^{\alpha_1} (\Delta a_{ij}^n)^{\alpha_2} (\Delta b_{ij}^n)^{\alpha_3} (\Delta\sigma_{ij}^n)^{\alpha_4}$$

$$\times X^{\eta_1} \exp((\beta_{1j}^*)^\top X) \cdot \frac{\partial^{\eta_2+\eta_3} f}{\partial h_1^{\eta_2} \partial\sigma^{\eta_3}} (Y|(a_j^*)^\top X + b_j^*, \sigma_j^*) + R_4(X,Y),$$

where the set $\mathcal{I}_{\eta_1,\eta_2,\eta_3}$ is defined in equation (41) while $R_i(X,Y)$ is a Taylor remainder such that $R_i(X,Y)/\mathcal{D}_3(G_n,G_*) \to 0$ as $n \to \infty$ for $i \in \{3,4\}$. Similarly, we also decompose $B_n$ as

$$B_n = - \sum_{j:|\mathcal{C}_j|=1} \sum_{i \in \mathcal{C}_j} \sum_{|\gamma|=1} \frac{\exp(\beta_{0i}^n)}{\gamma!} (\Delta\beta_{1i}^n) \cdot X^\gamma \exp((\beta_{1j}^*)^\top X) g_{G_n}(Y|X) + R_5(X,Y)$$

$$- \sum_{j:|\mathcal{C}_j|>1} \sum_{i \in \mathcal{C}_j} \sum_{1 \le |\gamma| \le 2} \frac{\exp(\beta_{0i}^n)}{\gamma!} (\Delta\beta_{1i}^n) \cdot X^\gamma \exp((\beta_{1j}^*)^\top X) g_{G_n}(Y|X) + R_6(X,Y),$$

where $R_5(X, Y)$ and $R_6(X, Y)$ are Taylor remainders such that their ratios over $\mathcal{D}_3(G_n, G_*)$ approach zero as $n \to \infty$. Subsequently, let us define

$$S^n_{j, \eta_1, \eta_2, \eta_3} := \sum_{i \in \mathcal{C}_j} \sum_{\alpha \in \mathcal{I}_{\eta_1, \eta_2, \eta_3}} \frac{\exp(\beta^n_{0i})}{\alpha!} \cdot (\Delta \beta^n_{1ij})^{\alpha_1} (\Delta a^n_{ij})^{\alpha_2} (\Delta b^n_{ij})^{\alpha_3} (\Delta \sigma^n_{ij})^{\alpha_4},$$

$$T^n_{j, \gamma} := - \sum_{i \in \mathcal{C}_j} \frac{\exp(\beta^n_{0i})}{\gamma!} (\Delta \beta^n_{1ij})^\gamma = -S^n_{j, \gamma, 0, 0},$$

for any $(\eta_1, \eta_2, \eta_3) \neq (\mathbf{0}_d, 0, 0)$ and $|\gamma| \neq \mathbf{0}_d$, while for $(\eta_1, \eta_2, \eta_3) = (\mathbf{0}_d, 0)$ we set

$$S^n_{i, \mathbf{0}_d, 0} = -T^n_{i, \mathbf{0}_d} := \sum_{i \in \mathcal{C}_j} \exp(\beta^n_{0i}) - \exp(\beta^*_{0i}).$$

As a consequence, it follows that

$$H_n = \sum_{j=1}^K \sum_{\eta_3=0}^{1+\mathbf{1}_{\{|\mathcal{C}_j|>1\}}} \sum_{|\eta_1|+\eta_2=0}^{2(1+\mathbf{1}_{\{|\mathcal{C}_j|>1\}})-\eta_3} S^n_{j,\eta_1,\eta_2,\eta_3} \cdot X^{\eta_1} \exp((\beta^*_{1j})^\top X) \cdot \frac{\partial^{\eta_2} f}{\partial h_1^{\eta_2}}(Y|(a^*_j)^\top X + b^*_j, \sigma^*_j)$$

$$+ \sum_{j=1}^K \sum_{|\gamma|=0}^{1+\mathbf{1}_{\{|\mathcal{C}_j|>1\}}} T^n_{j,\gamma} \cdot X^\gamma \exp((\beta^*_{1j})^\top X) g_{G_n}(Y|X) + R_5(X, Y) + R_6(X, Y). \tag{46}$$

**Step 2 - Non-vanishing coefficients**:

In this step, we will prove by contradiction that at least one among the ratios $S^n_{j,\eta_1,\eta_2,\eta_3}/\mathcal{D}_3(G_n, G_*)$ does not converge to zero as $n \to \infty$. Assume that all these terms go to zero, then by employing arguments for deriving equations (43) and (44), we get that

$$\frac{1}{\mathcal{D}_3(G_n, G_*)} \cdot \Big[ \sum_{j=1}^K \Big| \sum_{i \in \mathcal{C}_j} \exp(\beta^n_{0i}) - \exp(\beta^*_{0j}) \Big|$$

$$+ \sum_{j:|\mathcal{C}_j|=1} \sum_{i \in \mathcal{C}_j} \exp(\beta^n_{0i}) \Big( \|\Delta \beta^n_{1ij}\| + \|\Delta a^n_{ij}\| + |\Delta b^n_{ij}| + |\Delta \sigma^n_{ij}| \Big) \Big] \to 0.$$

Next, let $e_j := (0, \ldots, 0, \underbrace{1}_{j-th}, 0, \ldots, 0)$ for any $j \in [d]$. Then, we have

$$\frac{1}{\mathcal{D}_3(G_n, G_*)} \cdot \sum_{i=1}^K \exp(\beta^n_{0i}) \|\Delta \beta^n_{1ij}\|^2 = \sum_{i=1}^K \sum_{\eta_1 \in \{2e_1, \ldots, 2e_d\}} \frac{|U^n_{j, \eta_1, 0, 0}|}{\mathcal{D}_3(G_n, G_*)} \to 0.$$

Similarly, we also get that

$$\frac{1}{\mathcal{D}_3(G_n, G_*)} \cdot \sum_{i=1}^K \exp(\beta^n_{0i}) \|\Delta b^n_{ij}\|^2 \to 0, \quad \frac{1}{\mathcal{D}_3(G_n, G_*)} \cdot \sum_{i=1}^K \exp(\beta^n_{0i}) \|\Delta \sigma^n_{ij}\|^2 \to 0$$

Moreover, note that

$$\frac{1}{\mathcal{D}_3(G_n, G_*)} \cdot \sum_{i=1}^K \exp(\beta^n_{0i}) \|\Delta a^n_{ij}\|^2 = \sum_{i=1}^K \sum_{\eta_1 \in \{2e_1, \ldots, 2e_d\}} \frac{|U^n_{j, \eta_1, 2, 0}|}{\mathcal{D}_3(G_n, G_*)} \to 0.$$

Gathering all the above limits, we obtain that $1 = \mathcal{D}_3(G_n, G_*)/\mathcal{D}_3(G_n, G_*) \to 0$ as $n \to \infty$, which is a contradiction. Thus, at least one among the terms $S^n_{j,\eta_1,\eta_2,\eta_3}/\mathcal{D}_3(G_n, G_*)$ does not converge to zero as $n \to \infty$

**Step 3 - Fatou's contradiction**:

It follows from the hypothesis that $\mathbb{E}_X[V(\overline{g}_{G_n}(\cdot|X), g_{G_*}(\cdot|X))]/\mathcal{D}_3(G_n, G_*) \to 0$ as $n \to \infty$. Then, by applying the Fatou's lemma, we get

$$0 = \lim_{n \to \infty} \frac{\mathbb{E}_X[V(\overline{g}_{G_n}(\cdot|X), g_{G_*}(\cdot|X))]}{\mathcal{D}_3(G_n, G_*)} = \frac{1}{2} \cdot \int \liminf_{n \to \infty} \frac{|\overline{g}_{G_n}(Y|X) - g_{G_*}(Y|X)|}{\mathcal{D}_3(G_n, G_*)} dX dY,$$

which implies that $|\overline{g}_{G_n}(Y|X) - g_{G_*}(Y|X)|/\mathcal{D}_3(G_n, G_*) \to 0$ as $n \to \infty$ for almost surely $(X, Y)$.

Next, we define $m_n$ as the maximum of the absolute values of $S^n_{j,\eta_1,\eta_2/\mathcal{D}_3(G_n,G_*)}$. It follows from Step 2 that $1/m_n \not\to \infty$. Moreover, by arguing in the same way as in Step 3 in Appendix E.1, we receive that

$$H_n/[m_n\mathcal{D}_3(G_n, G_*)] \to 0 \tag{47}$$

as $n \to \infty$. By abuse of notations, let us denote

$$S^n_{j,\eta_1,\eta_2,\eta_3}/[m_n\mathcal{D}_3(G_n, G_*)] \to \tau_{j,\eta_1,\eta_2,\eta_3}$$

Here, at least one among $\tau_{j,\eta_1,\eta_2,\eta_3}$ is non-zero. Then, by putting the results in equations (46) and (47) together, we get

$$\sum_{j=1}^{K} \sum_{\eta_3=0}^{1+\mathbf{1}_{\{|\mathcal{C}_j|>1\}}} \sum_{|\eta_1|+\eta_2=0}^{2(1+\mathbf{1}_{\{|\mathcal{C}_j|>1\}})-\eta_3} \tau_{j,\eta_1,\eta_2,\eta_3} \cdot X^{\eta_1} \exp((\beta^*_{1j})^\top X) \frac{\partial^{\eta_2+\eta_3} f}{\partial h_1^{\eta_2} \partial \sigma^{\eta_3}}(Y|(a^*_j)^\top X + b^*_j, \sigma^*_j)$$

$$+ \sum_{j=1}^{K} \sum_{|\gamma|=0}^{1+\mathbf{1}_{\{|\mathcal{C}_j|>1\}}} -\tau^n_{j,\gamma,0,0} \cdot X^\gamma \exp((\beta^*_{1j})^\top X) g_{G_*}(Y|X) = 0,$$

for almost surely $(X, Y)$. Arguing in a similar fashion as in Step 3 of Appendix E.1, we obtain that $\tau_{j,\eta_1,\eta_2,\eta_3} = 0$ for any $j \in [K], 0 \le |\eta_1| + \eta_2 + \eta_3 \le 2(1 + \mathbf{1}_{\{|\mathcal{C}_j|>1\}})$ and $0 \le |\gamma| \le 1 + \mathbf{1}_{\{|\mathcal{C}_j|>1\}}$. This contradicts the fact that at least one among them is non-zero. Hence, the proof is completed.

