# OpenReview forum: "Statistical Perspective of Top-K Sparse Softmax Gating Mixture of Experts"
_ICLR.cc/2024/Conference — ICLR 2024 poster_

### Official Review · Reviewer_dwg3 · 2023-10-30

**Soundness:** 3 good
**Presentation:** 3 good
**Contribution:** 3 good
**Rating:** 6
**Confidence:** 1

**Summary:**

Based on the Gaussian mixture expert model, the influence of top-K sparse softmax gating function on density estimation and parameter estimation was analyzed. Novel loss functions and Voronoi metrics were used to characterize the behavior of different regions in the input space.

**Strengths:**

S1. The writing is clear and concise, making it easy to understand theory.

S2. From the author's introduction, this paper is the first to perform convergence analysis for maximum likelihood estimation (MLE) under top-K sparse softmax gated Gaussian MoE. The results of this paper are innovative and helpful in inspiring new expert mixture system designs.

**Weaknesses:**

W1. The theoretical results of this paper lack experimental verification.

W2. It only considers Gaussian mixture expert models and does not consider other types of models.

**Questions:**

Considering W2, I would like to know the difficulties in generalizing the results of this article to MoE methods without Gaussian assumption.

**Details Of Ethics Concerns:**

Nan.

---

> ### Author Response · Authors · 2023-11-15
> **Response to Reviewer dwg3 (Part 1)**
>
> Dear Reviewer dwg3,
>
> Thanks for providing your valuable reviews, and giving **good (3)** grades to the **soundness, presentation** and **contribution** of our paper. We hope that we address your concerns regarding our paper with the responses below.
>
> **Q1: The theoretical results of this paper lack experimental verification.**
>
> Thanks for raising your concern. Actually, we already conducted numerical experiments to empirically verify our theoretical results in Appendix D. In that appendix, we show that the empirical convergence rates of parameter estimation totally match the theoretical rates under both the exact-specified and over-specified settings. Therefore, we would like to refer the reviewer to Appendix D for further details.

---

> ### Author Response · Authors · 2023-11-15
> **Response to Reviewer dwg3 (Part 2)**
>
> **Q2: The paper only considers Gaussian mixture-of-experts models. What are the difficulties in generalizing the results to MoE methods without Gaussian assumption?**
>
> Thanks for your question. We would like to clarify that the proof technique in the paper can be extended to other types of models. The reason that we consider only top-K sparse softmax Gaussian MoE is because it is the most popular one in practice. Per your suggestion, we have generalized the results of the paper to the other settings of MoE when $f$ belongs to the family of location-scale distributions $\mathcal{F}:=\{f(Y|h_1(X,a,b),\sigma):(a,b,\sigma)\in\Theta\}$, where the expert function $h_1(X,a,b):=a^{\top}X+b$ stands for the location, $\sigma$ denotes the scale and $\Theta$ is a compact subset of $\mathbb{R}^d\times\mathbb{R}\times\mathbb{R}_+$. Due to the space limit, we will summarize our findings here, and leave further details, including rigorous proofs, in Appendix E of the current manuscript.
>
> Firstly, we characterize the family $\mathcal{F}$ based on the following notion of strong identifiability:
>
> **Definition 1 (Strong Identifiability).** We say that the family $\mathcal{F}$ is strongly identifiable if $f(Y|h_1(X,a,b),\sigma)$ is twice differentiable w.r.t its parameters and the following assumption holds true:
>
> For any $k\geq 1$ and $k$ pairwise different tuples $(a_1,b_1,\sigma_1),\ldots,(a_k,b_k,\sigma_k)\in\Theta$, if there exist real coefficients
> $\alpha^{(i)}_{\ell_1,\ell_2}$, for $i\in[k*]$ and $0\leq \ell_1+\ell_2\leq 2$, such that
>
> $$
> \sum_{i=1}^{k}\sum_{\ell_1+\ell_2=0}^{2} \alpha^{(i)}_{\ell_1,\ell_2}\cdot\dfrac{\partial^{\ell_1+\ell_2}f}{\partial h_1^{\ell_1}\partial \sigma^{\ell_2}}(Y|(a_i)^{\top}X+b_i,\sigma_i)=0,
> $$
>
> for almost surely $(X,Y)$, then we obtain that $\alpha^{(i)}_{\ell_1,\ell_2}=0$ for any $i\in[k*]$ and $0\leq \ell_1+\ell_2\leq 2$.
>
> **Example.** The families of Student's t-distributions and Laplace distributions are strongly identifiable, while the family of location-scale Gaussian distributions is not. Therefore, it suggests that the location-scale Gaussian distribution is weakly identifiable.
>
> Since the convergence rates of maximum likelihood estimation when $\mathcal{F}$ is a family of Gaussian distributions, which is not strongly identifiable, have already been studied, we focus on the scenario when $\mathcal{F}$ is strongly identifiable. Under that assumption, the density $f(Y|h_1,\sigma)$ is twice differentiable w.r.t its parameters, thus, it is also Lipschitz continuous. As a consequence, the density estimation rates under both the exact-specified and over-specified in Theorem 1 and Theorem 3 still hold true. Therefore, we consider only the parameter estimation problem in the sequel.
>
> **Parameter estimation under the strongly identifiable MoE:** In high level, we need to establish the Total Variation lower bound
>
> $$E_X[V(g_{G}(\cdot|X),g_{G*}(\cdot|X))]\gtrsim \mathcal{D}(G,G*),$$
>
> for any $G\in\mathcal{O}_k(\Omega)$.
>
> Then, this bound together with the density estimation rate in Theorem 1 (resp. Theorem 3) leads to the parameter estimation rates in Theorem 2 (resp. Theorem 4). Here, the key step is to decompose the difference between the density $g_{\widehat{G}_n}(Y|X)$
>
> and the true density $g_{G*}(Y|X)$ into a combination of linearly independent terms using Taylor expansions. Therefore, we have to involve the above notion of strong identifiability, and separate our convergence analysis based on that notion.
>
> **Under the exact-specified settings**, we demonstrate in Theorem 5 that the rates for estimating true parameters
>
> $\exp(\beta^{\ast}_{0i})$,
>
> $\beta^{\ast}_{1i}$ , $a^{\ast}_i$, $b^{\ast}_i$, $\sigma^{\ast}_i$ are of order $O(n^{-1/2})$, which match those in Theorem 2.
>
> **Under the over-specified settings**, we show in Theorem 6 that
> the rates for estimating true parameters $\beta^{\ast}_{1i},a^{\ast}_i,b^{\ast}_i,\sigma^{\ast}_i$, which are fitted by more than one component, are of order $\widetilde{\mathcal{O}}(n^{-1/4})$. Notably, these rates are no longer determined by the solvability of the system of polynomial equations in Eq. (7). Thus, they are significantly faster than those in Theorem 4. The main reason for this improvement is when $\mathcal{F}$ is strongly identifiable, the interaction among expert parameters via the second partial differential equation in Eq. (6) does not occur.
>
> Finally, we would like to emphasize that we consider only Gaussian MoE models in our original manuscript since Gaussian is the most challenging distributions. In particular, as the family of location-scale Gaussian distributions does not satisfy the strong identifiability conditions, we encounter an interaction among parameters via the PDE in Eq. (6), which subsequently leads to the complex system of polynomial equations in Eq. (7). As a result, the convergence analysis for the Gaussian MoE models are much more challenging than that for the strongly identifiable MoE models.

---

> > ### Author Response · Authors · 2023-11-21
> >
> > Dear Reviewer dwg3,
> >
> > We would like to thank you for your thorough review. We hope that our rebuttal addresses your previous concerns regarding our work. However, since the discussion period is going to close soon, please feel free to let us know if you have any further comments on our paper. We would be more than happy to address any additional concerns from you.
> >
> > Thank you again for spending time on the paper, we really appreciate that!
> >
> > Best regards,
> >
> > The Authors

---

### Official Review · Reviewer_X5io · 2023-11-01

**Soundness:** 3 good
**Presentation:** 3 good
**Contribution:** 3 good
**Rating:** 6
**Confidence:** 3

**Summary:**

This paper analyzes Top-K sparse softmax gating mixture of experts. In particular, they focus on Gaussian mixture of experts and present theoretical results on the effect of MoE on density and parameter estimations.

**Strengths:**

* Analysis of MoE is a highly relevant and impactful direction of research.
* The analysis seems thorough and rigorous at a high level and the presented results are interesting.

**Weaknesses:**

* The paper is quite dense, and it would have been helpful to outline high level ideas before delving into notation-heavy math.
* Chen et al. [1] also analyze MoE, but in the deep learning setting. How does the analysis presented here differ from this work?

[1] Towards Understanding Mixture of Experts in Deep Learning https://arxiv.org/pdf/2208.02813.pdf

**Questions:**

1. What are the practical implications of the work, if any, for the use of Top-K sparse MoE as a means of conditional computation?
2. How does this work compare to that of [1]?

[1] Towards Understanding Mixture of Experts in Deep Learning https://arxiv.org/pdf/2208.02813.pdf

---

> ### Author Response · Authors · 2023-11-15
> **Response to Reviewer X5io**
>
> Dear Reviewer X5io
>
> We would like to thank you for the wealth of your comments, and giving **good (3)** grades to the **soundness, presentation** and **contribution** of our paper. Below are our responses to your questions. We hope that we address your concerns regarding our paper.
>
> **Q1: Chen et al. [1] also analyze MoE, but in the deep learning setting. How does the analysis presented here differ from this work?**
>
> Thanks for your question. We would like to highlight the differences between Chen et al. [1] and our work in terms of goals, settings and results as follows:
>
> **1. Goals.**
>
> - Chen et. al. [1] studied the mechanism of the mixture of experts (MoE) layers for deep learning from the perspective of a classification problem. Firstly, they would like to know whether using the mixture of experts yields a higher test accuracy than using a single expert. Secondly, they considered the top-K sparse softmax gating MoE model, where $K=1$. They aimed to explore how the router learns to dispatch the data to the right expert.
>
> - In our work, we aim to understand the effects of the top-K sparse softmax gating function, for arbitrary $K\geq 1$, on the MoE models via the density and parameter estimation problems. Then, we compare the performance of the top-K sparse softmax gating MoE to that of the MoE with dense softmax gating function. Finally, we provide insight into finding the best choices of $K$ based on the theoretical results.
>
> **2. Settings.**
>
> - Chen et. al. [1] sampled the data from a distribution with multiple clusters, and generated their label from the set {$1,-1$} uniformly. Then, they trained a top-1 sparse softmax gating MoE layer based on the data using the gradient descent method to minimize an empirical logistic loss. Here, they used a linear gating network and formulated each expert as a two-layer convolutional neural network.
>
> - In our work, we assume that the data are generated from the ground-truth top-K sparse softmax gating Gaussian MoE model, where the mean of each Gaussian distribution is a linear expert network (an extension to general expert network is possible but will require more complex calculations and we leave that for the future work). Then, we estimate the true density function and true parameters of that model by using the maximum likelihood method. Note that, parameter estimation is important to understand as they directly lead to the estimation of experts and softmax weights.
>
> **3. Results.**
>
> - Chen et. al. [1] demonstrated that the top-K sparse softmax gating mixture of non-linear experts can achieve nearly 100\% test accuracy, while a single expert can only reach a test accuracy of no more than 87.5\% on the data distribution. Moreover, they proved that each expert will be specialized to a specific portion of the data, which is determined by the initialization of the weights. Finally, they figured out that the router can learn the cluster-center features and route the input data to the right experts.
>
> - In our work, we establish the convergence rates of density estimation and parameter estimation under the top-K sparse softmax gating Gaussian MoE model when the true number of experts is known and unknown. From our theoretical results, we realize that using the top-K sparse softmax gating function not only helps scale up the model capacity but also keeps the model performance comparable to that when using the dense softmax gating function. Additionally, we also find out that using the top-K sparse softmax gating function, for $K=1$, leads to faster convergence rates of parameter estimation than other values of $K$. For further details, please refer to our response to your Question 2.
>
> **Q2: What are the practical implications of the work, if any, for the use of Top-K sparse MoE as a means of conditional computation?**
>
> Thanks for your question. We would like to refer the reviewer to our response to the Common Question 1 in the General Response (part 2) for further details.
>
> **Q3: It would have been helpful to outline high level ideas before delving into notation-heavy math.**
>
> Thanks for your suggestion. We would like to refer the reviewer to the Common Question 2 in the General Response (part 2) for further details.
>
> **References**
>
> [1] Z. Chen, Y. Deng, Y. Wu, Q. Gu, Y. Li. Towards understanding mixture of experts in deep learning. In NeurIPS, 2022.

---

> > ### Author Response · Authors · 2023-11-21
> >
> > Dear Reviewer X5io,
> >
> > We would like to thank you for your thorough review. We hope that our rebuttal addresses your previous concerns regarding our work. However, since the discussion period is going to close soon, please feel free to let us know if you have any further comments on our paper. We would be more than happy to address any additional concerns from you.
> >
> > Thank you again for spending time on the paper, we really appreciate that!
> >
> > Best regards,
> >
> > The Authors

---

### Official Review · Reviewer_qHcA · 2023-11-07

**Soundness:** 2 fair
**Presentation:** 2 fair
**Contribution:** 3 good
**Rating:** 6
**Confidence:** 2

**Summary:**

The paper provides theoretical analysis on the convergence properties of sparse softmax gating mixtures of Gaussian experts. The key contributions are proving convergence rates in two cases - when the number of experts is exactly-specified, and when there is an overspecified number of experts.

For the exact-specified case, the authors show a convergence rate of $O(n^{-1/2})$ to the true parameters under maximum likelihood estimation. This is an important theoretical result as it quantifies the sample complexity.

For the overspecified case, the convergence rate depends on the cardinality of the Voronoi cells induced by the gate activations

**Strengths:**

- The theoretical analysis of convergence rates for MoE models is novel and useful. Understanding sample complexity of different MoE architectures is valuable.
- The paper is technically strong, with detailed proofs of the main results. The analysis for the overspecified case considering Voronoi cells is creative.
- The assumptions are clearly laid out, making the results easy to interpret and apply.

**Weaknesses:**

- As noted, the paper is quite dense with heavy notation. More intuition and examples earlier on could make it more accessible.
- The writing in the universal assumptions section is unclear, and should be revised for readability.
- The implications of the theory for practitioners could be expanded on more in the discussion. Guidance on model design is lacking.

**Questions:**

- How do these convergence results compare to prior analysis on softmax vs sparse gating? Does this theory suggest one gating approach over the other?
- Due to the instability of the EM algorithm, there is no error bar provided in Figure 3b. Can we still consider this experiment to be reliable to to justify the theoretical results?

---

> ### Author Response · Authors · 2023-11-15
> **Response to Reviewer qHcA**
>
> Dear Reviewer qHcA,
>
> We would like to thank you for providing your constructive feedback and giving **good (3)** grade to the **contribution** of our paper. We hope that we can address your concerns with the responses below, and eventually convince you to raise your final score.
>
> **Q1: The writing in the universal assumptions section is unclear, and should be revised for readability.**
>
> Thanks for your suggestions. We have revised the universal assumptions in the current manuscript (in blue color) as follows:
>
> **(U.1) Convergence of MLE:** To ensure the convergence of parameter estimation, we assume that the parameter space $\Omega$ is compact subset of $\mathbb{R}\times\mathbb{R}^d\times\mathbb{R}^d\times\mathbb{R}\times\mathbb{R}_+$, while the input space $\mathcal{X}$ is bounded.
>
> **(U.2) Identifiability:** Next, we assume that $\beta^*_{1k_*}=0_d$ and $\beta^*_{0k_*}=0$ so that the top-K sparse softmax gating Gaussian mixture of experts is identifiable. Under that assumption, we show in Appendix C that if $G$ and $G'$ are two mixing measures such that $g_{G}(Y|X)=g_{G'}(Y|X)$ for almost surely $(X,Y)$, then it follows that that $G\equiv G'$. Without that assumption, the result that $G\equiv G'$ does not hold, which leads to unncessarily complicated loss functions (see Proposition 1 in [1]).
>
> **(U.3) Distinct Experts:** To guarantee that experts  in the mixture [1] are different from each other, we assume that parameters $(a^{\ast}_i,b^{\ast}_i,\sigma^{\ast}_i)$, for $i\in[k*]$, are pairwise distinct.
>
> **(U.4) Input-dependent Gating Functions:** To make sure that the gating functions depend on the input $X$, we assume that at least one among parameters $\beta^*_{11},\ldots,\beta^*_{1k_*}$ is different from zero. Otherwise, the gating functions would be independent of the input $X$, which simplifies the problem significantly. In particular, the model (1) would reduce to an input-free gating Gaussian mixture of experts, which was already studied in [2].
>
> **Q2: The implications of the theory for practitioners could be expanded on more in the discussion.**
>
> Thanks for your suggestion. We would like to refer the reviewer to our response to the Common Question 1 in the General Response (Part 2) for further details.
>
> **Q3: How do these convergence results compare to prior analysis on softmax vs sparse gating?**
>
> Thanks for your question. It is worth emphasizing that our paper is the very first work establishing the theoretical guarantee for the effects of top-K sparse gating function on the mixture of experts via the parameter estimation problem. Additionally, the most related work to ours is [1], which established the convergence rates of parameter estimation under the dense softmax gating Gaussian mixture of experts. For the comparison of parameter estimation rates under these two models, please refer to the paragraph "No trade-off between model capacity and model performance" in your Question 2.
>
> **Q4: Guidance on model design is lacking. Does this theory suggest one gating approach over the other?**
>
> Thanks for your question. We would like to refer the reviewer to our response to the Common Question 1 in the General Response (part 2) for further details.
>
> **Q5: Due to the instability of the EM algorithm, there is no error bar provided in Figure 3b. Can we still consider this experiment to be reliable to to justify the theoretical results?**
>
> Thanks for your comment. Indeed, as we mentioned in the Appendix, the EM iterates are unstable, which leads to high error bars. That is why we did not include that in the previous version of our manuscript. In the new version of our manuscript, per your suggestion, we have included error bars for the EM iterates in Figure 3b.
>
> **Q6: More intuition and examples earlier on could make it more accessible.**
>
> Thanks for your suggestion. We would like to refer the reviewer to the Common Question 2 in part 2 of the General Response section for further details.
>
> **References**
>
> [1] H. Nguyen, TT. Nguyen, and N. Ho. Demystifying softmax gating in Gaussian mixture of experts. Advances in Neural Information Processing Systems, 2023a.
>
> [2] N. Ho, C. Y. Yang, and M. I. Jordan. Convergence rates for Gaussian mixtures of experts. Journal of Machine Learning Research, 23(323):1–81, 2022.

---

> ### Comment · Reviewer_qHcA · 2023-11-16
>
> Thank you for the thorough response. I have increased the score to 6.

---

> > ### Author Response · Authors · 2023-11-16
> > **Thank you**
> >
> > We thank Reviewer qHcA for your positive evaluation of our paper after the rebuttal and for increasing the final score to 6. We really appreciate that.
> >
> > Best,
> >
> > The Authors

---

### Author Response · Authors · 2023-11-15
**General Response (Part 2)**

**CQ1: What are the practical implications of the paper for the use of Top-K sparse softmax gating function in MoE models?**

Thanks for your question. There are three main practical implications of our theoretical results for the use of top-K sparse softmax gating function in mixture of experts (we have included them in Section 4 of the current revision of our manuscript):

**1. No trade-off between model capacity and model performance:** In the top-K sparse softmax gating Gaussian mixture of experts, since the gating function is discontinuous and only a portion of experts are activated for each input to scale up the model capacity, the parameter estimation rates under that model are expected to be slower than those under the dense softmax gating Gaussian mixture of experts [1]. However, from our theories it turns out that the parameter estimation rates under these two models are the same under both the exact-specified and over-specified settings. As a result, we point out that using the top-K sparse softmax gating function allows us to scale up the model capacity without sacrificing the computational cost as well as the convergence rates of parameter and density estimation.

**2. Favourable gating function:** As mentioned in the paragraph of Voronoi metric in Section 3, due to an intrinsic interaction between gating parameters and expert parameters via the first partial differential equation in Eq. (6), the rates for estimating those parameters under the over-specified settings are determined by the solvability of the system of polynomial equations in Eq. (7), which are significantly slow. However, if we use the top-1 sparse softmax gating function, i.e. activating only a single expert for each input, then the gating value is either one or zero. As a result, the previous interaction no longer occurs, which helps improve the parameter estimation rates. This partially accounts for the efficacy of top-1 sparse softmax gating mixture of experts in scaling up deep learning architectures (see [2]).


**3. True/ Minimal number of experts:** One key challenge of utilizing top-K sparse mixture of experts in practical applications is to choose the true/ minimal number of experts. This practical problem can be partially addressed using theories developed from the paper. In particular, under the over-specified settings suppose that the MLE $\widehat{G}_n$ consists of $\hat{k}_n$ components.

When the sample size $n$ goes to infinity, every Voronoi cell among $\mathcal{C}^n_1,\ldots,\mathcal{C}^n_{k*}$ contains at least one element. Since the total number of elements of those Voronoi cells is $\hat{k}_n$, the maximum cardinality of a Voronoi cell turns out to be $\hat{k}_n-k*+1$.

This maximum value is attained when there is exactly one ground-truth component $(\beta^{\ast}_{1j},a^{\ast}_j,b^{\ast}_j,\sigma^{\ast}_j)$ fitted by more than one component. An instance for this scenario is when $|\mathcal{C}^n_1|=\hat{k}_n-k*+1$

and $|\mathcal{C}^n_2|=\ldots=|\mathcal{C}^n_{k*}|=1$.

Under this setting, the first true parameters $\beta^*_{11},b^*_1$ and $a^*_1,\sigma^*_1$ achieve their worst possible estimation rates of order $O(n^{-1/2\bar{r}(\hat{k}_n-k*+1)})$ and $O(n^{-1/\bar{r}(\hat{k}_n-k*+1)})$, respectively, which become significantly slow when the difference $\hat{k}_n-k*$ increases. As a consequence, the estimated number of experts $\hat{k}_n$ should not be very large compared to the true number of experts $k*$.

One possible approach to estimating $k_*$ is to reduce $\hat{k}_n$ until these rates reach the optimal order $O(n^{-1/2})$. That reduction can be done by merging close estimated parameters within their convergence rates to the true parameters [3] or by penalizing the log-likelihood function of the top-K sparse softmax gating Gaussian MoE using the differences among the parameters [4]. The detailed methodological approach based on that idea and theoretical guarantee for that approach are deferred to the future work.

**CQ2: Including more intuition and and high-level ideas earlier on could make the paper more accessible.**

Thanks for your suggestion. We have included a paragraph of high-level proof ideas and a table of density estimation and parameter estimation rates in the introduction section to make our work more accessible. Please refer to the revision of our manuscript for further details.

**References**

[1] H. Nguyen. Demystifying softmax gating in Gaussian mixture of experts. Advances in Neural Information Processing Systems, 2023a.

[2] W. Fedus. Switch transformers: Scaling to trillion parameter models with simple and efficient sparsity. Journal of Machine Learning Research, 23:1–39, 2022b.

[3] A. Guha. On posterior contraction of parameters and interpretability in Bayesian mixture modeling. Bernoulli, 27(4):2159–2188, 2021.

[4] T. Manole. Estimating the number of components in finite mixture models via the group-sort-fuse procedure. The Annals of Statistics, 49(6):3043–3069, 2021.

---

### Author Response · Authors · 2023-11-15
**General Response (Part 1)**

Dear AC and reviewers,

Thanks for your thoughtful reviews and valuable comments, which have helped us improve the paper substantially. We are encouraged by the endorsement that:

**1) Contributions:** Our convergence analysis for mixture-of-experts (MoE) models is **novel and valuable** (Reviewer qHcA), **highly relevant and impactful** direction of research (Reviewer X5io), **innovative and helpful** in inspiring new expert mixture system designs (Reviewer dwg3).

**2) Soundness:** The paper is **technically strong** with detailed proofs of the main results (Reviewer qHcA). The analysis is **thorough and rigorous** at a high level (Reviewer X5io).

**3) Presentation:** The assumptions are clearly laid out, making the results **easy to interpret and apply** (Reviewer qHcA). The writing is **clear and concise**, making it **easy to understand** the theory (Reviewer dwg3).

There are two main concerns from the reviewers:

1) The practical implications of our work for the use of Top-K sparse softmax gating function in mixture of experts (MoE) models;

2) Including more intuition and and high-level ideas earlier on could make the paper more accessible.

We will address these two concerns in part 2 of the general response. Moreover, we have included these changes to the revision of our manuscript.

---

### Meta-Review · Area_Chair_1AHr · 2023-12-11

**Metareview:**

This paper establishes theoretical foundation on the effects of the top-K sparse softmax gating function on both density and parameter estimations. In particular, for Gaussian mixture of experts, the paper provides convergence rates in (1) case where number of experts is exactly-specified and (2) case where number of experts is overspecified. In the former case, convergence rates that are parametric on the sample size are derived. Overall, the paper makes interesting theoretical contributions to this problem.

**Justification For Why Not Higher Score:**

The practical implications of the work haven't been discussed and highlighted sufficiently in the paper. It will be valuable to address this concern carefully in the paper.

**Justification For Why Not Lower Score:**

The paper takes a good first step towards theoretically understanding an important problem.

---

### Decision · Program_Chairs · 2024-01-16

Accept (poster)